# A Machine-Learned Comorbidity Index

**Suleman Baloch** [1]  **Kishlay Jha** [2]  **Alberto M. Segre** [1]  **Philip M. Polgreen** [3]  **Bijaya Adhikari** [1]

## Abstract

Traditional comorbidity scores (e.g., Charlson and Elixhauser) are widely used for risk adjustment and patient stratification, but they have two key limitations: (i) they are largely mortality-centric and do not align well with other clinical outcomes, and (ii) their linear, rule-based structure cannot capture nonlinear, outcome-specific risk relationships. We propose a Machine-Learned Comorbidity Index (MLCI) that maps diagnosis codes to a single scalar by maximizing the normalized Hilbert–Schmidt Independence Criterion (nHSIC) between the learned score and multiple clinical outcomes. MLCI captures nonlinear risk–outcome dependence and is supported by a theory that characterizes when a unified, informative admission-level ordering can be achieved across outcomes. Empirical results on multiple benchmark electronic health record (EHR) datasets show that MLCI outperforms strong baselines across multiple evaluation metrics.

## 1. Introduction

We define **comorbidity** as the overall severity and complexity of the diagnoses recorded for a hospital admission, reflected in diagnosis codes. Comorbidity scores are widely used for patient stratification (Ening et al., 2015), risk stratification (O'Hara et al., 2024), and risk adjustment (Ou et al., 2012; Quan et al., 2011). These applications rely largely on hand-engineered comorbidity indices that compress diagnosis information into a single scalar score. However, traditional indices such as Charlson (Charlson et al., 1987) and Elixhauser (Elixhauser et al., 1998; Van Walraven et al., 2009) have two key limitations.

First, as risk indicators, these indices were designed for mortality and often generalize poorly to other clinical outcomes because they rely on predefined comorbidity categories and mortality-calibrated fixed weights. This undermines the implicit assumption behind comorbidity-based stratification: that a patient's diagnosis burden during a hospital admission yields a shared ordering of admissions that is meaningful across multiple clinical outcomes (Byles et al., 2005). For example, admissions involving "sicker" patients typically face higher risk across outcomes such as mortality and ICU-level care than admissions involving "less sick" patients (Kuswardhani et al., 2020). However, existing indices lack a principled way to learn a data-driven severity ordering that is consistent across these outcomes. Clinicians also need more than a ranking: they need a cutoff that identifies high-severity admissions for intervention (Billings et al., 2006; Patel et al., 2021). Traditional indices identify such high-risk groups using mortality-calibrated thresholds, rather than thresholds learned to reflect consistent risk across outcomes.

Second, their linear, rule-based nature limits their ability to capture nonlinear relationships between risk and outcomes. Risk can change nonlinearly with comorbidity burden (Wei et al., 2023; Ly et al., 2025), as certain diagnosis combinations may amplify risk (Willadsen et al., 2018) while additional diagnoses can have diminishing impact at high baseline severity. Clinical risk tools often reflect this nonlinear structure: SAPS II maps severity to mortality through a logistic-style link, while NEWS uses score cutoffs to flag patients at higher risk of rapid deterioration and severe outcomes (Le Gall et al., 2005; Kim et al., 2021).

Together, these limitations motivate three central questions that guide our approach:

1. To what extent do common hospital outcomes share an underlying admission-level severity ordering, so that a single score can rank admissions consistently across outcomes?

2. If such an ordering exists, can we learn it in a principled, data-driven way while modeling outcome-specific nonlinear severity–risk links?

3. Beyond ranking admissions, can we learn a cutoff that isolates a high-severity group consistently across outcomes?

[1]Department of Computer Science, University of Iowa, Iowa, USA [2]Department of Electrical and Computer Engineering, University of Iowa, Iowa, USA [3]Department of Internal Medicine, University of Iowa, Iowa, USA. Correspondence to: Suleman Baloch <suleman-baloch@uiowa.edu>, Bijaya Adhikari <bijaya-adhikari@uiowa.edu>.

*Proceedings of the 43rd International Conference on Machine Learning*, Seoul, South Korea. PMLR 306, 2026. Copyright 2026 by the author(s).

Answering these questions requires a model that produces a single severity score while capturing nonlinear diagnosis-code interactions. A multi-endpoint kernel dependence objective lets the score jointly capture nonlinear associations with clinical outcomes, thereby learning shared outcome-relevant signal beyond linear correlation.

To address these challenges, we propose **MLCI**, A MACHINE-LEARNED COMORBIDITY INDEX that maps the diagnosis-code representation $X_i$ for admission $i$ to a scalar score $s_i = s_\theta(X_i) \in \mathbb{R}$. MLCI is trained to recover a shared, clinically useful admission-level ordering when such an ordering exists, while allowing each outcome to have its own nonlinear severity–risk relationship. It uses normalized Hilbert–Schmidt Independence Criterion (nHSIC), a kernel-based, scale-comparable dependence measure across outcomes. We learn $s_i = s_\theta(X_i)$ by maximizing normalized dependence with multiple clinical endpoints, so the score captures shared outcome-relevant signal without being dominated by any single endpoint. After training, we estimate separate risk curves for each outcome, mapping the shared score to outcome-specific risk.

We motivate our approach by a **theoretical analysis** that characterizes when a single learned comorbidity score can serve as an approximate shared ordering across multiple clinical outcomes. In the theory, the finite-sample units are hospital admissions. We posit that each admission $i$ has an unobserved latent scalar severity $z_i$, representing admission-level comorbidity burden. The ordering of the $z_i$'s represents a ranking of admissions by latent severity. Because outcomes may depend on severity through unknown, potentially nonlinear relationships, the goal is to recover the ordering of the $z_i$'s rather than their absolute scale. The learned score $s_\theta(X_i)$ is therefore intended to preserve this latent admission-level ordering from diagnosis features, so that admissions with larger latent severity receive larger learned scores. Under this view, maximizing nHSIC compares admissions pairwise. On the score side, it asks which admissions are close under the learned score. On the label side, it asks which admissions have the same binary outcome. A high nHSIC value means these two pairwise patterns agree: admissions close in score tend to have similar labels. We then study what happens when this comparison is made across several outcomes. For each clinical outcome, we form a binary label vector over admissions, whose entries indicate whether that endpoint occurred for a given admission; we then center and stack these vectors across outcomes. If the stacked matrix is approximately rank one, then the outcomes share a dominant admission-level direction. This direction gives a simple threshold rule: choose the admission split whose centered step vector is most aligned with the shared direction. We use the strength of this shared component and the implied threshold split as diagnostics for shared ordering, rather than outcome-specific deviations.

Overall, the main contributions of this paper are summarized as follows:

1. **A novel single-score machine-learned comorbidity index.** We introduce MLCI, a data-driven comorbidity score that learns a shared admission-level latent risk by maximizing nHSIC with multiple clinical outcomes, capturing nonlinear effects in a one-dimensional summary.

2. **Theory for shared severity ordering and threshold stratification.** To our knowledge, this is the first finite-sample analysis linking multi-outcome nHSIC to a shared monotone admission-level ordering. When outcomes share a dominant severity signal, the objective identifies a common admission-level direction and motivates a principled high-severity cutoff, which we evaluate on MIMIC-III and MIMIC-IV.

3. **Consistent gains in dependence metrics.** On MIMIC-III/IV, MLCI shows the strongest score–outcome dependence, outperforming strong single-index clinical baselines in statistical dependence measures.

## 2. Related Work

Comorbidity indices are a longstanding foundation for clinical risk modeling. The Charlson Comorbidity Index (CCI) (Charlson et al., 1987) assigns fixed weights to predefined chronic conditions to predict mortality. Elixhauser measures expand the CCI with more diagnosis-derived conditions and often improve inpatient outcome prediction (Elixhauser et al., 1998). Van Walraven et al. distilled them into a single scalar score for in-hospital mortality that improves usability while retaining strong discrimination (Van Walraven et al., 2009). A common alternative to rule-based indices treats diagnosis codes as high-dimensional sparse features and learns outcome models directly from data; classical machine-learning methods remain competitive for clinical outcome prediction: logistic regression is interpretable yet can match complex diagnosis-code mortality models (Cowling et al., 2021); gradient-boosted trees capture nonlinearities (Li et al., 2025); and factorization machines model low-rank sparse-code interactions for multi-task prediction (Yin et al., 2025). More recently, neural models also embed diagnosis codes and learn nonlinear aggregations, including attention over ICD codes for early length of stay and in-hospital mortality prediction (Liu et al., 2020; Harerimana et al., 2021). Most clinical prediction models optimize a single-outcome likelihood, producing task-specific representations. A complementary approach is to learn representations by maximizing dependence with different outcomes. HSIC is a kernel-based dependence measure with an empirical centered-Gram estimator enabling sample-based independence testing (Gretton et al., 2005; 2007). Cen-

tered kernel alignment connects dependence measurement to normalized, scale-invariant HSIC-style criteria (Cortes et al., 2012), and HSIC has also been used directly as a dependence-maximization training signal, including self-supervised objectives that optimize kernel dependence between views (Li et al., 2021). HSIC is also used for nonlinear feature selection and biomarker discovery in prediction, reducing redundancy and dimensionality in clinical domains (Takahashi et al., 2020; Yu et al., 2023; Dai et al., 2025).

## 3. Preliminaries

**Comorbidity indices.** The two widely used comorbidity indices in the current literature are the Charlson Comorbidity Index (CCI) and the van Walraven-weighted Elixhauser Comorbidity Index (ECI). We implemented both indices using Quan et al.'s published coding algorithms (Quan et al., 2005). Detailed notation and scoring formulas are provided in Appendix D. To make the connection between classic indices and our approach explicit, we describe CCI and ECI through a common input–output lens. Both CCI and the van Walraven ECI map an admission's diagnosis-code representation $X_i$, restricted to ICD codes, to a scalar index score $c_i \in \mathbb{R}$ by forming comorbidity-category indicators from $X_i$ and summing them with fixed weights to summarize baseline illness burden.

## 4. Problem Formulation

Our goal is to develop a *machine-learned comorbidity score* that preserves the CCI/ECI input/output contract: given admission features $X_i$ for admission $i$, output a single scalar score $s_i := s_\theta(X_i) \in \mathbb{R}$, where $s_\theta$ is a neural network mapping admission features to a real-valued comorbidity score. Following the single-index philosophy of CCI/ECI, we aim for $s_\theta(X_i)$ to provide a *common admission-level ordering* that captures shared comorbidity burden across multiple clinical outcomes. We observe hospital admissions indexed by $i \in \{1, \ldots, n\}$. Each admission has diagnosis-code features $X_i$ and $T$ binary clinical outcomes. For task $t \in [T] := \{1, \ldots, T\}$, write $y_i^{(t)} \in \{0, 1\}$ for the outcome label of admission $i$ on task $t$. Outcomes may be missing; let $M_i^{(t)} \in \{0, 1\}$ indicate whether $y_i^{(t)}$ is observed.

**Shared latent severity model.** Comorbidity scores, while designed specifically for mortality, are commonly used to quantify the risks of other clinical outcomes such as ICU transfer (Katz et al., 2023). Based on this practice, we posit that each admission $i$ has an unobserved real-valued latent severity $z_i \in \mathbb{R}$, representing overall sickness or disease burden. Task $t$ has an outcome-specific response curve $\Pr\{y_i^{(t)} = 1 \mid z_i\} = f_t(z_i)$, where $f_t : \mathbb{R} \to (0, 1)$ is not assumed to have a parametric form. Thus, different outcomes may have different prevalence, onset, saturation, and noise

patterns, while still sharing a common one-dimensional severity signal. The learned score $s_\theta(X_i)$ is intended to recover this shared signal at the level of ordering. That is, after an optional global sign flip, it should be monotone in the realized latent severity: $z_i < z_j \implies s_\theta(X_i) \leq s_\theta(X_j)$, allowing ties. This matches the role of comorbidity indices in practice: they provide a scalar ordering of admissions rather than a separate risk model for each endpoint.

We interpret larger learned scores as indicating greater latent severity, but we do not require the observed binary labels to be perfectly monotone in the score. Clinical outcomes are noisy, and different outcomes may activate at different parts of the severity range. The theory therefore studies an ideal monotone oracle score $r$ that preserves the latent severity ordering, while the learned score $s_\theta(X_i)$ is treated as a noisy, data-driven approximation to that ordering.

**Learning from multiple outcomes under heterogeneity.** Training on a single outcome, such as mortality, can yield an outcome-specific score that may not generalize to other clinical outcomes. At the same time, naive multi-task training can suffer from interference or domination by one task because outcomes differ in prevalence, noise level, and severity-response pattern. We therefore treat outcomes as heterogeneous signals of a shared latent severity axis, with task-specific residual structure. This motivates learning one scalar score that captures the shared axis while preventing any single outcome from dominating.

## 5. Methodology

We learn a *single* scalar comorbidity/severity score $s_i = s_\theta(X_i) \in \mathbb{R}$ from each admission's diagnosis-code representation $X_i$. Diagnoses are represented as variable-length collections of ICD-prefix tokens (ICD-9 in MIMIC-III, ICD-10 in MIMIC-IV). This single-index representation compresses diagnosis-code information into one score that is evaluated for dependence with each outcome $y_i^{(1)}, \ldots, y_i^{(T)}$.

**Diagnosis representation.** Each raw ICD code is normalized by converting to uppercase and removing punctuation and whitespace, then mapped to a prefix token using the first $k = 4$ characters. We build a diagnosis vocabulary from the training split only and augment it with <PAD> (index 0) and <UNK> (index 1). Using this fixed training vocabulary, we map diagnoses in the train/val/test splits to a variable-length token collection $X_i = \{C_{i,1}, \ldots, C_{i,m_i}\}$, where $m_i$ is the number of retained diagnosis tokens after truncation and $D_{\max} = 256$ is the maximum retained length.

**Permutation-invariant encoder.** We implement $s_\theta$ using a DeepSets-style encoder (Zaheer et al., 2017). Let $e_j \in \mathbb{R}^d$ be an embedding of token $j$, and $\phi : \mathbb{R}^d \to \mathbb{R}^d$ be an elementwise MLP applied to each token, producing $h_j = \phi(e_j)$. We aggregate token features using masked mean pooling

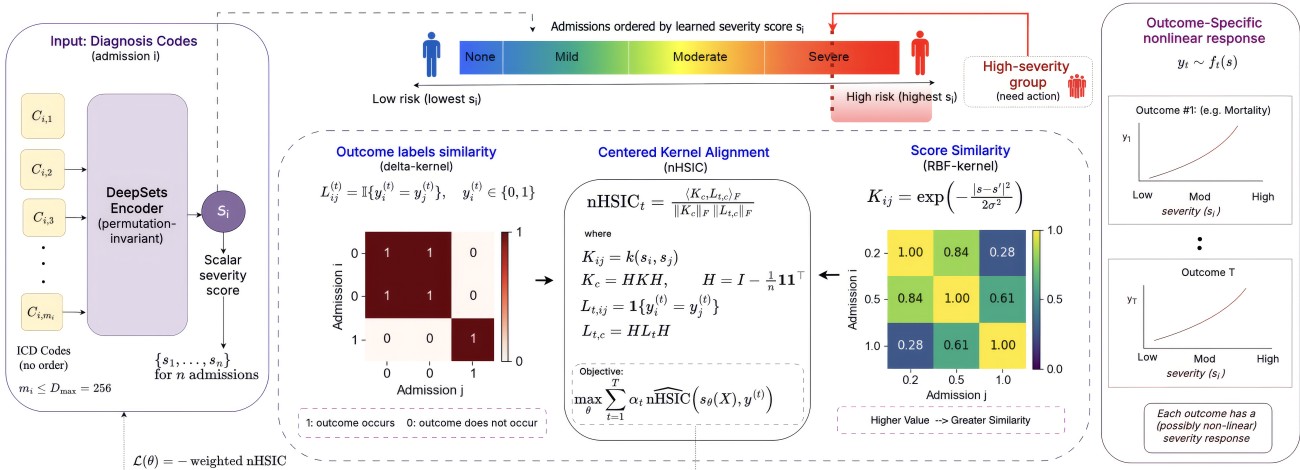

*Figure 1.* MLCI: learning a shared severity score via multi-outcome nHSIC (centered kernel alignment).

and masked max pooling and concatenate the results to form an admission representation, then map it to a scalar through a second MLP $\rho$: $s_i = s_\theta(X_i) = \rho\Big(\mathrm{Agg}\big(\{\phi(e_j)\}\big)\Big) \in \mathbb{R}$. Architecture and training details are in Appendix A.

**Motivation for outcome-aligned dependence objective.** We assume multiple binary outcomes share a latent severity signal but may follow distinct, possibly nonlinear response curves. Thus linear correlation can miss score–outcome dependence. We train $s_\theta(X)$ with a kernel dependence objective based on normalized HSIC (nHSIC), which measures general (linear or nonlinear) dependence via kernels.

**Normalized HSIC (nHSIC).** To compute nHSIC on a mini-batch of size $n_b$, with scores $\{s_i\}_{i=1}^{n_b}$ and task labels $\{y_i^{(t)}\}_{i=1}^{n_b}$, define $K_{ij}^{(b)} := k(s_i, s_j)$, $L_{ij}^{(b,t)} := \mathbb{I}\{y_i^{(t)} = y_j^{(t)}\}$, and the mini-batch centering matrix $H_b := I - \frac{1}{n_b}\mathbf{1}\mathbf{1}^\top$. The centered mini-batch Gram matrices are $K_c^{(b)} := H_b K^{(b)} H_b$, $L_{t,c}^{(b)} := H_b L^{(b,t)} H_b$. We optimize the normalized criterion

$$\widehat{\mathrm{nHSIC}}\Big(s, y^{(t)}\Big) = \frac{\langle K_c^{(b)}, L_{t,c}^{(b)}\rangle_F}{\max\{\|K_c^{(b)}\|_F, \varepsilon_0\}\,\|L_{t,c}^{(b)}\|_F}, \quad (1)$$

where $\varepsilon_0 > 0$ floors $\|K_c^{(b)}\|_F$ for numerical stability.

**Instantiation.** We use the Gaussian RBF kernel on the scalar score, $k_\sigma(s, s') = \exp\Big(-\frac{|s-s'|^2}{2\sigma^2}\Big)$, and the delta label Gram matrix $L^{(b,t)}$ defined above. We set the RBF bandwidth $\sigma$ using a stabilized median heuristic on training scores (Appendix A.5).

**Missing labels and validity.** Let $M_i^{(t)} \in \{0,1\}$ indicate whether label $y_i^{(t)}$ is observed. MLCI uses the intersection-valid cohort, $M_i^\cap = \prod_{t=1}^T M_i^{(t)} = 1$, for multi-task nHSIC training, validation, and model selection; see Appendix A.2.

BCE baselines use task-specific masks because their losses decompose by outcome. For fair per-outcome comparison, Tables 1–2 evaluate every model on the same task-specific valid test admissions for that outcome; see Appendix A.10.

**Task-weighted multi-task dependence training.** We train a single model to produce one shared score $s_i = s_\theta(X_i)$ that is simultaneously dependent on multiple binary outcome vectors $y^{(1)}, \ldots, y^{(T)}$. Our training objective maximizes a weighted sum of per-task nHSIC values:

$$\max_\theta \sum_{t=1}^T \alpha_t \widehat{\mathrm{nHSIC}}\Big(s_\theta(X), y^{(t)}\Big), \quad (2)$$

where $s_\theta(X)$ denotes the vector of scores in the current mini-batch, $y^{(t)}$ denotes the corresponding mini-batch labels for task $t$, and $\alpha_t \geq 0$ controls each task's contribution.

**Two-stage weighting.** Outcomes vary in prevalence and alignment with the shared signal, so unweighted training can be dominated by a few tasks. We set task weights from single-task nHSIC baselines: (Stage 1) train one model per outcome and record best validation nHSIC $\widehat{h}_t$; (Stage 2) train a new multi-task model with stabilized inverse-strength weights

$$\alpha_t \propto \left(\frac{\widehat{h}_{\max}}{\max(\widehat{h}_t, \varepsilon_{\mathrm{wt}})}\right)^{\gamma_{\mathrm{wt}}}, \qquad \widehat{h}_{\max} = \max_t \widehat{h}_t, \quad (3)$$

with $\varepsilon_{\mathrm{wt}} > 0$ and $\gamma_{\mathrm{wt}} \in (0,1)$. We clip and renormalize for robustness (Appendix A).

**Score orientation.** Because the RBF kernel depends only on pairwise score distances, the objective is sign-invariant in $s$. For reporting, we orient $s$ using validation mortality: if the Pearson correlation between $s_i$ and $y_i^{(\mathrm{mort})}$ on validation admissions is negative, we set $s_i \leftarrow -s_i$ for all admissions, so larger scores correspond to higher mortality risk.

# 6. Theoretical Analysis

Let us assume $n$ admissions in a given system, such as a hospital system, have latent severities ordered as $z_1 < \cdots < z_n$. An oracle score is a bounded nondecreasing map $r : \{z_1, \ldots, z_n\} \to [-B, B]$, $\quad z_i < z_j \implies r(z_i) \leq r(z_j)$. For each ordered admission $i$, define $r_i := r(z_i)$. Thus the oracle score induces the finite score vector $r = (r_1, \ldots, r_n)^\top \in [-B, B]^n$, with $r_1 \leq r_2 \leq \cdots \leq r_n$. The theory studies nHSIC objectives between this finite monotone score vector and the observed binary label vectors. The goal is to characterize when the multi-outcome nHSIC objective supports a shared severity ordering across outcomes. The argument has three steps. First, each binary outcome is converted into a centered label profile over admissions. Second, these profiles are stacked across tasks, producing an admission-to-admission label-alignment matrix. Third, when the stacked label profiles are approximately rank one, the leading admission-level direction $v$ yields an explicit monotone threshold rule: choose the admission split whose centered step vector is most aligned with $v$.

**Oracle setup.** For a fixed $B > 0$, define the finite-sample monotone score class $\mathcal{R}_B^\uparrow := \{r \in [-B, B]^n : r_1 \leq \cdots \leq r_n\}$. For $T$ outcomes, write $[T] := \{1, \ldots, T\}$. For each task $t \in [T]$, let $y^{(t)} = (y_1^{(t)}, \ldots, y_n^{(t)})^\top \in \{0, 1\}^n$ be the observed binary label vector across the $n$ admissions. Define the delta label Gram matrix $(L_t)_{ij} := \mathbb{I}\{y_i^{(t)} = y_j^{(t)}\}$, where $\mathbb{I}\{\cdot\}$ denotes the indicator function. For scores, use a bounded continuous positive semidefinite radial kernel $k(x, x') = \kappa(|x - x'|)$, where $\kappa$ is strictly decreasing on $[0, 2B]$. Define $K_{ij}(r) := k(r_i, r_j) = \kappa(|r_i - r_j|)$. Let $H := I - \frac{1}{n}\mathbf{1}\mathbf{1}^\top$, where $\mathbf{1} \in \mathbb{R}^n$ is the all-ones vector. This is the centering matrix. The centered score Gram matrix is $K_c(r) := HK(r)H$. Since $K(r) \succeq 0$ for every $r \in \mathcal{R}_B^\uparrow$, it follows that $K_c(r) \succeq 0$. The centered label Gram matrix for task $t$ is $L_{t,c} := HL_tH$. The per-task normalized HSIC objective between the oracle score vector and task $t$'s labels is

$$\text{nHSIC}\left(r, y^{(t)}\right) := \frac{\langle K_c(r), L_{t,c}\rangle_F}{\|K_c(r)\|_F \|L_{t,c}\|_F}.$$

We use the convention $\text{nHSIC}(r, y^{(t)}) = 0$ whenever either denominator factor is zero. The multi-task objective is

$$J(r) := \sum_{t=1}^T \alpha_t \, \text{nHSIC}\left(r, y^{(t)}\right), \qquad \alpha_t > 0,$$

with fixed task weights $\alpha_t > 0$.

**Link to the learned score.** The analysis is stated for deterministic oracle scores $r \in \mathcal{R}_B^\uparrow$. The learned score vector $s = (s_1, \ldots, s_n)^\top$, with $s_i = s_\theta(X_i)$, is a noisy, data-driven proxy for $r$. Appendix B.6 shows that when $K_c(s)$ concentrates around a nondegenerate mean centered kernel,

the realized learned-score $\text{nHSIC}(s, y^{(t)})$ is close to the normalized HSIC computed from that averaged centered kernel and the task labels.

**From binary labels to an admission-level label-alignment matrix.** Assume each included task is nonconstant. Define the centered label profile $\ell^{(t)} := Hy^{(t)}$. Then $\|\ell^{(t)}\|_2 > 0$. For binary labels under the delta kernel, Appendix B.1 shows that each centered label Gram matrix has the rank-one form $L_{t,c} = 2\ell^{(t)}\ell^{(t)\top}$. Therefore, $\|L_{t,c}\|_F = 2\|\ell^{(t)}\|_2^2$. Whenever $\|K_c(r)\|_F > 0$,

$$\text{nHSIC}\left(r, y^{(t)}\right) = \frac{\ell^{(t)\top}K_c(r)\ell^{(t)}}{\|K_c(r)\|_F \|\ell^{(t)}\|_2^2}.$$

When $\|K_c(r)\|_F = 0$, we keep the convention $\text{nHSIC}(r, y^{(t)}) = 0$. So each task contributes a quadratic form of $K_c(r)$ along its centered label vector $\ell^{(t)}$. Including the fixed task coefficient $\alpha_t$, whenever $\|K_c(r)\|_F > 0$, the multi-task objective becomes

$$J(r) = \frac{1}{\|K_c(r)\|_F} \sum_{t=1}^T \frac{\alpha_t}{\|\ell^{(t)}\|_2^2} \ell^{(t)\top}K_c(r)\ell^{(t)}.$$

The purpose of the following stacking is to establish the equality between this weighted sum of task-wise quadratic forms and a single Frobenius inner product. To simplify notation, combine the fixed task coefficient with the nHSIC normalization by defining $\beta_t := \alpha_t / \|\ell^{(t)}\|_2^2$. Now form the stacked label-profile matrix $\widetilde{W} \in \mathbb{R}^{T \times n}$, whose $t$-th row is the weighted centered label vector $\sqrt{\beta_t}\,\ell^{(t)\top}$. Equivalently,

$$\widetilde{W} = \begin{bmatrix} \sqrt{\beta_1}\ell^{(1)\top} \\ \sqrt{\beta_2}\ell^{(2)\top} \\ \vdots \\ \sqrt{\beta_T}\ell^{(T)\top} \end{bmatrix}.$$

By construction, $\widetilde{W}^\top\widetilde{W} = \sum_{t=1}^T \beta_t\,\ell^{(t)}\ell^{(t)\top}$. Therefore, $\left\langle K_c(r), \widetilde{W}^\top\widetilde{W} \right\rangle_F = \sum_{t=1}^T \beta_t \left\langle K_c(r), \ell^{(t)}\ell^{(t)\top} \right\rangle_F$. Since $\left\langle K_c(r), \ell^{(t)}\ell^{(t)\top} \right\rangle_F = \ell^{(t)\top}K_c(r)\ell^{(t)}$, we obtain

$$\left\langle K_c(r), \widetilde{W}^\top\widetilde{W} \right\rangle_F = \sum_{t=1}^T \frac{\alpha_t}{\|\ell^{(t)}\|_2^2} \ell^{(t)\top}K_c(r)\ell^{(t)}.$$

Hence, for $\|K_c(r)\|_F > 0$, the multi-task objective can be written as

$$J(r) = \frac{\langle K_c(r), \widetilde{W}^\top\widetilde{W}\rangle_F}{\|K_c(r)\|_F}. \tag{4}$$

When $\|K_c(r)\|_F = 0$, we keep the convention $J(r) = 0$.

The matrix $\widetilde{W}^\top\widetilde{W}$ is a cross-task label-alignment matrix: its $(i, j)$-entry compares admissions $i$ and $j$ using all centered

label profiles across tasks, $(\widetilde{W}^\top \widetilde{W})_{ij} = \sum_{t=1}^{T} \beta_t \ell_i^{(t)} \ell_j^{(t)}$. Thus it summarizes how similarly two admissions behave across all weighted outcomes. The score $r$ enters only through $K_c(r)$, and the objective asks this score-similarity matrix to align with the combined cross-task label structure. We then ask whether this combined label structure is mainly driven by one unified admission direction.

**Rank-one projection to a shared admission direction.** Let $\widetilde{W}_1 = \sigma_1 uv^\top$ be the best rank-one approximation of $\widetilde{W}$, with $\|u\|_2 = \|v\|_2 = 1$. Here $u \in \mathbb{R}^T$ is the task-side direction and $v \in \mathbb{R}^n$ is the admission-level direction. When $\sigma_1 > 0$, $v$ is a leading eigenvector of $\widetilde{W}^\top \widetilde{W}$, with eigenvalue $\sigma_1^2$, and captures the dominant admission-level pattern shared across tasks. If $\sigma_1 = 0$, then $\widetilde{W} = 0$, and the projected objective below is identically zero. Define the projected objective

$$J_1(r) := \frac{\langle K_c(r), \widetilde{W}_1^\top \widetilde{W}_1 \rangle_F}{\|K_c(r)\|_F}.$$

We set $J_1(r) = 0$ whenever $\|K_c(r)\|_F = 0$.

**Lemma 6.1** (Reduction under rank-one projection). *For every $r \in \mathcal{R}_B^\uparrow$ with $\|K_c(r)\|_F > 0$, $J_1(r) = \sigma_1^2 \bar{\Delta}_v(r)$, where*

$$\bar{\Delta}_v(r) := \frac{v^\top K_c(r) v}{\|K_c(r)\|_F}.$$

*Therefore, if $\sigma_1 > 0$, $\arg\max_{r \in \mathcal{R}_B^\uparrow : \|K_c(r)\|_F > 0} J_1(r) = \arg\max_{r \in \mathcal{R}_B^\uparrow : \|K_c(r)\|_F > 0} \bar{\Delta}_v(r)$. If $\sigma_1 = 0$, then $J_1 \equiv 0$.*

*Proof sketch.* Since $\widetilde{W}_1 = \sigma_1 uv^\top$ with $\|u\|_2 = 1$, $\widetilde{W}_1^\top \widetilde{W}_1 = \sigma_1^2 vv^\top$. Substituting this into $J_1(r)$ gives

$$J_1(r) = \sigma_1^2 \frac{\langle K_c(r), vv^\top \rangle_F}{\|K_c(r)\|_F} = \sigma_1^2 \frac{v^\top K_c(r) v}{\|K_c(r)\|_F}.$$

The maximizer statement follows because multiplication by the positive constant $\sigma_1^2$ does not change the argmax. If $\sigma_1 = 0$, then $\widetilde{W}_1^\top \widetilde{W}_1 = 0$, so $J_1 \equiv 0$. $\square$

Thus, under a rank-one shared label structure, the projected multi-task objective reduces to aligning the centered score Gram matrix with $v$. This same direction determines the monotone admission-level cutoff whose centered step vector is most aligned with the shared signal.

**Threshold certificate from the shared direction.** For a split $j \in \{1, \ldots, n-1\}$, define $L_j := \{1, \ldots, j\}$, $R_j := \{j+1, \ldots, n\}$. Let $g_j := H\mathbf{1}_{R_j}$. Here $g_j$ is the centered indicator of the candidate upper-tail group.

**Lemma 6.2** (Global bound and threshold certificate). *Fix any centered target direction $w \in \mathbb{R}^n$, so $\mathbf{1}^\top w = 0$. This lemma is stated for a general direction $w$; after the lemma we apply it to the centered shared admission direction $v$. Define*

$$\bar{\Delta}_w(r) := \frac{w^\top K_c(r) w}{\|K_c(r)\|_F},$$

*with value 0 whenever $\|K_c(r)\|_F = 0$. If $w = 0$, then $\bar{\Delta}_w \equiv 0$, and the statements are trivial. Assume $w \neq 0$ below. Then $\bar{\Delta}_w(r) \leq \|w\|_2^2$ for all $r \in \mathcal{R}_B^\uparrow$. Moreover, consider a nontrivial two-level threshold score on the split $L_j/R_j$, meaning a score $r^{(j)} \in \mathcal{R}_B^\uparrow$ of the form*

$$r_i^{(j)} = \begin{cases} a, & i \in L_j, \\ b, & i \in R_j, \end{cases} \qquad \text{with } a < b.$$

*That is, all admissions below the cutoff receive one score level, and all admissions above the cutoff receive another score level. For any such threshold score, we have the exact value*

$$\bar{\Delta}_w(r^{(j)}) = \|w\|_2^2 \rho_j^2, \qquad \rho_j := \frac{w^\top g_j}{\|w\|_2 \|g_j\|_2}.$$

*Thus the best two-level threshold is obtained by $j^\star \in \arg\max_{1 \leq j \leq n-1} \rho_j^2$. The achieved value certifies a fraction $\rho_\star^2 := \max_{1 \leq j \leq n-1} \rho_j^2$ of the global upper bound $\|w\|_2^2$.*

*Proof sketch.* Because $w$ is centered, $w^\top K_c(r) w = w^\top K(r) w$, and $w^\top K_c(r) w = \langle K_c(r), ww^\top \rangle_F$. By Cauchy–Schwarz inequality, $\langle K_c(r), ww^\top \rangle_F \leq \|K_c(r)\|_F \|ww^\top\|_F = \|K_c(r)\|_F \|w\|_2^2$. Dividing by $\|K_c(r)\|_F$ gives the global bound whenever $\|K_c(r)\|_F > 0$; the zero-norm case follows from the convention. For a nontrivial two-level threshold score $r^{(j)}$, $K(r^{(j)})$ is block-constant. Let $c_0 := \kappa(0)$, $c_1 := \kappa(|a-b|)$. The value $c_0$ is the within-block similarity, and $c_1$ is the across-block similarity. Since $a < b$ and $\kappa$ is strictly decreasing, we have $c_0 - c_1 > 0$. After centering, the constant all-ones component disappears and only the block contrast remains. Hence $K_c(r^{(j)}) = 2(c_0 - c_1) g_j g_j^\top$. Equivalently, write $K_c(r^{(j)}) = c_j g_j g_j^\top$, $c_j := 2(c_0 - c_1) > 0$. Therefore,

$$\bar{\Delta}_w(r^{(j)}) = \frac{c_j (w^\top g_j)^2}{c_j \|g_j g_j^\top\|_F} = \frac{(w^\top g_j)^2}{\|g_j\|_2^2}$$

$$= \|w\|_2^2 \left( \frac{w^\top g_j}{\|w\|_2 \|g_j\|_2} \right)^2 = \|w\|_2^2 \rho_j^2. \qquad \square$$

Assume $\sigma_1 > 0$. Applying Lemma 6.2 to the shared admission direction turns $v$ into an explicit monotone admission-level cutoff. The lemma requires a centered direction. Since the rows of $\widetilde{W}$ are centered, $\widetilde{W}\mathbf{1} = 0$. Thus, any right singular vector associated with a positive singular value is

orthogonal to $\mathbf{1}$, and therefore satisfies $Hv = v$. Hence we may take $w = v$. The rank-one shared direction gives the explicit cutoff

$$j^\star \in \arg\max_{1 \le j \le n-1} \left( \frac{v^\top g_j}{\|v\|_2 \|g_j\|_2} \right)^2.$$

**Approximate rank-one structure.** The previous reduction is exact for the projected matrix $\widetilde{W}_1$. In real data, outcomes also contain task-specific variation, so $\widetilde{W}$ is generally only approximately rank one. We therefore compare the original objective to the projected rank-one objective using an $\varepsilon_0$-floored denominator

$$J_{\varepsilon_0}(r) := \frac{\langle K_c(r), \widetilde{W}^\top \widetilde{W} \rangle_F}{\max\{\|K_c(r)\|_F, \varepsilon_0\}},$$

$$J_{1,\varepsilon_0}(r) := \frac{\langle K_c(r), \widetilde{W}_1^\top \widetilde{W}_1 \rangle_F}{\max\{\|K_c(r)\|_F, \varepsilon_0\}}.$$

**Theorem 6.3** (Uniform approximation and near-optimality transfer)**.** *Define the exact uniform gap* $\Delta_{\mathrm{gap}} := \sup_{r \in \mathcal{R}_B^\uparrow} |J_{\varepsilon_0}(r) - J_{1,\varepsilon_0}(r)|$. *Let* $\varepsilon_{\mathrm{gap}}$ *denote the explicit upper bound from Appendix B.4, so that* $\Delta_{\mathrm{gap}} \le \varepsilon_{\mathrm{gap}}$. *If* $r_1^\star \in \arg\max_{r \in \mathcal{R}_B^\uparrow} J_{1,\varepsilon_0}(r)$, *then* $J_{\varepsilon_0}(r_1^\star) \ge \sup_{r \in \mathcal{R}_B^\uparrow} J_{\varepsilon_0}(r) - 2\varepsilon_{\mathrm{gap}}$.

*Proof sketch.* By definition of $\Delta_{\mathrm{gap}}$, for all $r \in \mathcal{R}_B^\uparrow$, $J_{\varepsilon_0}(r) \ge J_{1,\varepsilon_0}(r) - \Delta_{\mathrm{gap}}$, $J_{1,\varepsilon_0}(r) \ge J_{\varepsilon_0}(r) - \Delta_{\mathrm{gap}}$. By optimality of $r_1^\star$ for $J_{1,\varepsilon_0}$, for every $r \in \mathcal{R}_B^\uparrow$, $J_{1,\varepsilon_0}(r_1^\star) \ge J_{1,\varepsilon_0}(r)$. Therefore, for every $r \in \mathcal{R}_B^\uparrow$,

$$J_{\varepsilon_0}(r_1^\star) \ge J_{1,\varepsilon_0}(r_1^\star) - \Delta_{\mathrm{gap}} \ge J_{1,\varepsilon_0}(r) - \Delta_{\mathrm{gap}}$$
$$\ge J_{\varepsilon_0}(r) - 2\Delta_{\mathrm{gap}}.$$

Taking the supremum over $r \in \mathcal{R}_B^\uparrow$ and using $\Delta_{\mathrm{gap}} \le \varepsilon_{\mathrm{gap}}$, we obtain $J_{\varepsilon_0}(r_1^\star) \ge \sup_{r \in \mathcal{R}_B^\uparrow} J_{\varepsilon_0}(r) - 2\varepsilon_{\mathrm{gap}}$. $\square$

Thus, when the explicit residual bound $\varepsilon_{\mathrm{gap}}$ is small, a maximizer of the floored rank-one projected objective $J_{1,\varepsilon_0}$ is near-optimal for the original floored multi-task objective $J_{\varepsilon_0}$.

## 7. Experiments

**Datasets.** We used two large, de-identified real-world EHR benchmark datasets: **MIMIC-IV** and **MIMIC-III**. MIMIC-IV (Johnson et al., 2023) contains 546,028 hospital admissions from 223,452 patients; we restrict to an ICD-10 cohort (admissions with at least one ICD-10 diagnosis), yielding 254,377 admissions from 122,905 patients in our final dataset. MIMIC-III (Johnson et al., 2016) contains 58,976 admissions from 46,520 patients and is ICU-heavy:

57,786 admissions include an ICU stay (ratio 0.980) compared to 0.156 in MIMIC-IV, which motivates a stricter late-transfer ICU label for MIMIC-III (Appendix A.9).

**Splits.** We use patient-disjoint train/validation/test splits (70%/10%/20%) by hashing `subject_id`. This yields, for MIMIC-IV ICD-10: 178,178/25,228/50,971 admissions (train/val/test) and 85,980/12,235/24,690 patients. For MIMIC-III: 41,200/5,867/11,909 admissions and 32,488/4,630/9,402 patients.

**Evaluation Metrics.** We evaluate **general dependence** between a *single* severity score $s_i = s_\theta(X_i)$ and each binary outcome, to capture nonlinear score–outcome relationships. We report **distance correlation (dCorr)** (Székely et al., 2007), which detects systematic linear or nonlinear dependence between $s_i$ and $y_i^{(t)}$, and **mutual information (MI)**, which quantifies how much the score reduces uncertainty about the outcome. Details are provided in Appendix A.10.

**Clinical Outcomes.** We consider four clinically relevant binary outcomes: (i) **in-hospital mortality (MORT)**, defined by a recorded in-hospital death time for the admission; (ii) **30-day mortality (30M)**, defined as all-cause death within 30 days of admission using patient-level date of death; (iii) **length of stay (LOS)**, defined as a binary outcome representing length of stay $> 7$ days when `ADMITTIME` and `DISCHTIME` are available; and (iv) **ICU transfer (ICU)**, defined from the first ICU `INTIME` relative to admission. For ICU transfer, MIMIC-IV uses any ICU entry after admission, whereas MIMIC-III uses first ICU entry $> 24$ hours after admission. Details are provided in Appendix A.9.

**Baselines.** We evaluate our model against several competitive single-index baselines: (i) **traditional clinical indices**, including CCI (Charlson et al., 1987), van Walraven–weighted ECI (Van Walraven et al., 2009), and a CCI+ECI aggregate model; (ii) **classical ML baselines**, including $k$NN (Choi et al., 2022), Naive Bayes (Wolfson et al., 2015), logistic regression (Cowling et al., 2021), factorization machines (Yin et al., 2025), and gradient-boosted trees (Li et al., 2025); and (iii) **deep learning baselines**, including neural set/sequence models such as bag-of-codes MLPs (Yin et al., 2025), attention-based MIL pooling (Harerimana et al., 2021), and DeepSets (Liu et al., 2020). Full details are provided in Appendices A.7 and A.8.

**Experimental protocol and uncertainty.** Deep models use the same tokenization, patient-disjoint splits, batch size, epochs, and validation-based checkpoint selection. For neural models, we report mean±SD over seeds $\{11, 101, 1001\}$; seeds were fixed, but CUDA/cuDNN was not forced to be bitwise deterministic, so exact reruns may show small variation. Classical baselines are deterministic or use fixed randomness. Metrics use valid-label admissions only; details are provided in Appendix A.10.

*Table 1.* **Distance Correlation between each model's unified risk score and clinical outcomes.** Higher is better. Bolded values highlight the best performer per outcome. *For readability, each column is scaled by the power of 10 shown in the header; divide by that factor to recover the original dCorr.*

| Model | MIMIC-IV | | | | MIMIC-III | | | |
|---|---|---|---|---|---|---|---|---|
| | **MORT** $(\times 10^2)$ | **30M** $(\times 10^2)$ | **LOS** $(\times 10^2)$ | **ICU** $(\times 10^2)$ | **MORT** $(\times 10^2)$ | **30M** $(\times 10^2)$ | **LOS** $(\times 10^2)$ | **ICU** $(\times 10^2)$ |
| *Traditional Clinical Indices* | | | | | | | | |
| Charlson (CCI) | 12.59 | 18.44 | 24.55 | 16.09 | 15.54 | 20.26 | 15.28 | 13.07 |
| Elixhauser (ECI) | 19.98 | 23.87 | 33.15 | 25.98 | 21.38 | 24.70 | 23.10 | 13.38 |
| CCI+ECI (scalar) | 17.46 | 22.67 | 30.89 | 22.44 | 19.79 | 24.06 | 20.82 | 14.15 |
| *Classical Machine Learning Baselines* | | | | | | | | |
| kNN | 22.40 | 23.98 | 30.60 | 34.79 | 22.56 | 23.22 | 23.55 | 11.15 |
| Multinomial Naive Bayes | 32.09 | 32.02 | 35.78 | 45.69 | 24.89 | 25.21 | 25.04 | 16.73 |
| Complement Naive Bayes | 32.08 | 32.05 | 35.85 | 45.58 | 24.69 | 25.02 | 23.89 | 16.76 |
| FusedLogits LR | 34.99 | 34.92 | 51.02 | 57.23 | 31.19 | 31.20 | 46.05 | 18.96 |
| Factorization Machine (FM score) | 36.41 | 35.82 | 50.80 | 56.77 | 31.96 | 31.28 | 45.38 | 20.51 |
| Gradient Boosted Trees (Score/Logit) | 34.54 | 34.58 | 51.00 | 55.61 | 32.28 | 31.88 | 48.16 | 20.17 |
| *Deep Learning Baselines* | | | | | | | | |
| Deep and Cross Network (DCN) | $28.44 \pm 0.44$ | $28.73 \pm 0.57$ | $48.11 \pm 0.41$ | $51.97 \pm 0.43$ | $30.06 \pm 0.62$ | $29.24 \pm 0.48$ | $47.12 \pm 0.23$ | $19.70 \pm 0.81$ |
| Deep MLP Bag-of-Codes (EmbeddingBag) | $28.88 \pm 0.21$ | $29.13 \pm 0.18$ | $48.22 \pm 0.29$ | $52.04 \pm 0.18$ | $30.17 \pm 0.98$ | $29.40 \pm 0.82$ | $47.67 \pm 0.72$ | $20.18 \pm 0.44$ |
| Star-GAT | $26.97 \pm 1.27$ | $27.48 \pm 1.23$ | $47.19 \pm 0.99$ | $50.24 \pm 0.68$ | $29.44 \pm 0.97$ | $29.01 \pm 0.91$ | $48.25 \pm 0.73$ | $20.06 \pm 0.76$ |
| Pure MIL Attention Pooling | $29.25 \pm 0.62$ | $29.53 \pm 0.43$ | $48.37 \pm 0.36$ | $52.20 \pm 0.63$ | $31.11 \pm 0.70$ | $30.40 \pm 0.31$ | $48.11 \pm 0.43$ | $20.41 \pm 0.53$ |
| Set Transformer (single-logit) | $28.50 \pm 0.68$ | $29.02 \pm 0.63$ | $49.12 \pm 0.37$ | $52.04 \pm 0.19$ | $30.29 \pm 0.62$ | $29.92 \pm 0.74$ | $48.08 \pm 0.43$ | $20.51 \pm 1.10$ |
| DeepSets (mean $\oplus$ max pool) | $28.51 \pm 0.33$ | $28.84 \pm 0.27$ | $48.60 \pm 0.60$ | $51.92 \pm 0.41$ | $29.29 \pm 0.49$ | $29.11 \pm 0.32$ | $48.88 \pm 0.43$ | $\mathbf{21.52 \pm 0.26}$ |
| **Our Model** | $\mathbf{54.80 \pm 1.40}$ | $\mathbf{49.42 \pm 1.34}$ | $\mathbf{51.15 \pm 0.88}$ | $\mathbf{61.97 \pm 0.64}$ | $\mathbf{39.06 \pm 3.27}$ | $\mathbf{37.39 \pm 2.63}$ | $\mathbf{49.84 \pm 1.38}$ | $18.55 \pm 0.19$ |

*Table 2.* **Mutual Information between each model's unified risk score and clinical outcomes.** Higher is better. Bolded values highlight the best performer per outcome. *For readability, each column is scaled by the power of 10 shown in the header; divide by that factor to recover the original MI.*

| Model | MIMIC-IV | | | | MIMIC-III | | | |
|---|---|---|---|---|---|---|---|---|
| | **MORT** $(\times 10^3)$ | **30M** $(\times 10^3)$ | **LOS** $(\times 10^2)$ | **ICU** $(\times 10^2)$ | **MORT** $(\times 10^3)$ | **30M** $(\times 10^3)$ | **LOS** $(\times 10^2)$ | **ICU** $(\times 10^3)$ |
| *Traditional Clinical Indices* | | | | | | | | |
| Charlson (CCI) | 9.64 | 18.99 | 3.41 | 1.99 | 16.35 | 26.32 | 0.94 | 14.60 |
| Elixhauser (ECI) | 20.17 | 28.73 | 6.26 | 3.85 | 26.51 | 30.97 | 3.14 | 14.72 |
| CCI+ECI (scalar) | 19.94 | 28.66 | 6.39 | 4.86 | 30.99 | 36.11 | 3.40 | 16.52 |
| *Classical Machine Learning Baselines* | | | | | | | | |
| kNN | 57.70 | 62.71 | 8.43 | 13.06 | 68.28 | 68.07 | 8.55 | 20.65 |
| Multinomial Naive Bayes | 56.44 | 57.74 | 7.90 | 13.37 | 57.28 | 54.45 | 7.27 | 22.29 |
| Complement Naive Bayes | 56.68 | 56.99 | 7.75 | 13.26 | 57.75 | 55.43 | 7.12 | 27.53 |
| FusedLogits LR | 53.55 | 54.33 | 13.75 | 17.60 | 56.87 | 53.66 | 13.25 | 28.19 |
| Factorization Machine (FM score) | 54.50 | 54.27 | 13.88 | 16.99 | 62.97 | 57.98 | 13.62 | 27.58 |
| Gradient Boosted Trees (Score/Logit) | 64.30 | 63.30 | 14.86 | 17.67 | 65.24 | 66.81 | 14.76 | 24.55 |
| *Deep Learning Baselines* | | | | | | | | |
| Deep and Cross Network (DCN) | $65.33 \pm 0.62$ | $61.70 \pm 0.24$ | $14.48 \pm 0.15$ | $18.33 \pm 0.33$ | $65.46 \pm 2.76$ | $59.57 \pm 1.60$ | $13.47 \pm 0.48$ | $27.20 \pm 0.64$ |
| Deep MLP Bag-of-Codes (EmbeddingBag) | $65.88 \pm 0.54$ | $63.63 \pm 1.19$ | $14.73 \pm 0.12$ | $18.45 \pm 0.06$ | $66.20 \pm 4.99$ | $61.39 \pm 3.21$ | $13.83 \pm 0.23$ | $29.89 \pm 2.13$ |
| Star-GAT | $67.38 \pm 2.48$ | $65.32 \pm 2.87$ | $14.89 \pm 0.27$ | $18.89 \pm 0.47$ | $70.31 \pm 2.27$ | $63.74 \pm 2.44$ | $14.10 \pm 0.54$ | $29.28 \pm 2.68$ |
| Pure MIL Attention Pooling | $67.89 \pm 0.94$ | $65.16 \pm 2.32$ | $14.93 \pm 0.04$ | $18.71 \pm 0.12$ | $71.72 \pm 1.57$ | $64.64 \pm 0.79$ | $14.32 \pm 0.37$ | $26.14 \pm 2.26$ |
| Set Transformer (single-logit) | $68.35 \pm 2.40$ | $65.84 \pm 3.19$ | $15.16 \pm 0.18$ | $18.59 \pm 0.25$ | $75.15 \pm 1.62$ | $69.23 \pm 1.95$ | $14.54 \pm 0.31$ | $29.64 \pm 3.14$ |
| DeepSets (mean $\oplus$ max pool) | $69.92 \pm 1.26$ | $67.22 \pm 1.60$ | $15.36 \pm 0.11$ | $\mathbf{19.24 \pm 0.19}$ | $73.69 \pm 3.29$ | $68.08 \pm 1.88$ | $15.17 \pm 0.30$ | $\mathbf{32.42 \pm 4.22}$ |
| **Our Model** | $\mathbf{74.22 \pm 0.24}$ | $\mathbf{72.19 \pm 0.93}$ | $\mathbf{15.42 \pm 0.18}$ | $19.01 \pm 0.29$ | $\mathbf{84.52 \pm 10.89}$ | $\mathbf{77.77 \pm 7.53}$ | $\mathbf{16.45 \pm 0.92}$ | $26.42 \pm 3.77$ |

**Main results.** Across both datasets and dependence measures, MLCI is substantially more informative for mortality, while remaining competitive on other outcomes. On both MIMIC-IV/III, our method achieves the **strongest dependence** for both in-hospital mortality and 30-day mortality. For LOS, our model is **still best or essentially tied** in both datasets (MIMIC-IV dCorr $= 51.15 \pm 0.88$; MIMIC-III dCorr $= 49.84 \pm 1.38$; MI shows the same pattern). For ICU transfer, our method is **best** on MIMIC-IV (dCorr $= 61.97 \pm 0.64$), while on MIMIC-III the strongest ICU-transfer dependence is achieved by a DeepSets BCE baseline. This is consistent with MIMIC-III being ICU-heavy and our ICU label capturing *late* transfers influenced by workflow/triage beyond a shared severity axis.

## 8. Theory Diagnostics

The theory motivates two empirical diagnostics for when a *single monotone severity score* captures shared structure across binary endpoints. The first is the rank-one energy ratio $\sigma_1^2 / \|\widetilde{W}\|_F^2$, which measures how much of the stacked centered label structure is explained by one shared admission direction. The second is the threshold scan $j^\star \in \arg\max_j (v^\top g_j)^2 / (\|v\|_2^2 \|g_j\|_2^2)$, which tests whether this direction identifies a compact high-severity upper-tail group. We compute these diagnostics on MIMIC-IV and MIMIC-III using the four clinical outcomes, restricted to admissions with all labels present. We use $n = 5000$ uniformly sampled admissions for MIMIC-IV and the full intersection-valid MIMIC-III cohort ($n = 11909$), ordered by the learned scores $s_i = s_\theta(X_i)$. Reported diagnostics

are from the original experimental runs.

**Diagnostics computed.** Let $\ell^{(t)} := Hy^{(t)}$ denote the centered label vector for task $t$. For task weights $\alpha_t > 0$, form the stacked normalized label-profile matrix $\widetilde{W} \in \mathbb{R}^{T \times n}$ with rows $\widetilde{W}_{t,:} = \widetilde{w}^{(t)\top} = \frac{\sqrt{\alpha_t}}{\|\ell^{(t)}\|_2} \ell^{(t)\top}$. Equivalently, $\widetilde{W}^\top \widetilde{W} = \sum_{t=1}^{T} \frac{\alpha_t}{\|\ell^{(t)}\|_2^2} \ell^{(t)} \ell^{(t)\top}$. For the diagnostic results reported below, we set $\alpha_t \equiv 1$ for all tasks. Compute the SVD $\widetilde{W} = U\Sigma V^\top$ and define the shared admission direction $v := V_{:,1}$. We report: (i) the rank-one energy ratio $\sigma_1^2/\|\widetilde{W}\|_F^2$, (ii) the full multi-task objective $J(s)$ evaluated at the learned score vector, and (iii) the rank-one projected objective $J_1(s) = \sigma_1^2 \bar{\Delta}_v(s)$.

**Diagnostic 1: Rank-one alignment across tasks.** Table 3 shows that $\widetilde{W}$ is *moderately* rank-one aligned in both cohorts: $\sigma_1^2/\|\widetilde{W}\|_F^2 = 0.49$ (MIMIC-IV) and $0.45$ (MIMIC-III). The corresponding rank-one surrogate captures a substantial portion of the learned-score objective (MIMIC-IV: $J_1/J \approx 0.64$; MIMIC-III: $J_1/J \approx 0.34$), indicating a meaningful shared cross-task component while allowing for additional task-specific structure.

*Table 3.* Rank-one alignment and objective values on learned scores. $J(s)$ is the full multi-task nHSIC objective evaluated on the learned score vector; $J_1(s)$ is the rank-one projected objective.

| Dataset | $\sigma_{1:4}$ of $\widetilde{W}$ | $\sigma_1^2/\|\widetilde{W}\|_F^2$ | $J(s)$ | $J_1(s)$ |
|---|---|---|---|---|
| MIMIC-IV | (1.40, 1.05, 0.81, 0.53) | 0.49 | 0.622 | 0.397 |
| MIMIC-III | (1.34, 1.14, 0.84, 0.47) | 0.45 | 0.453 | 0.152 |

**Diagnostic 2: Shared direction predicts the optimal monotone threshold split.** Lemma 6.2, applied to the shared direction $v$, implies that the optimal split for the rank-one projected objective within two-level monotone threshold scores can be found by scanning $\rho_j^2(v) = \frac{(v^\top g_j)^2}{\|v\|_2^2 \|g_j\|_2^2}$, where $g_j := H\mathbf{1}_{\{j+1,\dots,n\}}$. Here $\rho_j(v)$ is the cosine similarity between $v$ and the centered step vector $g_j$, corresponding to a two-level threshold split after index $j$. Since both vectors are centered, this is equivalently their Pearson correlation.

*Table 4.* Two-level monotone threshold diagnostics. $j_J$ is the best split for the full multi-task threshold objective within the threshold family; $j_v$ maximizes $\rho_j^2(v)$, equivalently selecting the best threshold for the rank-one projected objective.

| Data | $j_J$ | $j_v$ | Tail size | $\rho_\star^2(v)$ |
|---|---|---|---|---|
| MIMIC-IV | 4862 | 4862 | 137/5000 (2.74%) | 0.554 |
| MIMIC-III | 11265 | 11265 | 643/11909 (5.40%) | 0.220 |

Table 4 compares the $v$-selected split with the best split for the full multi-task threshold objective. In the original diagnostic analysis reported here, the two coincide in both datasets. This supports the rank-one diagnostic interpretation: $v$ identifies a compact high-severity upper-tail

subgroup. In general, deviations between $j_v$ and $j_J$ would quantify residual task-specific heterogeneity beyond the leading rank-one component.

**Task-wise heterogeneity.** Table 5 shows that different endpoints relate to the learned score in different ways. The per-task nHSIC values measure continuous dependence with the learned score: in MIMIC-III, this dependence is strongest for length of stay, whereas in MIMIC-IV it is strongest for ICU transfer and length of stay. The threshold certificates show a complementary pattern: mortality endpoints have strong upper-tail threshold structure, especially in MIMIC-IV. Thus, some outcomes vary smoothly along the severity score, while others are mainly captured by a high-risk cutoff, helping explain why a single monotone threshold can still support multiple endpoints.

*Table 5.* Task-wise diagnostics: (top) nHSIC of learned scores and (bottom) each task's best threshold certificate from Lemma 6.2.

| Dataset | Metric | MORT | 30M | LOS | ICU |
|---|---|---|---|---|---|
| MIMIC-IV | nHSIC on learned score | 0.063 | 0.068 | 0.212 | 0.278 |
| | best threshold $\rho_\star^2$ | 0.593 | 0.457 | 0.221 | 0.372 |
| MIMIC-III | nHSIC on learned score | 0.079 | 0.070 | 0.271 | 0.033 |
| | best threshold $\rho_\star^2$ | 0.212 | 0.180 | 0.238 | 0.029 |

## 9. Limitations

This study has several limitations. First, MLCI assumes that different clinical outcomes share a common severity signal; this may be weaker when clinical outcomes are dominated by task-specific variation. Second, MLCI uses only diagnosis codes, which may be incomplete or noisy and may miss severity information contained in notes, labs, or vital signs. Thus, it should not be interpreted as a complete physiological severity measure. Third, some outcomes reflect care processes as well as patient severity; for example, ICU transfer can depend on bed availability, triage practice, and hospital workflow in addition to disease burden.

## 10. Conclusion

Across two EHR benchmarks, dependence results show that MLCI learns a severity score with strong general dependence on multiple clinical outcomes, while theory diagnostics indicate that centered admission-level outcome profiles across these outcomes exhibit moderate rank-one structure and that the induced shared admission direction $v$ yields a monotone threshold rule identifying compact high-severity subgroups. More broadly, MLCI shows how data-driven comorbidity scoring can move beyond fixed linear indices by learning nonlinear severity structure while preserving the practical simplicity of a one-dimensional clinical score.

## Acknowledgements

This work was supported by NSF CAREER Award IIS-2442159 (B.A. and S.B.) and NSF SCH-2500344 (K.J.).

## Impact Statement

In this study, we introduce MLCI, a machine-learned comorbidity index for more flexible and outcome-aware severity scoring in electronic health records. Unlike traditional indices such as Charlson and Elixhauser, which rely on fixed mortality-centric weights, MLCI learns a single scalar severity score from diagnosis codes by maximizing normalized Hilbert–Schmidt Independence Criterion across multiple clinical endpoints. This framework supports data-driven severity summarization, multi-outcome risk stratification, and retrospective risk adjustment while preserving the usability of a one-dimensional clinical score. Future work should extend MLCI to richer EHR modalities and prospectively validate it before clinical deployment.

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

# Appendix

This appendix accompanies "A MACHINE-LEARNED COMORBIDITY INDEX" and is organized as follows:

- **Appendix A: Implementation Details.** Data preprocessing, model architecture, training procedure, and hyperparameters.

- **Appendix B: Multi-task nHSIC Theory.** Theoretical results underpinning the multi-task dependence objective.

- **Appendix C: Additional Experiments.** Supplementary dependence evaluations and ablations.

- **Appendix D: Comorbidity Indices.** Notation and scoring formulas for the Charlson and van Walraven Elixhauser comorbidity indices.

- **Appendix E: Post-hoc Risk Curves.** Estimation and visualization of outcome-specific risk curves as a function of the learned score.

## A. Implementation Details

### A.1. Data processing and diagnosis tokenization

**ICD normalization and prefixing.** Each diagnosis code is converted to uppercase, punctuation and whitespace are removed, and the first $k = 4$ characters are used as the diagnosis token. A vocabulary is built *from the training split only* and augmented with <PAD> (index 0) and <UNK> (index 1). Each admission $i$ is represented as a variable-length token collection $X_i = \{C_{i,1}, \ldots, C_{i,m_i}\}$, where $m_i$ is the number of retained diagnosis tokens after truncation and $D_{\max} = 256$ is the maximum retained length.

**Train/val/test mapping.** We build the vocabulary from the training diagnoses file, then map diagnoses into token collections separately for train/val/test using the shared training vocabulary.

### A.2. Outcomes, missingness masks, and validity

**Task set.** We consider $T$ binary outcomes, such as in-hospital mortality, 30-day mortality, length of stay, and ICU transfer, with possible missing labels. For task $t \in [T]$, the label for admission $i$ is denoted $y_i^{(t)} \in \{0, 1\}$.

**Mask definition.** Let $M_i^{(t)} \in \{0, 1\}$ indicate whether $y_i^{(t)}$ is defined. For most outcomes, $M_i^{(t)} = 1$ if and only if the label value is non-missing. For long_stay, validity is defined by a dedicated indicator column long_stay_defined; we set

$$M_i^{(\text{los})} = \texttt{long\_stay\_defined}$$

and treat long_stay labels outside this mask as invalid. In storage, missing labels are filled with 0, but are excluded from objectives and metrics using the mask.

**Intersection validity for MLCI.** For MLCI's multi-task nHSIC training, validation, and model selection, we enforce intersection validity

$$M_i^{\cap} = \prod_{t=1}^{T} M_i^{(t)}, \tag{5}$$

and compute every per-task nHSIC contribution only on rows with $M_i^{\cap} = 1$. This ensures that all task contributions in the multi-task nHSIC objective are evaluated on the same admission set. Decomposable BCE baselines instead use task-specific validity masks, as described below.

### A.3. DeepSets architecture for a single scalar score

**Embedding and token MLP.** We use an embedding dimension $d = 128$. Each token embedding is transformed by an elementwise MLP

$$\phi : \text{Linear}(d, d) \rightarrow \text{ReLU} \rightarrow \text{Linear}(d, d) \rightarrow \text{ReLU}.$$

**Pooling.** Given masked token features $\{h_j\}$, we compute masked mean pooling and masked max pooling, then concatenate and apply LayerNorm:

$$p = \mathrm{LN}([\bar{h}; \hat{h}]) \in \mathbb{R}^{2d}.$$

**Score head.** We map $p$ to a scalar via

$$\rho: \ \mathrm{Linear}(2d, d) \rightarrow \mathrm{ReLU} \rightarrow \mathrm{Linear}(d, d) \rightarrow \mathrm{ReLU} \rightarrow \mathrm{Linear}(d, 1),$$

producing the learned score $s_i = s_\theta(X_i) \in \mathbb{R}$.

### A.4. nHSIC computation

**Kernels.** On a mini-batch of scores $\{s_i\}_{i=1}^{n_b}$, we use an RBF score kernel

$$K_{ij}^{(b)} = \exp\left(-\frac{(s_i - s_j)^2}{2\sigma^2}\right).$$

For task $t$, the binary label Gram matrix is

$$L_{ij}^{(b,t)} = \mathbb{I}\{y_i^{(t)} = y_j^{(t)}\}.$$

**Centering.** Let

$$H_b := I - \frac{1}{n_b}\mathbf{1}\mathbf{1}^\top$$

be the mini-batch centering matrix. The centered mini-batch Gram matrices are

$$K_c^{(b)} := H_b K^{(b)} H_b, \qquad L_{t,c}^{(b)} := H_b L^{(b,t)} H_b.$$

Equivalently, this subtracts row and column means and adds the grand mean.

**Normalized criterion with floor.** For task $t$, we compute

$$\widehat{\mathrm{nHSIC}}\left(s, y^{(t)}\right) = \frac{\langle K_c^{(b)}, L_{t,c}^{(b)} \rangle_F}{\max\{\|K_c^{(b)}\|_F, \varepsilon_0\}\|L_{t,c}^{(b)}\|_F}.$$

We set the nHSIC floor to

$$\varepsilon_0 = \texttt{NHSIC\_FLOOR} = 10^{-4}$$

during training.

**Per-task masking and guards.** For MLCI, each per-task nHSIC term is computed on the intersection-valid mini-batch indices $\{i : M_i^\cap = 1\}$. If fewer than 3 valid points are available in a batch, or if the valid labels for task $t$ are single-class, we set $\widehat{\mathrm{nHSIC}}(s, y^{(t)}) = 0$ for that batch.

### A.5. Bandwidth selection

We set the RBF bandwidth $\sigma$ once per run using a stabilized median heuristic on training scores. Concretely, we run the current model on a fixed-size training subsample, standardize the sampled scores to zero mean and unit variance, compute pairwise absolute differences on a deterministic subsample, and set $\sigma$ to the median of the nonzero differences. We apply a lower bound of $0.05$ to avoid degenerate micro-kernel behavior when scores are nearly tied.

This standardization is used only for selecting the bandwidth. The learned score itself is not standardized in the nHSIC objective: training uses the scalar model outputs directly in

$$K_{ij}^{(b)} = \exp\left(-\frac{(s_i - s_j)^2}{2\sigma^2}\right).$$

### A.6. Optimization and training protocol

**Optimizer and schedule.** We use AdamW with learning rate $10^{-3}$, batch size 256, and no weight decay. Training uses two stages: Stage 1 trains single-task models for 10 epochs each to estimate task weights; Stage 2 trains a multi-task model for 10 epochs using the weighted objective.

**Two-stage weights.** Let $\widehat{h}_t$ be the best validation nHSIC for task $t$ in Stage 1, and let

$$\widehat{h}_{\max} := \max_t \widehat{h}_t.$$

We compute stabilized inverse-strength weights

$$\alpha_t \propto \left( \frac{\widehat{h}_{\max}}{\max(\widehat{h}_t, \varepsilon_{\mathrm{wt}})} \right)^{\gamma_{\mathrm{wt}}},$$

with

$$\varepsilon_{\mathrm{wt}} = 0.02, \qquad \gamma_{\mathrm{wt}} = 0.25.$$

We clip weights to a bounded range with factor 3.0 and renormalize so the mean weight over active tasks is 1. The assumption $\alpha_t > 0$ is intended for non-degenerate tasks included in the objective. If a task has no valid labels or only one observed class in the training split, it is not included in the empirical objective; setting its code weight to zero is only the implementation equivalent of omitting that task from the sum.

**Model selection.** We select checkpoints by validation objective: Stage 1 selects the checkpoint maximizing single-task validation nHSIC; Stage 2 selects the checkpoint maximizing the weighted validation sum.

**Random seeds and deterministic execution.** For neural models, we fixed Python, NumPy, and PyTorch random seeds for each run and report mean±SD over seeds $\{11, 101, 1001\}$. CUDA/cuDNN execution was not forced to be bitwise deterministic, so exact reruns may exhibit small numerical variation. Reported neural results correspond to the runs with the specified seeds.

**Score orientation.** After Stage 2 training, we orient the score using validation mortality. If the Pearson correlation between $s_i$ and $y_i^{(\mathrm{mort})}$ on valid validation rows is negative, we multiply all scores by $-1$.

### A.7. Deep Learning Baselines

All deep learning baseline models use the *same* data processing, vocabulary construction, padding rules, train/validation/test splits, optimizer settings, and evaluation code described above. They use task-specific validity masks for BCE training and validation, as detailed below. The only component that changes across baselines is the architecture, which maps an unordered multiset of diagnosis tokens for admission $i$ to a single scalar baseline score $s_i^{\mathrm{base}} = f_\theta(X_i)$.

**Single-index multi-outcome setting.** Each admission produces one shared scalar baseline score $s_i^{\mathrm{base}} \in \mathbb{R}$. This scalar is reused across all outcomes and is not an outcome-specific calibrated probability. Missing labels are handled by the task mask $M_i^{(t)}$ (Section A.2); padded tokens never contribute to the pooled representation.

**Two-stage training with weighted masked BCE.** All BCE baselines use a two-stage protocol. In Stage 1, we train separate single-task models, one per outcome, and select the best checkpoint by *minimum* masked validation BCE. Stage 1 validation losses are converted into task weights that emphasize tasks that are harder under the shared score constraint. In Stage 2, a new shared model is trained from scratch to minimize a normalized weighted masked BCE across outcomes:

$$\mathcal{L}_{\mathrm{BCE}} = \frac{\sum_{t=1}^{T} \alpha_t^{\mathrm{BCE}} \sum_i M_i^{(t)} \, \mathrm{BCEWithLogits}(s_i^{\mathrm{base}}, y_i^{(t)})}{\sum_{t=1}^{T} \alpha_t^{\mathrm{BCE}} \sum_i M_i^{(t)}}.$$

**Weight computation and skipped tasks.** Let $\widehat{\mathcal{L}}_t$ denote the best Stage 1 validation BCE for outcome $t$. We compute a nonnegative score

$$\mathrm{score}_t = \max(\log 2 - \widehat{\mathcal{L}}_t, 0)$$

and set baseline task weights using a capped inverse power law:

$$\alpha_t^{\mathrm{BCE}} = \mathrm{clip}\left(\left(\frac{\max_u \mathrm{score}_u}{\max(\mathrm{score}_t, \varepsilon_{\mathrm{wt}})}\right)^{\gamma_{\mathrm{wt}}}, \frac{1}{c}, c\right),$$

with

$$\varepsilon_{\mathrm{wt}} = 0.02, \qquad \gamma_{\mathrm{wt}} = 0.25, \qquad c = 3.0.$$

Weights are then mean-normalized over active tasks. Outcomes that are single-class, or have no valid labels in the training split, are skipped and assigned $\alpha_t^{\mathrm{BCE}} = 0$, so they do not contribute to Stage 2 training.

### A.8. Classical ML baselines

We implement several classical baselines on top of the same admission-level ICD-prefix representation and the same masking conventions as in Section A.2. For all classical models, diagnosis codes are normalized by converting to uppercase and removing punctuation and whitespace, then truncated to the first $k = 4$ characters to form ICD-prefix tokens. A prefix vocabulary is constructed *from the training split only* and augmented with a dedicated `<UNK>` token for out-of-vocabulary prefixes. Each admission is represented by at most $D_{\mathrm{max}} = 256$ retained diagnosis tokens in encounter order. Unless otherwise noted, classical baselines use the resulting sparse bag-of-words features; when TF–IDF is applied, its statistics are fit on TRAIN and then applied unchanged to VAL/TEST.

**Label validity and masking.** For each task $t$, we extract binary labels $y_i^{(t)}$ and a validity mask $M_i^{(t)}$, where $M_i^{(t)} = 1$ denotes that the label is defined for admission $i$. For most tasks, $M_i^{(t)} = 1$ if and only if the label entry is non-missing; for `long_stay`, validity is given by the dedicated indicator `long_stay_defined`. All training, validation objectives, and evaluation metrics for task $t$ use only rows with $M_i^{(t)} = 1$.

**DL-style validation loss.** To match the deep-learning evaluation convention used elsewhere, we compute validation binary cross-entropy as a mean over fixed-size batches. For each batch, we compute BCE on the subset of rows with $M_i^{(t)} = 1$; if a batch contains zero valid labels, its contribution is defined as $0$, and we then average over batches.

**Two-stage task weighting.** For multi-task fusion baselines, we derive per-task weights from Stage 1 validation BCE. Let $\widehat{\mathcal{L}}_t$ denote the Stage 1 validation BCE for task $t$, computed with masking and batch-averaging as above. We convert $\widehat{\mathcal{L}}_t$ to a nonnegative strength score

$$\mathrm{score}_t = \max(\log 2 - \widehat{\mathcal{L}}_t, 0),$$

and then set

$$\alpha_t^{\mathrm{BCE}} = \mathrm{clip}\left(\left(\frac{\max_u \mathrm{score}_u}{\max(\mathrm{score}_t, \varepsilon_{\mathrm{wt}})}\right)^{\gamma_{\mathrm{wt}}}, \frac{1}{c}, c\right).$$

Tasks that are unusable on TRAIN, because they have too few valid labels or are single-class on valid TRAIN rows, or have non-finite validation loss, receive $\alpha_t^{\mathrm{BCE}} = 0$. Active weights are normalized to have mean 1 over $\{t : \alpha_t^{\mathrm{BCE}} > 0\}$.

**Stage-1 per-task models.** We train per-task models for logistic regression (LR), factorization machines (FM), gradient-boosted trees (GBT), $k$-nearest neighbors ($k$NN), and Naive Bayes (NB). Each Stage 1 model is trained on TRAIN rows with $M_i^{(t)} = 1$ for that task and produces an admission-level score, logit, or margin $a_i^{(t),\mathrm{base}}$ for all admissions in TRAIN/VAL/TEST. When a model outputs probabilities, such as $k$NN, we convert to logits by

$$a_i^{(t),\mathrm{base}} = \log\left(\frac{\hat{p}_i^{(t)}}{1 - \hat{p}_i^{(t)}}\right),$$

with clipping for numerical stability.

**Stage-2 fusion into a single scalar score.** To obtain a single scalar score per admission for downstream dependence evaluation, we use one of two fusion strategies depending on the baseline. First, *pooled learner fusion* pools examples across tasks by repeating admissions for each valid task label and fits a single Stage 2 model with per-example sample weight $\alpha_t^{\text{BCE}}$. This is used for FM/GBT and for LR fusion over Stage 1 logits. Second, *weighted logit averaging* defines the final score as a weighted mean of Stage 1 logits across tasks,

$$s_i^{\text{base}} = \sum_t \widetilde{\alpha}_t^{\text{BCE}} \, a_i^{(t),\text{base}}, \qquad \widetilde{\alpha}_t^{\text{BCE}} \propto \alpha_t^{\text{BCE}}$$

over active tasks. This is used for lightweight fusion variants such as $k$NN and NB when no Stage 2 learner is trained.

## A.9. Outcomes and dataset construction

**Label construction overview.** We construct four admission-level binary outcomes from MIMIC-III and MIMIC-IV: in-hospital mortality, 30-day all-cause mortality, length of stay, and ICU transfer. Labels are derived from structured hospital tables, such as `ADMISSIONS`, `PATIENTS`, and `ICUSTAYS`, using timestamp-based rules relative to each admission's `ADMITTIME`. Unless otherwise noted, labels are fully observed for all included admissions; when required timestamps are missing, we treat the label as missing and exclude the admission for that task.

**In-hospital mortality.** We define an admission-level in-hospital mortality label using the hospital `ADMISSIONS` table. For each admission $i$, we set

$$y_i^{(\text{mort})} = 1$$

if and only if `DEATHTIME` is non-null for that admission, and set $y_i^{(\text{mort})} = 0$ otherwise. For MIMIC-IV, we restrict to an ICD-10 cohort by keeping admissions with at least one ICD-10 diagnosis record (`DIAGNOSES_ICD.icd_version=10`). The label is fully observed for all included admissions, so $M_i^{(\text{mort})} = 1$. We additionally compute age at admission, capped at 90 for ages $> 89$, using dataset-specific conventions.

**30-day all-cause mortality.** For each admission $i$, we define a 30-day all-cause mortality label relative to the admission start time. We use patient-level date-of-death, `PATIENTS.DOD`, as the all-cause death timestamp and compute the elapsed time in days from the admission time `ADMITTIME`. For admissions with valid `ADMITTIME`, we set $y_i^{(\text{mort30})} = 1$ if and only if `DOD` is observed and

$$0 \leq (\text{DOD} - \text{ADMITTIME}) \leq 30$$

days, and set $y_i^{(\text{mort30})} = 0$ otherwise. Thus the label includes both in-hospital deaths and post-discharge deaths within 30 days of admission. Admissions with missing `ADMITTIME` are excluded from this label and treated as missing. For MIMIC-IV, we restrict to an ICD-10 cohort by keeping admissions with at least one ICD-10 diagnosis record (`DIAGNOSES_ICD.icd_version=10`).

**Length of stay.** We define an admission-level length-of-stay label using stay duration computed from `ADMISSIONS.ADMITTIME` and `ADMISSIONS.DISCHTIME`. For each admission $i$ with valid timestamps, we compute

$$\text{los\_days} = (\text{DISCHTIME} - \text{ADMITTIME})$$

in days and set $y_i^{(\text{los})} = 1$ if and only if $\text{los\_days} > 7$, and set $y_i^{(\text{los})} = 0$ otherwise. Admissions with missing admission or discharge times are excluded from this label. For MIMIC-IV, we restrict to an ICD-10 cohort by keeping admissions with at least one ICD-10 diagnosis record (`DIAGNOSES_ICD.icd_version=10`).

**ICU transfer.** We derive ICU-transfer labels by linking hospital admissions, `hadm_id`, to ICU stays and extracting the first ICU entry time. For each admission $i$, we set `first_icu_intime` to the earliest valid `ICUSTAYS.INTIME` associated with that `hadm_id` and compute `time_to_icu_hours` as the elapsed hours from `ADMISSIONS.ADMITTIME` to `first_icu_intime`. We define `icu_any=1` if and only if an ICU stay exists for the admission. Admissions with missing `ADMITTIME` are excluded from this label. Because MIMIC-III is ICU-heavy, meaning nearly all admissions have an ICU stay, we define `icu_transfer` as a late ICU transfer indicator:

$$y_i^{(\text{icu})} = 1 \quad \text{if and only if} \quad \text{time\_to\_icu\_hours} > 24.$$

In MIMIC-IV, where ICU admission is less ubiquitous, we use an any-ICU-after-admission definition:

$$y_i^{(\text{icu})} = 1 \quad \text{if and only if} \quad \texttt{time\_to\_icu\_hours} > 0.$$

We restrict the MIMIC-IV cohort to ICD-10 admissions for consistency with the rest of our pipeline.

**Dataset construction and splits.** For each dataset, MIMIC-III and MIMIC-IV, we construct an admission-level cohort anchored on the set of unique (`subject_id`, `hadm_id`) pairs appearing in our in-hospital mortality label file, and merge all other outcome labels by (`subject_id`, `hadm_id`). For MIMIC-IV, this cohort is restricted to ICD-10 admissions.

We then create train/validation/test splits at the *patient* level by assigning each `subject_id` to a split via a deterministic hash with proportions 70%/10%/20%, and placing all of that patient's admissions in the same split. This avoids leakage across repeated admissions from the same individual and yields an evaluation that reflects generalization to previously unseen patients.

### A.10. Metric computation

The held-out dependence metrics reported in Tables 1–2 are computed *per outcome* using the task-specific validity mask for that outcome. Thus, for each outcome column, all models are evaluated on the same valid test admissions. For outcome $t$, let

$$\mathcal{I}_t = \{i : M_i^{(t)} = 1\}$$

denote admissions with a defined label. For any scalar score $q_i$ being evaluated, we compute each metric using pairs

$$\{(q_i, y_i^{(t)})\}_{i \in \mathcal{I}_t}.$$

If $|\mathcal{I}_t| < 3$ or $y_i^{(t)}$ is single-class on $\mathcal{I}_t$, we return 0.

**Distance correlation (dCorr).** We compute distance correlation between the scalar score and the binary outcome on $\mathcal{I}_t$ using the standard distance correlation implementation (`dcor.distance_correlation`). Because dCorr is signless, it measures the strength of dependence rather than direction.

**Mutual information (MI, nats).** We estimate mutual information between a scalar score $q_i$ and binary label $y_i^{(t)}$ on $\mathcal{I}_t$ using the $k$-nearest-neighbor estimator implemented in `sklearn.feature_selection.mutual_info_classif`. We first standardize scores on the valid subset by z-scoring, and use $k_{\text{MI}} = \min(k_0, |\mathcal{I}_t| - 1)$ neighbors with fixed random seed to avoid invalid settings when few valid labels are available. The resulting estimate is reported in nats, the natural-logarithm units of mutual information.

# B. Rank-One Multi-Task nHSIC Theory: Roadmap, Assumptions, and Notation

**Overview.** We study the multi-task dependence structure induced by normalized HSIC, where each task compares a centered score Gram matrix with a centered label Gram matrix. For binary labels, the centered label Gram matrix reduces to a rank-one object determined by the centered label profile. This lets us rewrite the weighted multi-task objective as an alignment between the centered score Gram matrix and a single cross-task label-alignment matrix.

The main idea is then to isolate the dominant shared component of this cross-task label structure. We stack the weighted centered label profiles across tasks and project the resulting matrix onto its best rank-one approximation. The projected objective reduces exactly to a single-admission-direction problem governed by the leading right singular vector $v$. Applying the global nHSIC ratio lemma to this direction yields an explicit two-level monotone threshold rule: the best split is the one whose centered step vector is most aligned with $v$.

Finally, when the rank-one residual makes the explicit bound $\varepsilon_{\text{gap}}$ small, the appendix transfers near-optimality from the projected objective back to the original multi-task objective using an $\varepsilon_0$-floored denominator. We also include a noise-averaging bridge showing that, when the learned-score Gram matrix concentrates around its noise average, the realized learned-score nHSIC is close to the corresponding averaged-kernel target.

**Finite-sample nHSIC objective.** For each task $t$, the appendix studies

$$\text{nHSIC}\left(r, y^{(t)}\right) := \frac{\langle K_c(r), L_{t,c}\rangle_F}{\|K_c(r)\|_F \|L_{t,c}\|_F}, \qquad K_c(r) := HK(r)H, \quad L_{t,c} := HL_t H,$$

with the convention that the ratio is 0 whenever a required denominator factor is zero. The multi-task objective is

$$J(r) := \sum_{t=1}^{T} \alpha_t \, \text{nHSIC}\left(r, y^{(t)}\right), \qquad \alpha_t > 0.$$

**Proof dependencies.** The appendix proceeds as follows.

1. Lemma B.1 shows that binary delta-label Grams satisfy $L_{t,c} = 2\ell^{(t)}\ell^{(t)\top}$, $\qquad \ell^{(t)} := Hy^{(t)}$.

2. Lemma B.2 studies
$$\bar{\Delta}_w(r) := \frac{w^\top K_c(r)w}{\|K_c(r)\|_F}, \qquad \mathbf{1}^\top w = 0,$$
proves $\bar{\Delta}_w(r) \leq \|w\|_2^2$, and gives the exact threshold value
$$\bar{\Delta}_w(r^{(j)}) = \|w\|_2^2 \rho_j^2, \qquad \rho_j := \frac{w^\top g_j}{\|w\|_2 \|g_j\|_2}, \qquad g_j := H\mathbf{1}_{R_j}.$$

3. Lemma B.3 proves the rank-one projection reduction $J_1(r) = \sigma_1^2 \bar{\Delta}_v(r)$ for $\widetilde{W}_1 = \sigma_1 uv^\top$.

4. Theorem B.4 defines the exact uniform gap $\Delta_{\text{gap}}$ and proves the explicit bound $\Delta_{\text{gap}} \leq \varepsilon_{\text{gap}}$. Corollary B.5 then transfers near-optimality from the floored projected objective to the original floored objective using this bound.

5. Lemma B.6 shows that if $K_c(s)$ concentrates around $\bar{K}_c := \mathbb{E}_\eta[K_c(s)]$, then $\text{nHSIC}(s, y^{(t)})$ is close to the averaged-kernel normalized target.

## B.1. Core assumptions and conventions

Unless stated otherwise, we assume:

**(A1)** Admissions are indexed by latent severity: $z_1 < \cdots < z_n$.

**(A2)** Each included task is nonconstant: $\|Hy^{(t)}\|_2 > 0$.

**(A3)** Scores lie in the bounded monotone class: $\mathcal{R}_B^\uparrow := \{r \in [-B, B]^n : r_1 \leq \cdots \leq r_n\}$.

**(A4)** The score kernel is radial, bounded, continuous, and positive semidefinite:

$$K_{ij}(r) = k(r_i, r_j) = \kappa(|r_i - r_j|),$$

where $\kappa$ is strictly decreasing on the relevant range $[0, 2B]$. We assume that $k$ is positive semidefinite on $[-B, B]$, so that for every $r \in \mathcal{R}_B^\uparrow$, the Gram matrix $K(r)$ is symmetric positive semidefinite. Consequently,

$$K_c(r) = HK(r)H$$

is also symmetric positive semidefinite.

**(A5)** All objectives use centered Gram matrices:

$$K_c(r) = HK(r)H, \qquad L_{t,c} = HL_t H, \qquad H = I - \frac{1}{n}\mathbf{1}\mathbf{1}^\top.$$

**(A6)** When invoked, the best rank-one approximation is $\widetilde{W}_1 = \sigma_1 uv^\top$, with $\sigma_1 \geq 0$, $\|u\|_2 = \|v\|_2 = 1$. Since each row of $\widetilde{W}$ is centered, $\widetilde{W}\mathbf{1}_n = 0$, where $\mathbf{1}_n \in \mathbb{R}^n$ denotes the all-ones vector. Hence, whenever $\sigma_1 > 0$, the associated right singular vector satisfies $\mathbf{1}^\top v = 0$, equivalently $Hv = v$.

**(A7)** nHSIC-type ratios are set to 0 whenever a required denominator factor is zero. Floored objectives use $\max\{\|K_c(r)\|_F, \varepsilon_0\}$.

## B.2. Notation at a glance

| Symbol | Meaning |
|---|---|
| $z_1 < \cdots < z_n$ | Ordered admissions by latent severity |
| $r = (r_1, \ldots, r_n)^\top$ | Monotone score vector assigned to the ordered admissions |
| $\mathcal{R}_B^\uparrow$ | Bounded monotone score class $\{r \in [-B, B]^n : r_1 \leq \cdots \leq r_n\}$ |
| $y^{(t)} \in \{0,1\}^n$ | Binary labels for task $t$ |
| $[T]$ | Task index set $\{1, \ldots, T\}$ |
| $H$ | Centering matrix $I - \frac{1}{n}\mathbf{1}\mathbf{1}^\top$ |
| $K(r)$ | Score Gram matrix, $K_{ij}(r) = \kappa(|r_i - r_j|)$ |
| $K_c(r)$ | Centered score Gram matrix $HK(r)H$ |
| $L_t$ | Delta label Gram matrix, $(L_t)_{ij} = \mathbb{I}\{y_i^{(t)} = y_j^{(t)}\}$ |
| $L_{t,c}$ | Centered label Gram matrix $HL_t H$ |
| $\ell^{(t)}$ | Centered label profile $Hy^{(t)}$ |
| $\mathrm{nHSIC}(r, y^{(t)})$ | Per-task normalized HSIC objective |
| $\alpha_t$ | Fixed task weight |
| $\beta_t$ | $\alpha_t / \|\ell^{(t)}\|_2^2$ |
| $\widetilde{w}^{(t)}$ | $\sqrt{\beta_t}\ell^{(t)}$ |
| $\widetilde{W}$ | Stacked weighted label-profile matrix |
| $J(r)$ | Multi-task objective $\sum_t \alpha_t \,\mathrm{nHSIC}(r, y^{(t)})$ |
| $\widetilde{W}_1$ | Best rank-one approximation $\sigma_1 uv^\top$ |
| $J_1(r)$ | Projected rank-one objective |
| $\bar{\Delta}_w(r)$ | Single-direction ratio $w^\top K_c(r)w/\|K_c(r)\|_F$ |
| $\bar{\Delta}_v(r)$ | Rank-one admission-direction ratio $v^\top K_c(r)v/\|K_c(r)\|_F$ |
| $L_j, R_j$ | Threshold split sets $\{1, \ldots, j\}, \{j+1, \ldots, n\}$ |
| $\mathbf{1}_{R_j}$ | Indicator vector of $R_j$ |
| $g_j$ | Centered step vector $H\mathbf{1}_{R_j}$ |
| $\rho_j$ | Cosine correlation $\frac{w^\top g_j}{\|w\|_2 \|g_j\|_2}$, for the fixed direction $w$ in the threshold lemma |
| $j^\star$ | Best threshold split index $\arg\max_j \rho_j^2$ |
| $\varepsilon_0$ | Denominator floor |
| $\Delta_{\mathrm{gap}}$ | Exact uniform gap $\sup_{r \in \mathcal{R}_B^\uparrow} |J_{\varepsilon_0}(r) - J_{1,\varepsilon_0}(r)|$ |
| $\varepsilon_{\mathrm{gap}}$ | Explicit upper bound satisfying $\Delta_{\mathrm{gap}} \leq \varepsilon_{\mathrm{gap}}$ |
| $\bar{K}_c$ | Noise-averaged centered score Gram matrix $\mathbb{E}_\eta[K_c(s)]$ |

**Lemma B.1** (Centered delta-kernel is rank one for binary labels). *Let $y \in \{0,1\}^n$ and define the delta label Gram matrix $L_{ij} := \mathbb{I}\{y_i = y_j\}$, where $\mathbb{I}\{\cdot\}$ denotes the indicator function. Let $L_c := HLH$. Then $L_c = 2(Hy)(Hy)^\top$, so $\mathrm{rank}(L_c) \leq 1$. Moreover, $\|L_c\|_F = 2\|Hy\|_2^2$.*

*Proof.* Using $L_{ij} = y_i y_j + (1 - y_i)(1 - y_j) = 1 - y_i - y_j + 2y_i y_j$, we get $L = \mathbf{1}\mathbf{1}^\top - y\mathbf{1}^\top - \mathbf{1}y^\top + 2yy^\top$. Since $H\mathbf{1} = 0$, centering kills the first three terms and

$$L_c = H(2yy^\top)H = 2(Hy)(Hy)^\top.$$

The Frobenius norm follows from $\|uu^\top\|_F = \|u\|_2^2$. $\qquad\square$

**Lemma B.2** (Global upper bound and threshold certificate for the nHSIC ratio). *Consider $n$ admissions with ordered latent severities $z_1 < \cdots < z_n$, and fix any centered target direction $w \in \mathbb{R}^n$, so $\mathbf{1}^\top w = 0$. If $w = 0$, then $\bar{\Delta}_w \equiv 0$ and all statements are trivial. Hence assume $w \neq 0$ below.*

*Let $k(x, x') = \kappa(|x - x'|)$ be a bounded continuous positive semidefinite radial kernel, where $\kappa$ is strictly decreasing on $[0, 2B]$. For any score vector $r \in \mathcal{R}_B^\uparrow$, define*

$$K_{ij}(r) := \kappa(|r_i - r_j|), \qquad K_c(r) := HK(r)H, \qquad H := I - \frac{1}{n}\mathbf{1}\mathbf{1}^\top.$$

*Define*

$$\bar{\Delta}_w(r) := \frac{w^\top K_c(r)w}{\|K_c(r)\|_F},$$

*with the convention $\bar{\Delta}_w(r) = 0$ whenever $\|K_c(r)\|_F = 0$.*

*For each split index $j \in \{1, \ldots, n-1\}$, define*

$$L_j := \{1, \ldots, j\}, \qquad R_j := \{j+1, \ldots, n\},$$

*and*

$$g_j := H\mathbf{1}_{R_j} = \mathbf{1}_{R_j} - \frac{|R_j|}{n}\mathbf{1},$$

*where $\mathbf{1}_{R_j} \in \mathbb{R}^n$ is the indicator vector of $R_j$. Since $w \neq 0$ and $g_j \neq 0$, define*

$$\rho_j := \frac{w^\top g_j}{\|w\|_2 \|g_j\|_2}.$$

*Then:*

1. *For every $r \in \mathcal{R}_B^\uparrow$,*

$$\bar{\Delta}_w(r) \leq \|w\|_2^2.$$

2. *For any nontrivial two-level threshold score $r^{(j)} \in \mathcal{R}_B^\uparrow$ of the form*

$$r_i^{(j)} = \begin{cases} a, & i \in L_j, \\ b, & i \in R_j, \end{cases} \qquad a < b,$$

   *we have*

$$\bar{\Delta}_w(r^{(j)}) = \|w\|_2^2 \rho_j^2.$$

   *In particular, this value is independent of the separation $|a - b|$.*

3. *Let*

$$\rho_\star^2 := \max_{1 \leq j \leq n-1} \rho_j^2,$$

*and let $j^\star$ attain the maximum. Then any nontrivial two-level threshold on $L_{j^\star}/R_{j^\star}$ satisfies*

$$\bar{\Delta}_w(r^{(j^\star)}) = \|w\|_2^2 \rho_\star^2$$

*and hence*

$$\bar{\Delta}_w(r^{(j^\star)}) \geq \rho_\star^2 \sup_{u \in \mathcal{R}_B^\uparrow} \bar{\Delta}_w(u).$$

*Thus $\rho_\star^2$ is an explicit certificate for the fraction of the global upper bound achieved by the best threshold.*

4. *There exists a nontrivial two-level threshold $r^{(j)}$ that achieves the global upper bound*

$$\bar{\Delta}_w(r^{(j)}) = \|w\|_2^2$$

*if and only if $\rho_\star^2 = 1$, equivalently $w$ is proportional to some centered step vector $g_j$.*

*Proof.* Since $\mathbf{1}^\top w = 0$, we have $Hw = w$. Therefore

$$w^\top K_c(r)w = w^\top HK(r)Hw = w^\top K(r)w.$$

Also,

$$w^\top K_c(r)w = \langle K_c(r), ww^\top \rangle_F.$$

Hence

$$\bar{\Delta}_w(r) = \frac{\langle K_c(r), ww^\top \rangle_F}{\|K_c(r)\|_F}.$$

If $\|K_c(r)\|_F = 0$, then $\bar{\Delta}_w(r) = 0$ by convention, so the bound is immediate. Otherwise, by Cauchy–Schwarz in the Frobenius inner product,

$$\bar{\Delta}_w(r) \leq \frac{\|K_c(r)\|_F \|ww^\top\|_F}{\|K_c(r)\|_F} = \|ww^\top\|_F = \|w\|_2^2,$$

which proves the global upper bound.

Now fix a nontrivial two-level threshold score $r^{(j)}$, with value $a$ on $L_j$ and value $b$ on $R_j$, where $a < b$. Let

$$c_0 := \kappa(0), \qquad c_1 := \kappa(|a - b|).$$

Since $\kappa$ is strictly decreasing and $a < b$, we have $c_0 - c_1 > 0$. The score Gram matrix is block-constant and can be written as

$$K(r^{(j)}) = c_1 \mathbf{1}\mathbf{1}^\top + (c_0 - c_1)\left(\mathbf{1}_{L_j}\mathbf{1}_{L_j}^\top + \mathbf{1}_{R_j}\mathbf{1}_{R_j}^\top\right).$$

Centering removes the first term because $H\mathbf{1} = 0$, so

$$K_c(r^{(j)}) = (c_0 - c_1)H\left(\mathbf{1}_{L_j}\mathbf{1}_{L_j}^\top + \mathbf{1}_{R_j}\mathbf{1}_{R_j}^\top\right)H.$$

Since

$$\mathbf{1}_{L_j} = \mathbf{1} - \mathbf{1}_{R_j},$$

we have

$$H\mathbf{1}_{L_j} = H\mathbf{1} - H\mathbf{1}_{R_j} = -g_j, \qquad H\mathbf{1}_{R_j} = g_j.$$

Therefore

$$H\left(\mathbf{1}_{L_j}\mathbf{1}_{L_j}^\top + \mathbf{1}_{R_j}\mathbf{1}_{R_j}^\top\right)H = (H\mathbf{1}_{L_j})(H\mathbf{1}_{L_j})^\top + (H\mathbf{1}_{R_j})(H\mathbf{1}_{R_j})^\top$$

$$= (-g_j)(-g_j)^\top + g_j g_j^\top = 2g_j g_j^\top.$$

Thus

$$K_c(r^{(j)}) = 2(c_0 - c_1)g_j g_j^\top.$$

Using $\|g_j g_j^\top\|_F = \|g_j\|_2^2$, we get

$$\|K_c(r^{(j)})\|_F = 2(c_0 - c_1)\|g_j\|_2^2.$$

Therefore

$$\bar{\Delta}_w(r^{(j)}) = \frac{\left\langle 2(c_0 - c_1)g_j g_j^\top, ww^\top \right\rangle_F}{2(c_0 - c_1)\|g_j\|_2^2}.$$

Since

$$\langle g_j g_j^\top, ww^\top \rangle_F = (w^\top g_j)^2,$$

we obtain

$$\bar{\Delta}_w(r^{(j)}) = \frac{(w^\top g_j)^2}{\|g_j\|_2^2} = \|w\|_2^2 \left( \frac{w^\top g_j}{\|w\|_2 \|g_j\|_2} \right)^2 = \|w\|_2^2 \rho_j^2.$$

This proves the exact threshold value and shows that $|a - b|$ cancels.

The best threshold statement follows by choosing $j^\star \in \arg\max_j \rho_j^2$. Since the global upper bound is $\|w\|_2^2$, the value

$$\|w\|_2^2 \rho_\star^2$$

is a $\rho_\star^2$-fraction of the global upper bound.

Finally, for a nontrivial two-level threshold,

$$K_c(r^{(j)}) = 2(c_0 - c_1)g_j g_j^\top,$$

with $2(c_0 - c_1) > 0$. Hence this threshold attains the global upper bound $\|w\|_2^2$ if and only if

$$\|w\|_2^2 \rho_j^2 = \|w\|_2^2,$$

equivalently $\rho_j^2 = 1$. This is equivalent to equality in the Cauchy–Schwarz inequality between $w$ and $g_j$, hence to $w \propto g_j$. Maximizing over $j$ gives the condition $\rho_\star^2 = 1$. $\qquad\square$

**Unfloored vs. $\varepsilon_0$-floored normalization.** Lemma B.2 characterizes the exact geometry of the unfloored ratio

$$\bar{\Delta}_w(r) = \frac{w^\top K_c(r) w}{\|K_c(r)\|_F}.$$

In later approximation and stability results, we introduce an $\varepsilon_0$-floor in the denominator to avoid degeneracy for nearly constant scores. When $\|K_c(r)\|_F \geq \varepsilon_0$, the unfloored and floored objectives coincide.

**Set up the multi-task finite-sample problem (nHSIC form).** Fix $n$ admissions with strictly ordered latent severities

$$z_1 < z_2 < \cdots < z_n.$$

For each task $t \in [T] := \{1, \ldots, T\}$, let

$$y^{(t)} \in \{0, 1\}^n$$

denote the observed binary labels on these $n$ admissions. Assume each included task is nonconstant, equivalently

$$\|Hy^{(t)}\|_2 > 0.$$

Let $B > 0$ be fixed, and define

$$\mathcal{R}_B^\uparrow := \{r \in [-B, B]^n : r_1 \leq \cdots \leq r_n\}.$$

Write $r = (r_1, \ldots, r_n)^\top$. Let $k(x, x') = \kappa(|x - x'|)$ be a bounded continuous positive semidefinite radial kernel, where $\kappa$ is strictly decreasing on $[0, 2B]$. Define

$$K_{ij}(r) := \kappa(|r_i - r_j|), \qquad K_c(r) := HK(r)H, \qquad H := I - \frac{1}{n}\mathbf{1}\mathbf{1}^\top.$$

**Single-task nHSIC objective.** For each task $t$, let $L_t$ be the delta label Gram matrix

$$(L_t)_{ij} := \mathbb{I}\{y_i^{(t)} = y_j^{(t)}\},$$

and define its centered version

$$L_{t,c} := HL_tH.$$

The finite-sample normalized HSIC objective is

$$\mathrm{nHSIC}\left(r, y^{(t)}\right) := \frac{\langle K_c(r), L_{t,c}\rangle_F}{\|K_c(r)\|_F \|L_{t,c}\|_F}.$$

We set $\mathrm{nHSIC}(r, y^{(t)}) = 0$ whenever either denominator factor is zero. By Lemma B.1, for binary labels,

$$L_{t,c} = 2\ell^{(t)}\ell^{(t)\top}, \qquad \ell^{(t)} := Hy^{(t)},$$

and

$$\|L_{t,c}\|_F = 2\|\ell^{(t)}\|_2^2, \qquad \langle K_c(r), L_{t,c}\rangle_F = 2\ell^{(t)\top}K_c(r)\ell^{(t)}.$$

Therefore, when $\|K_c(r)\|_F > 0$,

$$\mathrm{nHSIC}\left(r, y^{(t)}\right) = \frac{\ell^{(t)\top}K_c(r)\ell^{(t)}}{\|K_c(r)\|_F \|\ell^{(t)}\|_2^2}.$$

When $\|K_c(r)\|_F = 0$, we keep the convention $\mathrm{nHSIC}(r, y^{(t)}) = 0$.

**Multi-task weighted-sum nHSIC objective.** Let $\alpha_t > 0$ be fixed task weights and define

$$J(r) := \sum_{t=1}^{T} \alpha_t \, \mathrm{nHSIC}\left(r, y^{(t)}\right).$$

Using the previous display, on the event $\|K_c(r)\|_F > 0$,

$$J(r) = \frac{1}{\|K_c(r)\|_F} \sum_{t=1}^{T} \frac{\alpha_t}{\|\ell^{(t)}\|_2^2} \ell^{(t)\top}K_c(r)\ell^{(t)}.$$

If $\|K_c(r)\|_F = 0$, then $J(r) = 0$ by convention.

**Absorbing label norms into weights.** Define

$$\beta_t := \frac{\alpha_t}{\|\ell^{(t)}\|_2^2}, \qquad \widetilde{w}^{(t)} := \sqrt{\beta_t}\,\ell^{(t)}.$$

Let $\widetilde{W} \in \mathbb{R}^{T\times n}$ have rows

$$\widetilde{W}_{t,:} = \widetilde{w}^{(t)\top} = \sqrt{\beta_t}\,\ell^{(t)\top}.$$

Then

$$\widetilde{W}^{\top}\widetilde{W} = \sum_{t=1}^{T} \beta_t\,\ell^{(t)}\ell^{(t)\top}.$$

Hence, when $\|K_c(r)\|_F > 0$,

$$J(r) = \frac{\langle K_c(r), \widetilde{W}^{\top}\widetilde{W}\rangle_F}{\|K_c(r)\|_F}.$$

When $\|K_c(r)\|_F = 0$, we keep the convention $J(r) = 0$.

**Centering of task weight vectors.** Since $\ell^{(t)} = Hy^{(t)}$, we have $\mathbf{1}^{\top}\ell^{(t)} = 0$. Therefore

$$\mathbf{1}^{\top}\widetilde{w}^{(t)} = \sqrt{\beta_t}\,\mathbf{1}^{\top}\ell^{(t)} = 0$$

for all $t$. Equivalently,

$$\widetilde{W}\mathbf{1} = 0.$$

**Rank-one projected objective.** Let $\widetilde{W}_1$ be the best rank-one approximation to $\widetilde{W}$ in Frobenius norm:

$$\widetilde{W}_1 := \sigma_1 uv^\top, \qquad \sigma_1 \geq 0, \qquad \|u\|_2 = \|v\|_2 = 1.$$

Since $\widetilde{W}\mathbf{1} = 0$, we have

$$(\widetilde{W}^\top \widetilde{W})\mathbf{1} = 0.$$

If $\sigma_1 > 0$, then $v$ is an eigenvector of $\widetilde{W}^\top \widetilde{W}$ with eigenvalue $\sigma_1^2$, so

$$v \perp \mathbf{1}, \qquad Hv = v.$$

If $\sigma_1 = 0$, then $\widetilde{W}_1 = 0$ and the projected objective below is identically zero.

Define

$$J_1(r) := \frac{\langle K_c(r), \widetilde{W}_1^\top \widetilde{W}_1 \rangle_F}{\|K_c(r)\|_F},$$

with $J_1(r) = 0$ whenever $\|K_c(r)\|_F = 0$.

**Lemma B.3** (Reduction under rank-one projection). *For every $r \in \mathcal{R}_B^\uparrow$ with $\|K_c(r)\|_F > 0$,*

$$J_1(r) = \sigma_1^2 \bar{\Delta}_v(r), \qquad \bar{\Delta}_v(r) := \frac{v^\top K_c(r)v}{\|K_c(r)\|_F}.$$

*If $\sigma_1 > 0$, then*

$$\arg \max_{r \in \mathcal{R}_B^\uparrow : \|K_c(r)\|_F > 0} J_1(r) = \arg \max_{r \in \mathcal{R}_B^\uparrow : \|K_c(r)\|_F > 0} \bar{\Delta}_v(r).$$

*If $\sigma_1 = 0$, then $J_1 \equiv 0$.*

*Proof.* Since $\widetilde{W}_1 = \sigma_1 uv^\top$ with $\|u\|_2 = 1$,

$$\widetilde{W}_1^\top \widetilde{W}_1 = \sigma_1^2 vv^\top.$$

Substituting into $J_1(r)$ gives

$$J_1(r) = \sigma_1^2 \frac{\langle K_c(r), vv^\top \rangle_F}{\|K_c(r)\|_F} = \sigma_1^2 \frac{v^\top K_c(r)v}{\|K_c(r)\|_F} = \sigma_1^2 \bar{\Delta}_v(r).$$

The maximizer statement follows because multiplication by the positive constant $\sigma_1^2$ does not change the argmax. If $\sigma_1 = 0$, then $\widetilde{W}_1^\top \widetilde{W}_1 = 0$, so $J_1 \equiv 0$. $\square$

**Threshold implication for the rank-one direction.** Assume $\sigma_1 > 0$. Since $Hv = v$, Lemma B.2 applies with $w = v$. For each split

$$L_j := \{1, \dots, j\}, \qquad R_j := \{j+1, \dots, n\},$$

define

$$g_j := H\mathbf{1}_{R_j}, \qquad \rho_j := \frac{v^\top g_j}{\|v\|_2 \|g_j\|_2}.$$

Then every nontrivial monotone two-level threshold on split $j$ satisfies

$$\bar{\Delta}_v(r^{(j)}) = \|v\|_2^2 \rho_j^2.$$

Thus the best threshold split is

$$j^\star \in \arg \max_{1 \leq j \leq n-1} \rho_j^2,$$

and it achieves a fraction

$$\rho_\star^2 := \max_{1 \leq j \leq n-1} \rho_j^2$$

of the global upper bound for the rank-one direction objective. Since $J_1 = \sigma_1^2 \bar{\Delta}_v$, the same split also maximizes $J_1$ over two-level thresholds.

**Uniform approximation bound for the rank-one projection.** Define

$$\kappa_{\max}^{\text{abs}} := \sup_{d \in [0,2B]} |\kappa(d)| < \infty.$$

Let

$$E := \widetilde{W} - \widetilde{W}_1, \qquad e^{(t)} := \widetilde{w}^{(t)} - \widehat{\widetilde{w}}^{(t)},$$

where

$$\widehat{\widetilde{w}}^{(t)} := \sigma_1 u_t v.$$

When $\sigma_1 > 0$, we have $Hv = v$, so $\widehat{\widetilde{w}}^{(t)}$ is centered. When $\sigma_1 = 0$, $\widehat{\widetilde{w}}^{(t)} = 0$, so it is centered as well. Fix $\varepsilon_0 > 0$ and define

$$J_{\varepsilon_0}(r) := \frac{\langle K_c(r), \widetilde{W}^\top \widetilde{W} \rangle_F}{\max\{\|K_c(r)\|_F, \varepsilon_0\}}, \qquad J_{1,\varepsilon_0}(r) := \frac{\langle K_c(r), \widetilde{W}_1^\top \widetilde{W}_1 \rangle_F}{\max\{\|K_c(r)\|_F, \varepsilon_0\}}.$$

**Theorem B.4** (Uniform error bound for $J_{\varepsilon_0}$ versus $J_{1,\varepsilon_0}$). *For every $r \in \mathcal{R}_B^\uparrow$,*

$$|J_{\varepsilon_0}(r) - J_{1,\varepsilon_0}(r)| \le \frac{\kappa_{\max}^{\text{abs}}}{\varepsilon_0} \sum_{t=1}^{T} \sum_{i,j=1}^{n} \left| \widetilde{w}_i^{(t)} \widetilde{w}_j^{(t)} - \widehat{\widetilde{w}}_i^{(t)} \widehat{\widetilde{w}}_j^{(t)} \right|.$$

*Moreover, for each task $t$,*

$$\sum_{i,j=1}^{n} \left| \widetilde{w}_i^{(t)} \widetilde{w}_j^{(t)} - \widehat{\widetilde{w}}_i^{(t)} \widehat{\widetilde{w}}_j^{(t)} \right| \le \left( \|\widetilde{w}^{(t)}\|_1 + \|\widehat{\widetilde{w}}^{(t)}\|_1 \right) \|e^{(t)}\|_1.$$

*Consequently, the exact uniform gap*

$$\Delta_{\text{gap}} := \sup_{r \in \mathcal{R}_B^\uparrow} |J_{\varepsilon_0}(r) - J_{1,\varepsilon_0}(r)|$$

*satisfies*

$$\Delta_{\text{gap}} \le \varepsilon_{\text{gap}},$$

*where the explicit upper bound is*

$$\varepsilon_{\text{gap}} := \frac{\kappa_{\max}^{\text{abs}}}{\varepsilon_0} \sum_{t=1}^{T} \left( \|\widetilde{w}^{(t)}\|_1 + \|\widehat{\widetilde{w}}^{(t)}\|_1 \right) \|e^{(t)}\|_1.$$

*Proof.* If $\|K_c(r)\|_F = 0$, then $J_{\varepsilon_0}(r) = J_{1,\varepsilon_0}(r) = 0$. Otherwise, since $r_i \in [-B, B]$,

$$|r_i - r_j| \in [0, 2B], \qquad |\kappa(|r_i - r_j|)| \le \kappa_{\max}^{\text{abs}}.$$

Because $\widetilde{w}^{(t)}$ and $\widehat{\widetilde{w}}^{(t)}$ are centered,

$$(\widetilde{w}^{(t)})^\top K_c(r) \widetilde{w}^{(t)} = (\widetilde{w}^{(t)})^\top K(r) \widetilde{w}^{(t)},$$

and similarly for $\widehat{\widetilde{w}}^{(t)}$. Hence

$$J_{\varepsilon_0}(r) - J_{1,\varepsilon_0}(r) = \frac{\sum_{t=1}^{T} \sum_{i,j=1}^{n} \left( \widetilde{w}_i^{(t)} \widetilde{w}_j^{(t)} - \widehat{\widetilde{w}}_i^{(t)} \widehat{\widetilde{w}}_j^{(t)} \right) K_{ij}(r)}{\max\{\|K_c(r)\|_F, \varepsilon_0\}}.$$

Taking absolute values and using $K_{ij}(r) = \kappa(|r_i - r_j|)$ proves the first bound.

For the second bound, expand

$$\widetilde{w}_i^{(t)} \widetilde{w}_j^{(t)} - \widehat{\widetilde{w}}_i^{(t)} \widehat{\widetilde{w}}_j^{(t)} = \widetilde{w}_i^{(t)} e_j^{(t)} + \widehat{\widetilde{w}}_j^{(t)} e_i^{(t)}.$$

Therefore

$$\left| \widetilde{w}_i^{(t)} \widetilde{w}_j^{(t)} - \widehat{\widetilde{w}}_i^{(t)} \widehat{\widetilde{w}}_j^{(t)} \right| \le |\widetilde{w}_i^{(t)}| \, |e_j^{(t)}| + |\widehat{\widetilde{w}}_j^{(t)}| \, |e_i^{(t)}|.$$

Summing over $i, j$ gives

$$\sum_{i,j=1}^{n} \left| \widetilde{w}_i^{(t)} \widetilde{w}_j^{(t)} - \widehat{\widetilde{w}}_i^{(t)} \widehat{\widetilde{w}}_j^{(t)} \right| \leq \left( \|\widetilde{w}^{(t)}\|_1 + \|\widehat{\widetilde{w}}^{(t)}\|_1 \right) \|e^{(t)}\|_1.$$

Substituting this into the first bound yields the stated uniform bound. □

**Consequence: near-optimality of the projected maximizer for the floored objective.**

**Corollary B.5.** *Let*

$$r_1^\star \in \arg \max_{r \in \mathcal{R}_B^\uparrow} J_{1,\varepsilon_0}(r).$$

*Let $\Delta_{\mathrm{gap}}$ and $\varepsilon_{\mathrm{gap}}$ be as in Theorem B.4. Then*

$$J_{\varepsilon_0}(r_1^\star) \geq \sup_{r \in \mathcal{R}_B^\uparrow} J_{\varepsilon_0}(r) - 2\varepsilon_{\mathrm{gap}}.$$

*Proof.* By definition of $\Delta_{\mathrm{gap}}$, for every $r \in \mathcal{R}_B^\uparrow$,

$$J_{\varepsilon_0}(r) \geq J_{1,\varepsilon_0}(r) - \Delta_{\mathrm{gap}}, \qquad J_{1,\varepsilon_0}(r) \geq J_{\varepsilon_0}(r) - \Delta_{\mathrm{gap}}.$$

By optimality of $r_1^\star$ for $J_{1,\varepsilon_0}$, for every $r \in \mathcal{R}_B^\uparrow$,

$$J_{1,\varepsilon_0}(r_1^\star) \geq J_{1,\varepsilon_0}(r).$$

Therefore, for every $r \in \mathcal{R}_B^\uparrow$,

$$J_{\varepsilon_0}(r_1^\star) \geq J_{1,\varepsilon_0}(r_1^\star) - \Delta_{\mathrm{gap}} \geq J_{1,\varepsilon_0}(r) - \Delta_{\mathrm{gap}} \geq J_{\varepsilon_0}(r) - 2\Delta_{\mathrm{gap}}.$$

Since this holds for every $r \in \mathcal{R}_B^\uparrow$, it also holds after taking the supremum over $r$. Using $\Delta_{\mathrm{gap}} \leq \varepsilon_{\mathrm{gap}}$, we obtain

$$J_{\varepsilon_0}(r_1^\star) \geq \sup_{r \in \mathcal{R}_B^\uparrow} J_{\varepsilon_0}(r) - 2\varepsilon_{\mathrm{gap}}.$$

□

**Lemma B.6** (Approximate noise-averaging for normalized nHSIC). *Condition on a fixed task $t$ and the fixed data $(r, y^{(t)})$. Let*

$$s_i = r_i + \eta_i$$

*with i.i.d. noise. Let*

$$\bar{K}_c := \mathbb{E}_\eta[K_c(s)]$$

*and*

$$\overline{\mathrm{nHSIC}}\left(r, y^{(t)}\right) := \frac{\langle \bar{K}_c, L_{t,c} \rangle_F}{\|\bar{K}_c\|_F \|L_{t,c}\|_F}.$$

*Assume $\|L_{t,c}\|_F > 0$ and $\|\bar{K}_c\|_F \geq \gamma > 0$. On any event where*

$$\|K_c(s) - \bar{K}_c\|_F \leq \gamma/2,$$

*we have both $\|K_c(s)\|_F \geq \gamma/2$ and*

$$\left| \mathrm{nHSIC}\left(s, y^{(t)}\right) - \overline{\mathrm{nHSIC}}\left(r, y^{(t)}\right) \right| \leq \frac{4}{\gamma} \|K_c(s) - \bar{K}_c\|_F.$$

*Proof.* Let

$$F(K) := \frac{\langle K, L_{t,c}\rangle_F}{\|K\|_F}.$$

On the event $\|K_c(s) - \bar{K}_c\|_F \leq \gamma/2$, the reverse triangle inequality gives

$$\|K_c(s)\|_F \geq \|\bar{K}_c\|_F - \|K_c(s) - \bar{K}_c\|_F \geq \gamma/2.$$

Now take any $K, K'$ with $\|K\|_F, \|K'\|_F \geq \gamma/2$. Then

$$
\begin{aligned}
|F(K) - F(K')| &= \left| \frac{\langle K, L_{t,c}\rangle_F}{\|K\|_F} - \frac{\langle K', L_{t,c}\rangle_F}{\|K'\|_F} \right| \\
&\leq \left| \frac{\langle K - K', L_{t,c}\rangle_F}{\|K\|_F} \right| + |\langle K', L_{t,c}\rangle_F| \left| \frac{1}{\|K\|_F} - \frac{1}{\|K'\|_F} \right| \\
&\leq \frac{2}{\gamma} \|K - K'\|_F \|L_{t,c}\|_F + \|L_{t,c}\|_F \frac{2}{\gamma} \|K - K'\|_F \\
&= \frac{4\|L_{t,c}\|_F}{\gamma} \|K - K'\|_F.
\end{aligned}
$$

In the second term, we used

$$|\langle K', L_{t,c}\rangle_F| \leq \|K'\|_F \|L_{t,c}\|_F$$

and

$$\left| \frac{1}{\|K\|_F} - \frac{1}{\|K'\|_F} \right| = \frac{\left| \|K'\|_F - \|K\|_F \right|}{\|K\|_F \|K'\|_F} \leq \frac{\|K - K'\|_F}{\|K\|_F \|K'\|_F}.$$

Taking $K = K_c(s)$ and $K' = \bar{K}_c$, then dividing by $\|L_{t,c}\|_F > 0$, converts $F(K_c(s))$ into $\mathrm{nHSIC}(s, y^{(t)})$ and $F(\bar{K}_c)$ into $\overline{\mathrm{nHSIC}}(r, y^{(t)})$. This proves the result. $\qquad\square$

## C. Additional Experiments: Single-Index Baselines and nHSIC Ablations

We report additional dependence-based evaluations to further compare MLCI against BCE-trained single-index models and to assess the sensitivity of the proposed nHSIC framework to encoder architecture and score-kernel choice. The BCE-trained models serve as additional predictive baselines: they use the same diagnosis-code inputs and produce scalar scores, but they are optimized with supervised binary cross-entropy objectives rather than the proposed multi-outcome nHSIC objective. In contrast, the nHSIC models in this appendix should be interpreted as ablations of the MLCI framework rather than as external competing baselines, since they share the same goal of learning a single scalar diagnosis-code index by maximizing multi-outcome nHSIC.

The default MLCI configuration uses the DeepSets mean⊕max encoder with a single RBF score kernel. This choice is motivated by the structure and intended use of comorbidity scores. Diagnosis codes form unordered sets, so a permutation-invariant DeepSets encoder is appropriate. Mean pooling summarizes distributed comorbidity burden across the diagnosis set, while max pooling can retain strong diagnosis-level feature activations, including signals from rare or high-severity codes. Combining the two gives a compact representation that captures both broad disease burden and salient severe-code signals, while preserving the simplicity and single-score form expected of a comorbidity index.

Multi-RBF kernels and higher-capacity encoder variants can improve selected dependence metrics, showing that the nHSIC framework is extensible. However, these variants introduce additional architectural and kernel-design choices. We therefore use the DeepSets mean⊕max encoder with a single RBF kernel as the default MLCI configuration, and report the remaining nHSIC variants here as architecture and kernel ablations.

All models reported in this section are single-index models: each maps diagnosis-code inputs to one scalar score, and we evaluate the dependence between that scalar score and each clinical outcome. Specifically, we report mutual information (MI) between each model's scalar score and each outcome, as well as distance correlation (dCorr), which captures both linear and nonlinear dependence. All results are computed on the same evaluation splits. Per-outcome MI and dCorr values are computed using the corresponding outcome's valid-label mask.

*Table 6.* **Distance Correlation between each model's unified risk score and clinical outcomes.** Higher is better. Bolded values indicate the best performer per outcome; underlined values indicate the second-best performer. *For readability, each column is scaled by the power of 10 shown in the header; divide by that factor to recover the original dCorr.*

| Model | MIMIC-IV | | | | MIMIC-III | | | |
|---|---|---|---|---|---|---|---|---|
| | **MORT** ($\times 10^2$) | **30M** ($\times 10^2$) | **LOS** ($\times 10^2$) | **ICU** ($\times 10^2$) | **MORT** ($\times 10^2$) | **30M** ($\times 10^2$) | **LOS** ($\times 10^2$) | **ICU** ($\times 10^2$) |
| *Traditional Clinical Indices* | | | | | | | | |
| Charlson (CCI) | 12.59 | 18.44 | 24.55 | 16.09 | 15.54 | 20.26 | 15.28 | 13.07 |
| Elixhauser (ECI) | 19.98 | 23.87 | 33.15 | 25.98 | 21.38 | 24.70 | 23.10 | 13.38 |
| CCI+ECI (scalar) | 17.46 | 22.67 | 30.89 | 22.44 | 19.79 | 24.06 | 20.82 | 14.15 |
| *Classical Machine Learning Baselines* | | | | | | | | |
| kNN | 22.40 | 23.98 | 30.60 | 34.79 | 22.56 | 23.22 | 23.55 | 11.15 |
| Multinomial Naive Bayes | 32.09 | 32.02 | 35.78 | 45.69 | 24.89 | 25.21 | 25.04 | 16.73 |
| Complement Naive Bayes | 32.08 | 32.05 | 35.85 | 45.58 | 24.69 | 25.02 | 23.89 | 16.76 |
| FusedLogits LR | 34.99 | 34.92 | 51.02 | 57.23 | 31.19 | 31.20 | 46.05 | 18.96 |
| Factorization Machine (FM score) | 36.41 | 35.82 | 50.80 | 56.77 | 31.96 | 31.28 | 45.38 | 20.51 |
| Gradient Boosted Trees (Score/Logit) | 34.54 | 34.58 | 51.00 | 55.61 | 32.28 | 31.88 | 48.16 | 20.17 |
| *Deep Learning: BCE-Trained Baselines* | | | | | | | | |
| Deep and Cross Network (DCN) | $28.44 \pm 0.44$ | $28.73 \pm 0.57$ | $48.11 \pm 0.41$ | $51.97 \pm 0.43$ | $30.06 \pm 0.62$ | $29.24 \pm 0.48$ | $47.12 \pm 0.23$ | $19.70 \pm 0.81$ |
| Deep MLP Bag-of-Codes (EmbeddingBag) | $28.88 \pm 0.21$ | $29.13 \pm 0.18$ | $48.22 \pm 0.29$ | $52.04 \pm 0.18$ | $30.17 \pm 0.98$ | $29.40 \pm 0.82$ | $47.67 \pm 0.72$ | $20.18 \pm 0.44$ |
| Star-GAT | $26.97 \pm 1.27$ | $27.48 \pm 1.23$ | $47.19 \pm 0.99$ | $50.24 \pm 0.68$ | $29.44 \pm 0.97$ | $29.01 \pm 0.91$ | $48.25 \pm 0.73$ | $20.06 \pm 0.76$ |
| Pure MIL Attention Pooling | $29.25 \pm 0.62$ | $29.53 \pm 0.43$ | $48.37 \pm 0.36$ | $52.20 \pm 0.63$ | $31.11 \pm 0.70$ | $30.40 \pm 0.31$ | $48.11 \pm 0.43$ | $20.41 \pm 0.53$ |
| Set Transformer (single-logit) | $28.50 \pm 0.68$ | $29.02 \pm 0.63$ | $49.12 \pm 0.37$ | $52.04 \pm 0.19$ | $30.29 \pm 0.62$ | $29.92 \pm 0.74$ | $48.08 \pm 0.43$ | $20.51 \pm 1.10$ |
| DeepSets (sum pool) | $27.89 \pm 0.61$ | $28.17 \pm 0.61$ | $47.75 \pm 0.90$ | $51.12 \pm 0.75$ | $30.72 \pm 0.83$ | $29.96 \pm 0.51$ | $49.01 \pm 0.11$ | $20.58 \pm 0.77$ |
| DeepSets (mean pool) | $27.76 \pm 0.76$ | $28.20 \pm 0.67$ | $47.59 \pm 0.69$ | $50.71 \pm 1.19$ | $30.89 \pm 0.49$ | $30.06 \pm 0.41$ | $49.07 \pm 0.11$ | $20.89 \pm 0.75$ |
| DeepSets (attention pool) | $28.08 \pm 0.83$ | $28.17 \pm 0.37$ | $47.69 \pm 0.56$ | $51.20 \pm 1.03$ | $30.19 \pm 0.48$ | $29.45 \pm 0.19$ | $48.72 \pm 0.63$ | $20.61 \pm 0.77$ |
| DeepSets (gated pool) | $27.42 \pm 0.31$ | $27.83 \pm 0.43$ | $46.88 \pm 0.15$ | $50.11 \pm 0.21$ | $28.88 \pm 0.95$ | $28.48 \pm 0.84$ | $49.04 \pm 0.66$ | $20.86 \pm 0.44$ |
| DeepSets (mean $\oplus$ max pool) | $28.51 \pm 0.33$ | $28.84 \pm 0.27$ | $48.60 \pm 0.60$ | $51.92 \pm 0.41$ | $29.29 \pm 0.49$ | $29.11 \pm 0.32$ | $48.88 \pm 0.43$ | **$21.52 \pm 0.26$** |
| **Our Model** | $54.80 \pm 1.40$ | $49.42 \pm 1.34$ | **$51.15 \pm 0.88$** | **$61.97 \pm 0.64$** | $39.06 \pm 3.27$ | $37.39 \pm 2.63$ | **$49.84 \pm 1.38$** | $18.55 \pm 0.19$ |
| *nHSIC Ablations: MLCI Variants* | | | | | | | | |
| DeepSets (sum pool) + Single-RBF nHSIC | $53.18 \pm 3.45$ | $49.02 \pm 3.53$ | $50.06 \pm 1.87$ | $61.58 \pm 0.44$ | $38.39 \pm 0.99$ | $37.28 \pm 0.85$ | $46.97 \pm 0.73$ | $16.77 \pm 0.97$ |
| DeepSets (sum pool) + Multi-RBF nHSIC | $54.45 \pm 2.05$ | $50.35 \pm 1.28$ | $48.23 \pm 0.79$ | $60.90 \pm 0.57$ | $38.40 \pm 1.99$ | $36.42 \pm 1.81$ | $48.70 \pm 1.27$ | $18.63 \pm 0.62$ |
| DeepSets (mean pool) + Single-RBF nHSIC | $53.92 \pm 0.88$ | $49.52 \pm 1.47$ | $49.33 \pm 1.17$ | $61.73 \pm 0.97$ | $37.61 \pm 0.79$ | $35.94 \pm 0.69$ | $49.25 \pm 0.62$ | $18.84 \pm 0.71$ |
| DeepSets (mean pool) + Multi-RBF nHSIC | $55.97 \pm 3.42$ | $51.75 \pm 2.54$ | $47.91 \pm 0.55$ | $60.36 \pm 0.93$ | $38.59 \pm 0.91$ | $36.67 \pm 1.02$ | $48.62 \pm 0.91$ | $18.70 \pm 0.26$ |
| DeepSets (attention pool) + Single-RBF nHSIC | $55.05 \pm 1.40$ | $50.52 \pm 2.10$ | $48.66 \pm 1.47$ | $61.41 \pm 1.75$ | $36.07 \pm 1.17$ | $34.70 \pm 0.84$ | $49.74 \pm 0.46$ | $19.38 \pm 0.53$ |
| DeepSets (attention pool) + Multi-RBF nHSIC | $54.89 \pm 1.42$ | $51.13 \pm 1.17$ | $47.98 \pm 0.41$ | $60.68 \pm 0.11$ | $39.47 \pm 1.77$ | $37.35 \pm 1.85$ | $47.90 \pm 1.09$ | $18.11 \pm 1.21$ |
| DeepSets (gated pool) + Single-RBF nHSIC | $45.62 \pm 17.28$ | $42.10 \pm 16.21$ | $39.64 \pm 13.97$ | $50.80 \pm 18.35$ | $19.16 \pm 18.20$ | $18.75 \pm 16.19$ | $27.89 \pm 18.74$ | $11.44 \pm 6.81$ |
| DeepSets (gated pool) + Multi-RBF nHSIC | $55.78 \pm 1.09$ | $51.53 \pm 0.50$ | $47.66 \pm 0.44$ | $60.91 \pm 0.53$ | $29.02 \pm 15.82$ | $27.91 \pm 14.09$ | $37.70 \pm 19.85$ | $14.11 \pm 8.48$ |
| Set Transformer + Single-RBF nHSIC | $53.42 \pm 1.30$ | $48.21 \pm 1.40$ | $50.08 \pm 0.61$ | $61.49 \pm 0.53$ | $42.14 \pm 3.36$ | $39.25 \pm 2.50$ | $45.54 \pm 1.88$ | $17.07 \pm 1.30$ |
| Set Transformer + Multi-RBF nHSIC | **$60.73 \pm 0.99$** | **$56.09 \pm 0.92$** | $44.74 \pm 0.22$ | $58.46 \pm 0.85$ | $52.45 \pm 0.83$ | $47.89 \pm 0.92$ | $31.57 \pm 2.44$ | $8.92 \pm 1.20$ |

*Table 7.* **Mutual Information between each model's unified risk score and clinical outcomes.** Higher is better. Bolded values highlight the best performer per outcome; underlined values highlight the second-best performer. *For readability, each column is scaled by the power of 10 shown in the header; divide by that factor to recover the original MI.*

| Model | MIMIC-IV | | | | MIMIC-III | | | |
|---|---|---|---|---|---|---|---|---|
| | **MORT** ($\times 10^3$) | **30M** ($\times 10^3$) | **LOS** ($\times 10^2$) | **ICU** ($\times 10^2$) | **MORT** ($\times 10^3$) | **30M** ($\times 10^3$) | **LOS** ($\times 10^2$) | **ICU** ($\times 10^3$) |
| *Traditional Clinical Indices* | | | | | | | | |
| Charlson (CCI) | 9.64 | 18.99 | 3.41 | 1.99 | 16.35 | 26.32 | 0.94 | 14.60 |
| Elixhauser (ECI) | 20.17 | 28.73 | 6.26 | 3.85 | 26.51 | 30.97 | 3.14 | 14.72 |
| CCI+ECI (scalar) | 19.94 | 28.66 | 6.39 | 4.86 | 30.99 | 36.11 | 3.40 | 16.52 |
| *Classical Machine Learning Baselines* | | | | | | | | |
| kNN | 57.70 | 62.71 | 8.43 | 13.06 | 68.28 | 68.07 | 8.55 | 20.65 |
| Multinomial Naive Bayes | 56.44 | 57.74 | 7.90 | 13.37 | 57.28 | 54.45 | 7.27 | 22.29 |
| Complement Naive Bayes | 56.68 | 56.99 | 7.75 | 13.26 | 57.75 | 55.43 | 7.12 | 27.53 |
| FusedLogits LR | 53.55 | 54.33 | 13.75 | 17.60 | 56.87 | 53.66 | 13.25 | 28.19 |
| Factorization Machine (FM score) | 54.50 | 54.27 | 13.88 | 16.99 | 62.97 | 57.98 | 13.62 | 27.58 |
| Gradient Boosted Trees (Score/Logit) | 64.30 | 63.30 | 14.86 | 17.67 | 65.24 | 66.81 | 14.76 | 24.55 |
| *Deep Learning: BCE-Trained Baselines* | | | | | | | | |
| Deep and Cross Network (DCN) | $65.33 \pm 0.62$ | $61.70 \pm 0.24$ | $14.48 \pm 0.15$ | $18.33 \pm 0.33$ | $65.46 \pm 2.76$ | $59.57 \pm 1.60$ | $13.47 \pm 0.48$ | $27.20 \pm 0.64$ |
| Deep MLP Bag-of-Codes (EmbeddingBag) | $65.88 \pm 0.54$ | $63.63 \pm 1.19$ | $14.73 \pm 0.12$ | $18.45 \pm 0.06$ | $66.20 \pm 4.99$ | $61.39 \pm 3.21$ | $13.83 \pm 0.23$ | $29.89 \pm 2.13$ |
| Star-GAT | $67.38 \pm 2.48$ | $65.32 \pm 2.87$ | $14.89 \pm 0.27$ | $18.89 \pm 0.47$ | $70.31 \pm 2.27$ | $63.74 \pm 2.44$ | $14.10 \pm 0.54$ | $29.28 \pm 2.68$ |
| Pure MIL Attention Pooling | $67.89 \pm 0.94$ | $65.16 \pm 2.32$ | $14.93 \pm 0.04$ | $18.71 \pm 0.12$ | $71.72 \pm 1.57$ | $64.64 \pm 0.79$ | $14.32 \pm 0.37$ | $26.14 \pm 2.26$ |
| Set Transformer (single-logit) | $68.35 \pm 2.40$ | $65.84 \pm 3.19$ | $15.16 \pm 0.18$ | $18.59 \pm 0.25$ | $75.15 \pm 1.62$ | $69.23 \pm 1.95$ | $14.54 \pm 0.31$ | $29.64 \pm 3.14$ |
| DeepSets (sum pool) | $68.99 \pm 1.42$ | $66.21 \pm 2.18$ | $14.97 \pm 0.27$ | $19.17 \pm 0.27$ | $72.48 \pm 1.43$ | $66.54 \pm 1.71$ | $14.78 \pm 0.16$ | $30.46 \pm 3.08$ |
| DeepSets (mean pool) | $69.65 \pm 1.13$ | $66.72 \pm 1.78$ | $14.99 \pm 0.16$ | $19.04 \pm 0.11$ | $72.11 \pm 5.37$ | $65.13 \pm 4.05$ | $14.76 \pm 0.19$ | $29.39 \pm 4.24$ |
| DeepSets (attention pool) | $67.94 \pm 1.84$ | $63.65 \pm 2.40$ | $15.09 \pm 0.02$ | $19.16 \pm 0.28$ | $73.59 \pm 4.11$ | $65.15 \pm 5.87$ | $14.90 \pm 0.40$ | $26.39 \pm 3.32$ |
| DeepSets (gated pool) | $69.11 \pm 1.93$ | $66.33 \pm 3.40$ | $14.93 \pm 0.06$ | $19.01 \pm 0.04$ | $70.31 \pm 3.43$ | $61.88 \pm 1.45$ | $15.20 \pm 0.47$ | $28.05 \pm 4.19$ |
| DeepSets (mean $\oplus$ max pool) | $69.92 \pm 1.26$ | $67.22 \pm 1.60$ | $15.36 \pm 0.11$ | **$19.24 \pm 0.19$** | $73.69 \pm 3.29$ | $68.08 \pm 1.88$ | $15.17 \pm 0.30$ | **$32.42 \pm 4.22$** |
| **Our Model** | $74.22 \pm 0.24$ | $72.19 \pm 0.93$ | **$15.42 \pm 0.18$** | $19.01 \pm 0.29$ | $84.52 \pm 10.89$ | $77.77 \pm 7.53$ | **$16.45 \pm 0.92$** | $26.42 \pm 3.77$ |
| *nHSIC Ablations: MLCI Variants* | | | | | | | | |
| DeepSets (sum pool) + Single-RBF nHSIC | $75.44 \pm 2.00$ | $74.59 \pm 3.68$ | $14.62 \pm 0.70$ | $18.89 \pm 0.23$ | $80.98 \pm 2.45$ | $75.58 \pm 3.68$ | $14.76 \pm 0.47$ | $20.44 \pm 1.33$ |
| DeepSets (sum pool) + Multi-RBF nHSIC | $75.86 \pm 0.89$ | $75.66 \pm 1.56$ | $13.67 \pm 0.59$ | $18.65 \pm 0.10$ | $84.23 \pm 7.70$ | $75.74 \pm 7.61$ | $15.43 \pm 0.70$ | $21.54 \pm 2.80$ |
| DeepSets (mean pool) + Single-RBF nHSIC | $76.25 \pm 1.45$ | $74.95 \pm 2.29$ | $14.21 \pm 0.60$ | $18.86 \pm 0.38$ | $80.92 \pm 4.54$ | $73.46 \pm 2.30$ | $16.03 \pm 0.20$ | $24.65 \pm 1.28$ |
| DeepSets (mean pool) + Multi-RBF nHSIC | $76.88 \pm 0.89$ | $77.05 \pm 1.20$ | $13.80 \pm 0.50$ | $18.56 \pm 0.21$ | $84.41 \pm 3.50$ | $74.55 \pm 3.90$ | $15.60 \pm 0.73$ | $25.19 \pm 2.17$ |
| DeepSets (attention pool) + Single-RBF nHSIC | $76.23 \pm 1.48$ | $75.39 \pm 4.17$ | $13.99 \pm 0.80$ | $19.03 \pm 0.73$ | $75.86 \pm 4.47$ | $70.50 \pm 3.39$ | $15.70 \pm 0.16$ | $27.22 \pm 3.30$ |
| DeepSets (attention pool) + Multi-RBF nHSIC | $76.37 \pm 0.89$ | $76.85 \pm 1.68$ | $13.68 \pm 0.20$ | $18.56 \pm 0.13$ | $88.18 \pm 6.04$ | $78.70 \pm 3.55$ | $15.48 \pm 0.53$ | $27.13 \pm 2.77$ |
| DeepSets (gated pool) + Single-RBF nHSIC | $76.29 \pm 1.63$ | $75.53 \pm 2.47$ | $13.51 \pm 0.39$ | $18.72 \pm 0.34$ | $64.77 \pm 49.98$ | $56.15 \pm 43.45$ | $11.93 \pm 6.29$ | $18.78 \pm 8.69$ |
| DeepSets (gated pool) + Multi-RBF nHSIC | **$77.33 \pm 0.87$** | $77.25 \pm 2.75$ | $13.52 \pm 0.29$ | $18.87 \pm 0.43$ | $59.29 \pm 36.21$ | $54.75 \pm 30.96$ | $12.65 \pm 5.85$ | $19.61 \pm 11.27$ |
| Set Transformer + Single-RBF nHSIC | $72.50 \pm 1.32$ | $70.65 \pm 3.23$ | $14.64 \pm 0.37$ | $18.82 \pm 0.31$ | $86.89 \pm 5.70$ | $76.49 \pm 5.12$ | $15.00 \pm 0.06$ | $24.36 \pm 4.36$ |
| Set Transformer + Multi-RBF nHSIC | $76.71 \pm 0.99$ | **$78.50 \pm 3.29$** | $12.69 \pm 0.35$ | $18.22 \pm 0.63$ | **$114.12 \pm 4.11$** | **$98.58 \pm 3.40$** | $10.63 \pm 1.77$ | $13.22 \pm 1.96$ |

# D. Comorbidity indices and scoring details

### D.1. Input data and coding algorithms

Let $i$ index admissions. Each admission has a set or sequence of diagnosis codes

$$X_i = (d_{i1}, \ldots, d_{im_i}),$$

where $d_{ij}$ is an ICD code and $m_i$ is the number of recorded diagnosis codes for admission $i$. Unless otherwise noted, we use diagnosis codes recorded for the hospital admission to summarize admission-level comorbidity burden; these administrative codes may include diagnoses finalized after discharge and should not be interpreted as strictly present-on-admission measurements. CCI and Elixhauser comorbidities were assigned using Quan et al.'s coding algorithms (Quan et al., 2005): ICD-10 diagnoses used Quan's ICD-10 code lists, and ICD-9-CM diagnoses used the corresponding enhanced ICD-9-CM mappings.

### D.2. Charlson Comorbidity Index (CCI)

The Charlson Comorbidity Index (CCI) is a weighted summary of chronic disease burden, with higher scores indicating greater comorbidity and mortality risk. Following standard implementations, including Quan's mapping, ICD codes are first mapped into Charlson comorbidity categories, and each category contributes at most once.

Let $k \in \{1, \ldots, K\}$ index Charlson comorbidity categories, and define the category indicator

$$c_{ik}^{\mathrm{CCI}} = \mathbb{I}\{\text{any ICD code in } X_i \text{ maps to Charlson category } k\}.$$

Let $a_k \in \{1, 2, 3, 6\}$ denote the Charlson weight for category $k$. Categories not present contribute 0. The CCI for admission $i$ is

$$\mathrm{CCI}_i = \sum_{k=1}^{K} a_k \, c_{ik}^{\mathrm{CCI}}. \tag{6}$$

### D.3. Elixhauser comorbidities and the van Walraven weighted summary (ECI)

Elixhauser et al. (1998) represent comorbidity burden as $Q$ binary indicators included separately as covariates, commonly $Q = 30$. Let $q \in \{1, \ldots, Q\}$ index Elixhauser comorbidity groups, and define

$$e_{iq}^{\mathrm{ECI}} = \mathbb{I}\{\text{any ICD code in } X_i \text{ maps to Elixhauser group } q\}.$$

The original Elixhauser approach defines no single summary score; rather, the $Q$ indicators are entered separately.

Van Walraven et al. summarize the $Q$ Elixhauser indicators into a single weighted score by converting mortality-model coefficients into integer weights. Let $w_q$ denote the van Walraven weight for group $q$. The van Walraven weighted Elixhauser score is

$$\mathrm{ECI}_{\mathrm{VW},i} = \sum_{q=1}^{Q} w_q \, e_{iq}^{\mathrm{ECI}}. \tag{7}$$

Across this paper, the term **Elixhauser Comorbidity Index (ECI)** refers to the van Walraven weighted summary in (7).

### D.4. Common input–output formulation

To make the connection between classic indices and our approach explicit, we describe CCI and ECI through a common input–output lens. Both CCI and the van Walraven ECI map the admission-level diagnosis history $X_i$, restricted to ICD codes, to a scalar score $c_i \in \mathbb{R}$ by (i) forming comorbidity-category indicators from $X_i$ and (ii) summing them with fixed weights to summarize baseline illness burden.

# E. Risk Curves for the Learned Severity Score

This appendix visualizes how the learned scalar score $s_i = s_\theta(X_i)$ relates to empirical outcome risk across the four evaluation tasks: in-hospital mortality, 30-day mortality, length of stay $> 7$ days, and ICU transfer. Because our theory is ordering-based, we focus on the mapping from the learned score ordering to empirical outcome rates. In particular, for each task $t$, we plot estimates of

$$p_t(s) \approx \Pr\{y_i^{(t)} = 1 \mid s_i = s\}$$

as a function of the learned score.

**Estimators.** We report two complementary curve estimators: (i) an **unconstrained binned** estimate, which partitions test scores into quantile bins and plots the empirical event rate per bin; and (ii) a **monotone isotonic** estimate, which fits a nondecreasing function of the score using isotonic regression on valid labels for the given task. The binned curves reveal local non-monotonicities due to finite-sample noise and label sparsity, while the isotonic curves summarize the best monotone risk calibration implied by the learned ordering.

**Validity masking.** For each task $t$, curves are computed over the subset of test admissions with valid labels, $\{i : M_i^{(t)} = 1\}$. For length of stay, validity is defined by the dedicated indicator column `long_stay_defined`. This matches the evaluation protocol used in the main paper.

**Summary statistics.** Table 8 summarizes whether risk generally increases with the learned score for each outcome and reports the end-to-end change in the monotone fit.

| Dataset | MORT | 30M | LOS | ICU |
|---|---|---|---|---|
| **MIMIC-IV** | inc. ($\Delta = 0.053 \pm 0.009$) | inc. ($\Delta = 0.092 \pm 0.022$) | inc. ($\Delta = 0.712 \pm 0.003$) | inc. ($\Delta = 0.855 \pm 0.000$) |
| **MIMIC-III** | inc. ($\Delta = 0.399 \pm 0.099$) | inc. ($\Delta = 0.381 \pm 0.069$) | inc. ($\Delta = 0.751 \pm 0.023$) | inc. ($\Delta = 0.289 \pm 0.028$) |

*Table 8.* Risk-curve summary from isotonic fits of $\Pr\{y_i^{(t)} = 1 \mid s_i = s\}$ as a function of the learned score. We report a qualitative monotonicity label (inc./weakly inc.) and the end-to-end change $\Delta := p_{95} - p_5$, where $p_q$ denotes the $q$-th percentile of the fitted risk values along the score axis. For both datasets, $\Delta$ is reported as mean $\pm$ standard deviation over seeds $\{11, 101, 1001\}$.

**Figures.** Figures 2–5 report per-task risk curves for MIMIC-IV and MIMIC-III. For each dataset, we show (i) **unconstrained binned** empirical curves using quantile bins, to visualize sampling noise and possible local non-monotonicities, and (ii) **isotonic** fits, to estimate the best nondecreasing risk mapping implied by the learned ordering. The plotted curves correspond to seed 11; Table 8 reports the corresponding $\Delta$ summary across seeds $\{11, 101, 1001\}$.

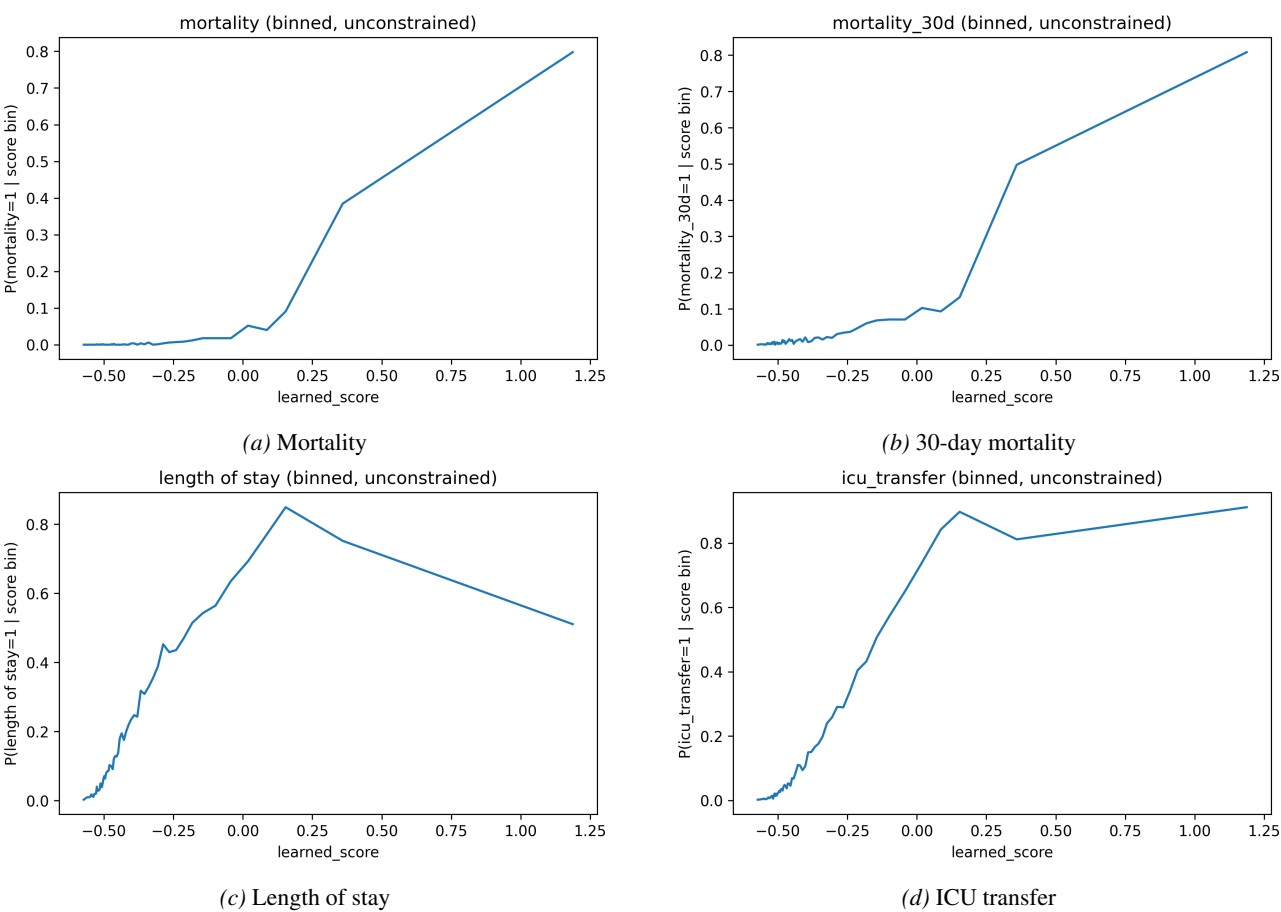

*(a)* Mortality

*(b)* 30-day mortality

*(c)* Length of stay

*(d)* ICU transfer

*Figure 2.* MIMIC-IV: binned (unconstrained) risk curves $\Pr\{y_i^{(t)} = 1 \mid s_i = s\}$ as a function of the learned score (60 quantile bins).

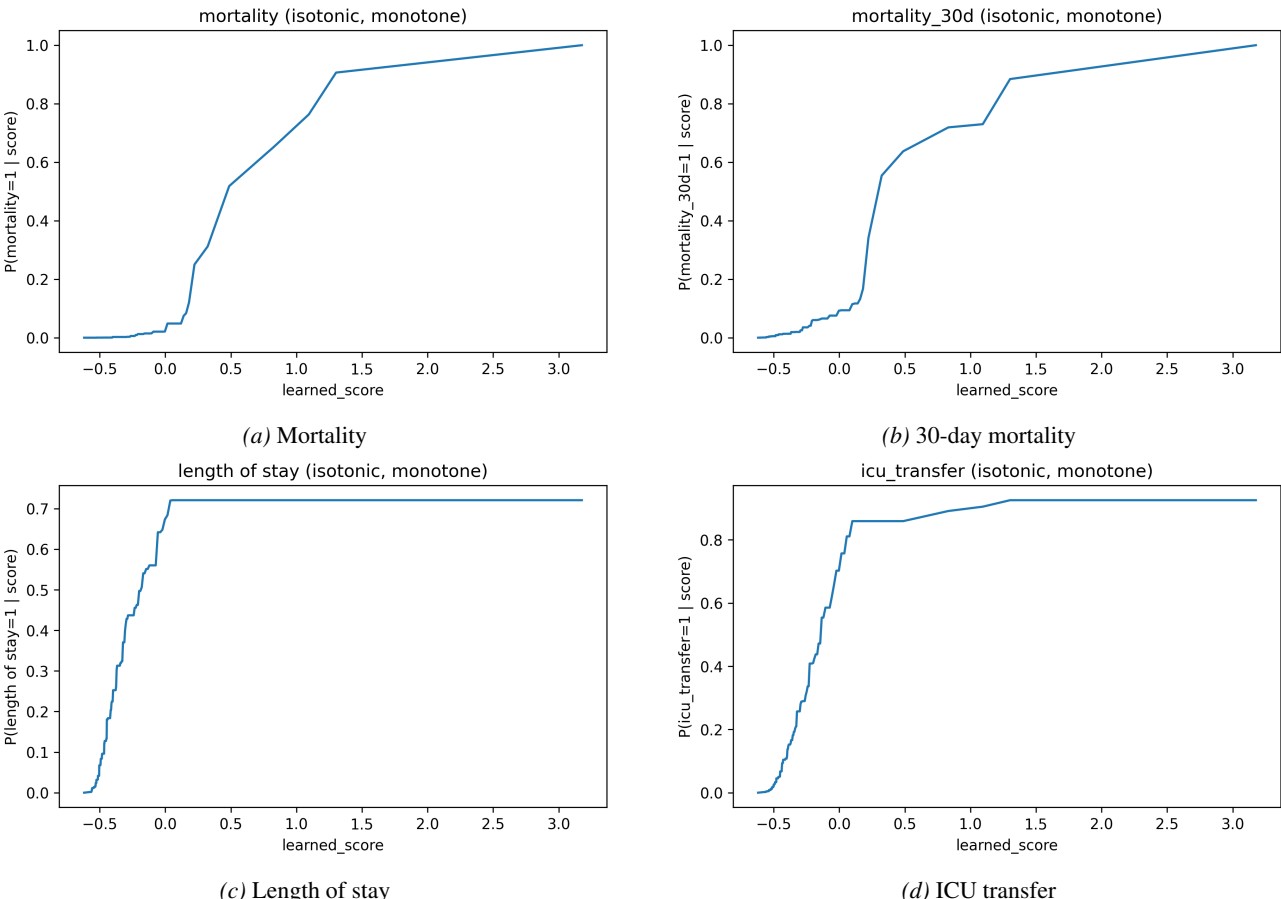

*(a)* Mortality

*(b)* 30-day mortality

*(c)* Length of stay

*(d)* ICU transfer

*Figure 3.* MIMIC-IV: isotonic (monotone) risk curves $\Pr\{y_i^{(t)} = 1 \mid s_i = s\}$ fit via isotonic regression on valid labels for each task.

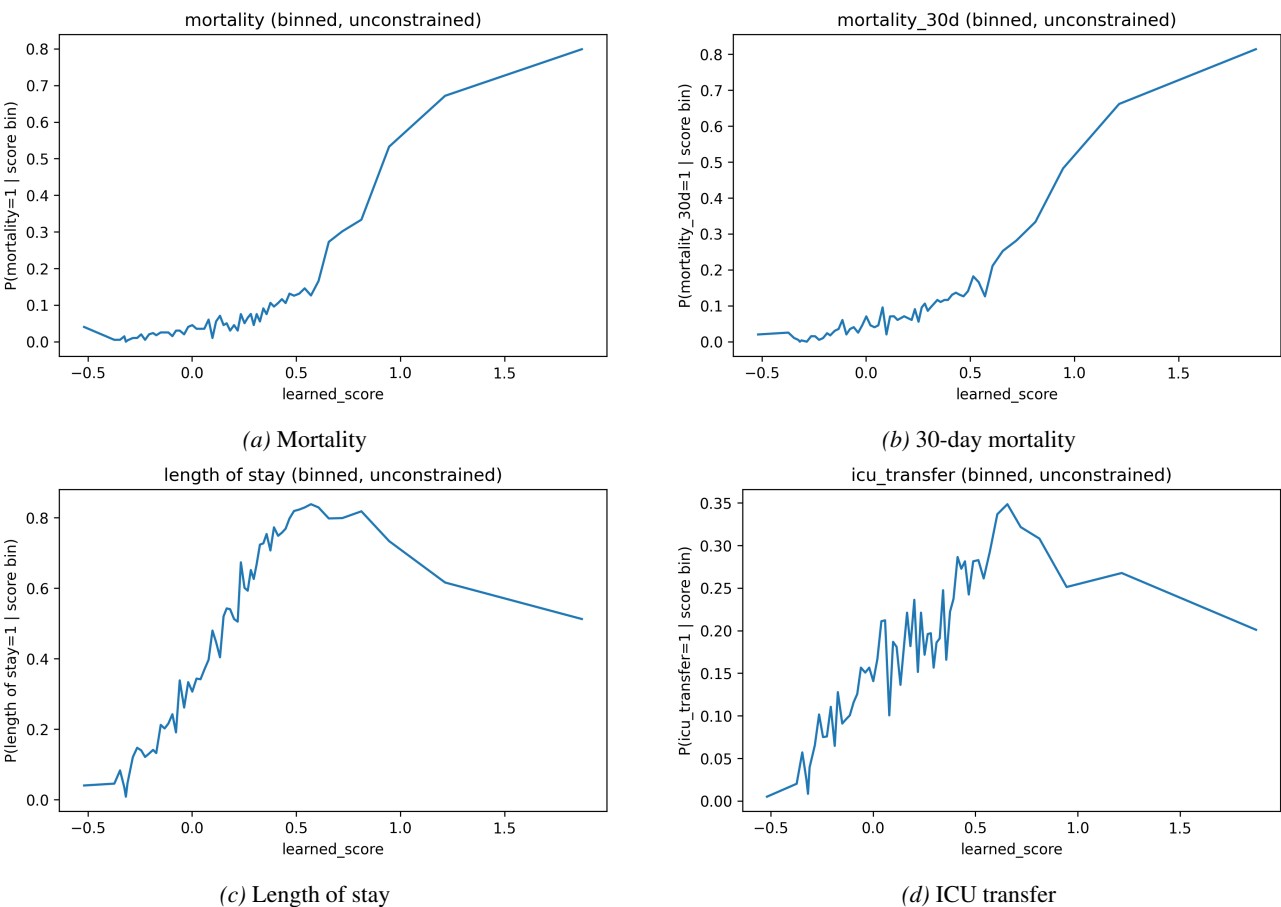

*(a)* Mortality

*(b)* 30-day mortality

*(c)* Length of stay

*(d)* ICU transfer

*Figure 4.* MIMIC-III: binned (unconstrained) risk curves $\Pr\{y_i^{(t)} = 1 \mid s_i = s\}$ as a function of the learned score (60 quantile bins).

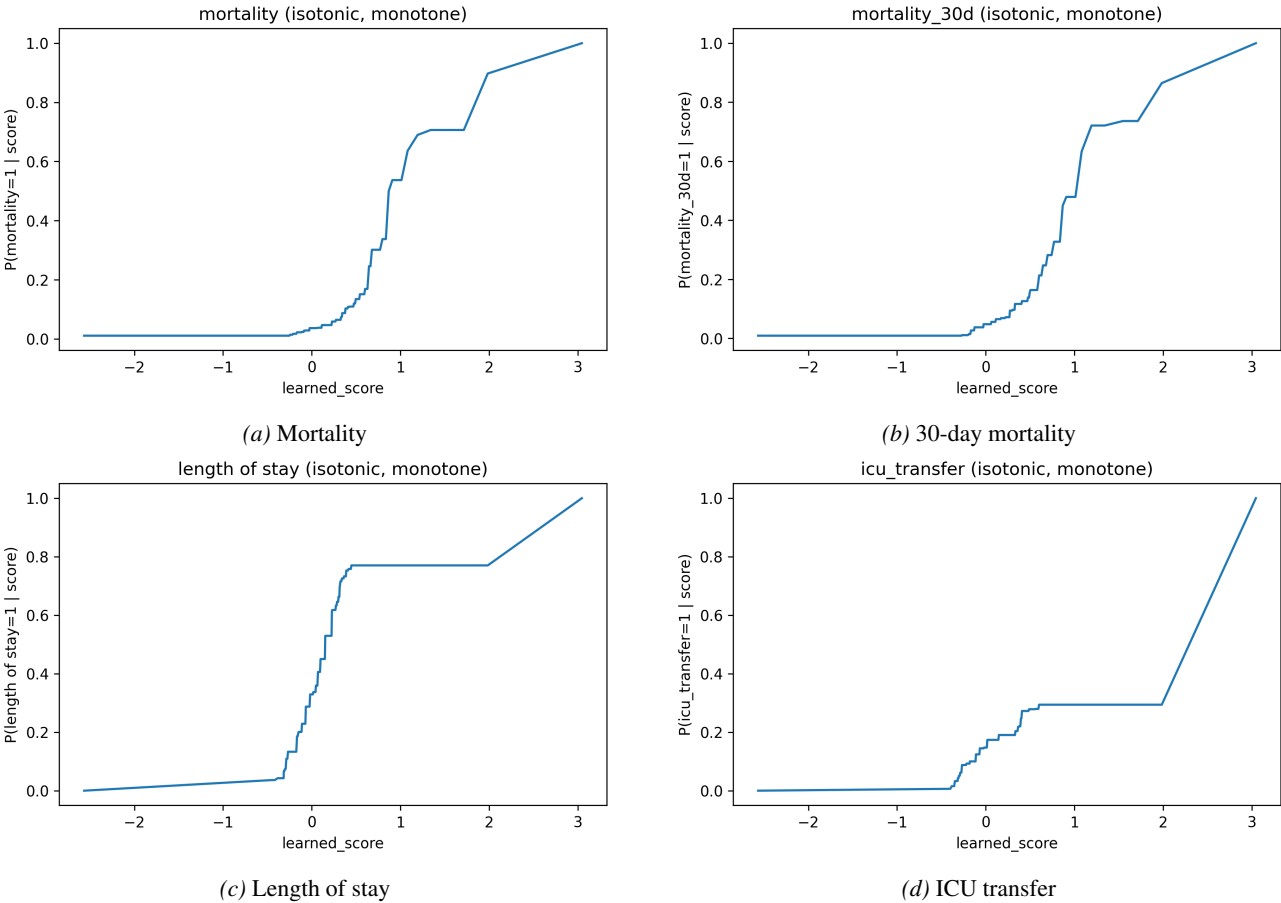

*(a)* Mortality

*(b)* 30-day mortality

*(c)* Length of stay

*(d)* ICU transfer

*Figure 5.* MIMIC-III: isotonic (monotone) risk curves $\Pr\{y_i^{(t)} = 1 \mid s_i = s\}$ fit via isotonic regression on valid labels for each task.

