# OpenReview forum: "A Machine-Learned Comorbidity Index"
_ICML.cc/2026/Conference — ICML 2026 regular_

### Official Review · Reviewer_vXge · 2026-03-09

**Soundness:** 3
**Presentation:** 4
**Significance:** 3
**Originality:** 2
**Overall Recommendation:** 4
**Confidence:** 4

**Summary:**

This paper presents a new scaler comorbidity index and assesses its usefulness in a few downstream clinical tasks.

**Compliance With Llm Reviewing Policy:**

Affirmed.

**Key Questions For Authors:**

n/a

**Limitations:**

Limitations are not discussed.

**Strengths And Weaknesses:**

Strengths
- The paper is organized well, and the ideas are presented clearly.
- The experiments on the two targeted datasets are fairly extensive, as well as the breadth of the chosen baselines seems broad.
- The theoritical part looks solid (perhaps not a major contrtribtion).

Weaknesses
- Considering rather strong claims that are made in the paper, limiting experiments to only MIMIC-type datasets seems concerning.
- The paper does not seem to refer to the interpretability aspect of the measure. Yes, the proposed measure may be better, but it is missing some key interpretable elements compared to established methods like Charlston.

---

> ### Author Rebuttal · Authors · 2026-03-30
>
> We thank the reviewer for their comments.
>
> # W1: Generalizability beyond ICU settings: EHRSHOT Dataset
>
>  To address generalizability beyond MIMIC, we added a new external evaluation on an **EHRSHOT-derived** cohort. Unlike ICU-centered MIMIC, EHRSHOT is built from **longitudinal structured EHR data** in the **OMOP common data model** and reflects a broader visit-level prediction setting. We therefore use 30-day readmission rather than ICU transfer to better match this non-ICU setting.
>
> ## Distance correlation on EHRSHOT
>
> *(higher is better; columns scaled by $10^{1}$)*
>
> | Model            |                  MORT |                   30M |                   LOS |               READMIT |
> | ---------------- | --------------------: | --------------------: | --------------------: | --------------------: |
> | CCI              |                 0.941 |                 1.280 |                 1.770 |                 1.078 |
> | ECI              |                 1.392 |                 1.348 |                 2.634 |                 1.445 |
> | CCI+ECI          |                 1.231 |                 1.405 |                 2.279 |                 1.298 |
> | kNN              |                 1.403 |                 1.502 |                 2.720 |                 1.203 |
> | MNB              |                 0.354 |                 0.438 |                 0.269 |                 1.413 |
> | FusedLogits LR   |                 1.255 |                 1.143 |                 1.353 |                 1.983 |
> | FM               |                 1.358 |                 1.203 |                 1.340 |                 2.789 |
> | GBT              |                 1.534 |                 1.118 |                 1.913 |                 3.039 |
> | DCN              |     1.087 $\pm$ 0.022 |     1.153 $\pm$ 0.013 |     1.100 $\pm$ 0.100 |     1.924 $\pm$ 0.222 |
> | EmbeddingBag MLP |     1.250 $\pm$ 0.050 |     1.091 $\pm$ 0.065 |     1.265 $\pm$ 0.104 |     1.762 $\pm$ 0.203 |
> | Star-GAT         |     1.125 $\pm$ 0.145 |     0.998 $\pm$ 0.174 |     1.231 $\pm$ 0.126 |     1.991 $\pm$ 0.083 |
> | MIL Attention    |     1.278 $\pm$ 0.041 |     1.225 $\pm$ 0.018 |     1.153 $\pm$ 0.028 |     2.074 $\pm$ 0.138 |
> | Set Transformer  |     1.213 $\pm$ 0.116 |     1.160 $\pm$ 0.104 |     0.755 $\pm$ 0.087 |     2.360 $\pm$ 0.291 |
> | DeepSets         |     1.684 $\pm$ 0.101 |     1.487 $\pm$ 0.094 |     1.664 $\pm$ 0.225 |     3.001 $\pm$ 0.039 |
> | **Our Model**    | **3.083 $\pm$ 0.259** | **2.104 $\pm$ 0.158** | **5.657 $\pm$ 0.117** | **3.367 $\pm$ 0.272** |
>
> ## Mutual information on EHRSHOT
>
> *(higher is better; columns scaled by $10^{2}$)*
>
> | Model            |                  MORT |                   30M |                    LOS |               READMIT |
> | ---------------- | --------------------: | --------------------: | ---------------------: | --------------------: |
> | CCI              |                 0.342 |                 0.698 |                  1.408 |                 0.763 |
> | ECI              |                 0.635 |                 0.710 |                  3.762 |                 2.747 |
> | CCI+ECI          |                 0.627 |                 1.132 |                  4.585 |                 3.631 |
> | kNN              |                 1.865 |                 2.100 |                  7.081 |                 5.462 |
> | MNB              |                 0.177 |                 0.141 |                  2.218 |                 4.956 |
> | FusedLogits LR   |                 0.965 |                 1.158 |                  3.343 |                 7.136 |
> | FM               |                 1.378 |                 1.549 |                  5.267 |                 7.705 |
> | GBT              |                 2.133 |                 1.362 |                  9.290 |                 8.747 |
> | DCN              |     0.997 $\pm$ 0.149 |     1.275 $\pm$ 0.133 |      3.775 $\pm$ 0.149 |     5.448 $\pm$ 0.112 |
> | EmbeddingBag MLP |     1.146 $\pm$ 0.073 |     1.116 $\pm$ 0.557 |      3.791 $\pm$ 0.588 |     5.515 $\pm$ 0.770 |
> | Star-GAT         |     1.078 $\pm$ 0.123 |     1.027 $\pm$ 0.012 |      3.844 $\pm$ 0.370 |     5.622 $\pm$ 0.577 |
> | MIL Attention    |     1.312 $\pm$ 0.114 |     1.397 $\pm$ 0.317 |      3.046 $\pm$ 0.293 |     5.465 $\pm$ 1.014 |
> | Set Transformer  |     1.139 $\pm$ 0.299 |     1.146 $\pm$ 0.466 |      3.565 $\pm$ 0.225 |     6.220 $\pm$ 0.434 |
> | DeepSets         |     2.481 $\pm$ 0.168 | **1.956 $\pm$ 0.222** |      5.540 $\pm$ 0.118 |     8.416 $\pm$ 0.193 |
> | **Our Model**    | **3.051 $\pm$ 0.131** |   _1.791 $\pm$ 0.285_ | **16.200 $\pm$ 0.393** | **9.965 $\pm$ 0.289** |
>
> **Main result.** On EHRSHOT, MLCI remains strongest in this non-ICU setting—best on all four outcomes by distance correlation and on three of four by mutual information.
>
> ## W2: Interpretability Concerns
>
> In response to this question, we have conducted additional interpretability analysis. Please see our response to W1 for reviewer 88QL.

---

> > ### Author Rebuttal · Reviewer_vXge · 2026-04-04
> >
> > Appreciate the authors' detailed responses.
> >
> > The response regarding interpretability seems to miss the point that there's almost no way to make such a method truly interpretable, especially compared to the traditional clinical scores.
> >
> > A score of 4 still seems reasonable to me.

---

### Official Review · Reviewer_pN3G · 2026-03-12

**Soundness:** 4
**Presentation:** 4
**Significance:** 4
**Originality:** 3
**Overall Recommendation:** 5
**Confidence:** 4

**Summary:**

This paper tries to learn a single comorbidity score from diagnosis codes in a more data-driven way than classic hand-crafted indices. The experiments use de-identified EHR cohorts from MIMIC-III and MIMIC-IV, and the score is trained against multiple outcomes such as mortality and ICU transfer. Instead of assigning fixed weights to diagnoses by expert rules, the method learns one scalar severity value by maximizing normalized HSIC between the score and several endpoints at the same time. This design keeps the score simple enough for clinical use while still reflecting nonlinear relationships in the data. The main novelty is a multi-outcome, dependence-based objective for learning one unified comorbidity axis.

**Compliance With Llm Reviewing Policy:**

Affirmed.

**Key Questions For Authors:**

1) External validity: results are shown on MIMIC-III/IV; do you have evidence that the learned ordering transfers to a different hospital system or claims data (e.g., eICU/OMOP)?
2) Outcome set/weights: how sensitive is the learned score to the choice of outcomes and the two-stage inverse-strength weighting (α, ε, clipping)? Any guidance for practitioners choosing tasks?
3) Diagnosis representation: you prefix ICD codes (k=4) and treat diagnoses as an unordered set. How do performance and interpretability change with different prefix lengths, using full codes, or incorporating timing/POA flags?
4) Thresholding: the theory motivates a single monotone cutoff for a high-severity group. How stable is the selected cutoff across random seeds/splits and across sub-populations (age/sex/service lines)?
5) Downstream use: beyond dependence (MI/dCorr), does using MLCI as a covariate in standard risk-adjustment models improve downstream calibration/discrimination vs CCI/ECI in a holdout cohort?

**Limitations:**

Yes

**Strengths And Weaknesses:**

This paper learns a single comorbidity/severity score from diagnosis codes, so it can be used like a modern replacement for CCI/ECI. The key idea is to learn the score by maximizing normalized HSIC dependence with multiple outcomes, not just one endpoint. Results on MIMIC-III/IV show stronger association with outcomes than classic scores and several ML baselines.
The experiments are mainly ICU datasets, so it is unclear how well the score transfers to claims data or other hospitals. The diagnosis encoder treats codes mostly as an unordered set, so it ignores timing and disease trajectories. Practical sensitivity (kernel choices, outcome weights, cutoff stability) should be studied more for deployment.

---

> ### Author Rebuttal · Authors · 2026-03-30
>
> We thank the reviewer for this comment and their suggestions.
>
> Q1: In response to this question, we have conducted additional experiments on EHRShot dataset. Please see our response (W1) to reviewer vXge.
>
> *Note: All ablations use MIMIC-IV, our largest cohort.*
>
> # Q2: Outcome-set Ablation
>
> We performed an additional outcome-set ablation by retraining MLCI on leave-one-out subsets of the full four-outcome setup, keeping everything else same. We then compared each variant to the full model and evaluated downstream one-score prediction, testing whether the learned scalar is robust to changes in the training outcome set.
>
> | Variant        | Spearman | Top-5% | Top-10% |
> | -------------- | -------: | -----: | ------: |
> | All 4          |    1.000 |  1.000 |   1.000 |
> | Drop ICU       |    0.866 |  0.801 |   0.776 |
> | Drop long stay |    0.773 |  0.789 |   0.787 |
> | Drop 30d mort. |    0.920 |  0.818 |   0.855 |
>
> Downstream results: https://anonymous.4open.science/r/OutcomeSetAblation-FB18/DownstreamResults.png
>
> **Results.** The score remains robust to outcome ablation: ranking agreement with the full model stays substantial, and downstream performance remains stable.
>
> # Q2: Weighting-hyperparameter Ablation
>
> We performed an additional weighting-hyperparameter ablation to test sensitivity to the two-stage inverse-strength multitask weighting scheme, while keeping the full four-task outcome set fixed. Starting from the default weighting rule
>
> $$
> w_t \propto \left(\frac{\max(\hat h_{\max},\epsilon)}{\max(\hat h_t,\epsilon)}\right)^\alpha,
> $$
>
> followed by clipping and renormalization, we retrained MLCI under several alternative settings.  For each variant, we compared test-set scores to the default-weight model by aligning hadm_ids and assessing ranking stability, then evaluated downstream one-score prediction via logistic regression.
>
> | Variant          | Spearman | Top-5% | Top-10% |
> | ---------------- | -------: | -----: | ------: |
> | Default          |    1.000 |  1.000 |   1.000 |
> | Equal            |    0.907 |  0.834 |   0.860 |
> | $\alpha=0.5$     |    0.917 |  0.836 |   0.874 |
> | $\alpha=1.0$     |    0.917 |  0.803 |   0.842 |
> | $\epsilon=0.005$ |    0.925 |  0.860 |   0.879 |
> | Cap $=5$         |    0.907 |  0.855 |   0.871 |
>
> Downstream resuts: https://anonymous.4open.science/r/WeightAblation-DC65/WeightDownstream.png
>
> **Results.** MLCI is robust to weighting choices: ranking agreement remains strong, and downstream performance stays stable.
>
> Q2: **Advice to practitioners.** MLCI is very useful for applications such as risk adjustment, triage, patient subgrouping, or identifying a high-severity cohort, where a single shared severity score is more practical and interpretable than maintaining separate models for each outcome.
>
> # Q3: Diagnosis-representation Ablation
>
> We ablated diagnosis representation by retraining MLCI with ICD prefixes of length $k=3,4,5$ and full codes, while keeping all other settings fixed. For each variant, we rebuilt the training vocabulary, retokenized admissions, compared it to the default $k=4$ model, evaluated downstream prediction, and recorded vocabulary size.
>
> | Repr. | Spearman | Top-5% | Top-10% |
> | ----- | -------: | -----: | ------: |
> | P3    |    0.858 |  0.837 |   0.847 |
> | P4    |    1.000 |  1.000 |   1.000 |
> | P5    |    0.890 |  0.845 |   0.859 |
> | Full  |    0.883 |  0.848 |   0.860 |
>
> Downstream results: https://anonymous.4open.science/r/PrefixAblation-0502/PrefixAblationDownstreamResults.png
>
> **Results.** MLCI is robust to diagnosis granularity: rankings remain similar, while downstream performance barely changes. Finer-grained codes give only modest gains at much larger vocabulary size (1,678 for P3 vs. 17,637 for full code), so prefix-4 remains a reasonable default.
>
> # Q4: Threshold robustness Ablation
>
> We performed an additional threshold stability analysis by retraining the default MLCI across random seeds, selecting a validation cutoff for the top q\% highest-risk admissions, and applying it to the test set. We then assessed how the cutoff and resulting high-risk group varied across seeds and clinically relevant subgroups, testing whether the monotone threshold is practically stable.
>
> Threshold Ablation Results:  https://anonymous.4open.science/r/ThresholdAblation-E512/ThresholdAblationResults.png
>
> **Results.** Although the raw cutoff varies across seeds (0.063–0.289), the induced high-risk group is highly stable: test selection stays near 9.6%, and age-subgroup patterns remain consistent. Overall, MLCI supports stable *percentile-based* high-risk stratification across retraining.
>
> Q5: In response to this question, we have conducted additional experiments on downstream risk adjustment and prediction tasks. Please see our response (W4) to reviewer gw2i for risk adjustment results and reviewer 88ql for prediction resuts (W2).

---

### Official Review · Reviewer_gw2i · 2026-03-12

**Soundness:** 2
**Presentation:** 3
**Significance:** 2
**Originality:** 2
**Overall Recommendation:** 2
**Confidence:** 3

**Summary:**

This paper proposes a machine-learned comorbidity index (MLCI) that maps diagnosis codes to a single scalar by maximizing the normalized Hilbert-Schmidt Independence Criteria (nHSIC) between the score and multiple clinical outcomes. The authors conducted extensive experiments to evaluate the performance of the proposed MLCI compared with existing comorbidity indices using several benchmark EHRs datasets.

**Compliance With Llm Reviewing Policy:**

Affirmed.

**Final Justification:**

I appreciate the authors' efforts to address my comments. In spite of their best efforts, I still find the research premise of this work to be questionable. The key assumption that that T binary clinical outcomes are coupled through a single unobserved, real-valued latent severity score Z representing overall sickness/disease burden is not supported by the real data results. For example, the first singular direction accounts for less than 50% of the overall objective in several analyses. Based on their real data experiments, a more plausible assumption is that T binary clinical outcomes are coupled through an unobserved low-dimensional vector Z representing overall sickness/disease burden and the direction with the highest variability has some valuable clinical utility. In addition, this paper offers incremental advances in methodological innovation by the ICML standards, particularly given that the major area of this paper is General Machine Learning, not Applications. Lastly, the dependence metrics between the comorbidity score and outcomes are confounded by treatments, making the interpretation of these dependences/associations difficult; the authors' responses regarding this limitation is not convincing.

**Key Questions For Authors:**

In Tables 3 and 4, the two testable predictions are based on the assumption that the task profiles are approximately rank-one aligned. It’s unclear whether this assumption holds for the use case of multi-comorbidities.

**Limitations:**

There is no discussion of the limitations of the proposed method. What happens when the key assumptions underlying the proposed method do not hold? Will the proposed method underperform the competing methods?

**Strengths And Weaknesses:**

Strengths: This work proposes an advanced modeling framework for learning a comorbidity score by maximizing the normalized Hilbert-Schmidt Independence Criteria (nHSIC) between the score and multiple clinical outcomes and provides some theoretical analysis.

Weaknesses:
1. The research premise of this work is questionable. The proposed approach (and associated theoretical results in Section 6) is predicated on the assumption that the T binary clinical outcomes are coupled through a single unobserved, real-valued latent severity score Z representing overall sickness/disease burden. The justification for this assumption is not compelling, as comorbidities have multiple dimensions. The resulting model for the scalar comorbidity score $s_i$, while elegant, is contrived, and so are the theoretical results in Section 6.
2. This reviewer is not convinced that the scalar comorbidity score would be useful in practice. For example, two patients with the same score value could have very different comorbidity profiles, which would require different treatment strategy etc.
3. Methodological innovation is limited.
4. In the numerical experiments, dependence metrics between the comorbidity score and outcomes are confounded by treatments; it is difficult to interpret such dependencies without controlling for treatment. Actually, to truly evaluate the usefulness of the proposed score, one would need to assess the impact of using the score to guide treatment on patient outcomes.
5. In Tables 3 and 4, the two testable predictions are based on the assumption that the task profiles are approximately rank-one aligned. It’s unclear whether this assumption holds for the use case of multi-comorbidities.

---

> ### Author Rebuttal · Authors · 2026-03-30
>
> # W1:
>
> We kindly note that scalar comorbidity scores are already standard in clinical practice. Widely used indices such as Charlson and Elixhauser show that multidimensional comorbidity information can be meaningfully collapsed into a single score [1]. The assumption of a latent severity is clinically motivated as prior latent-variable studies posit an unobserved patient health state associated with adverse outcomes [2]. We do not assume severity fully determines every outcome or that comorbidity is perfectly one dimensional; rather, we assume that greater illness burden generally increases the risk of adverse outcomes, consistent with prior severity literature [3–5]. Under this view, a principled scalar score is one that shows strong overall dependence with adverse hospital outcomes. Our primary contribution is to learn a single principle score directly from the data as opposed to manually handcrafting score like in clinical literature, while keeping everything else exactly the same. The principle contribution of our theory is that if such a latent severity variable exists, we can learn a scalar proxy by maximizing its dependence with the adverse outcomes it influences. We use HSIC because it captures both linear and nonlinear dependence [6].
>
> [1] Why summary comorbidity measures such as the Charlson Comorbidity Index and Elixhauser score work
>
> [2] Estimating patient's health state using latent structure inferred from clinical time series and text
>
> [3] APACHE II
>
> [4] SAPS II
>
> [5] SOFA score
>
> [6] Predicting clinical deterioration with nonlinear severity–outcome relationships
>
> # W2:
> While the reviewer is correct that two patients with same score could have different clinical profiles and treatment plans, scalar comorbidity scores are widely used for many clinical purposes beyond treatment selection, including prognosis [1, 2], patient stratification [3], risk stratification [4], and risk adjustment [5].
>
> [1] Outcome-specific Charlson comorbidity indices for predicting poor inpatient outcomes following noncardiac surgery using hospital administrative data.
>
> [2] The effect of the Charlson comorbidity index on in-hospital complications, hospital length of stay, mortality, and readmissions among patients hospitalized for acute stroke.
>
> [3] Charlson comorbidity index: an additional prognostic parameter for preoperative glioblastoma patient stratification.
>
> [4] Charlson and Elixhauser comorbidity indices for prediction of mortality and hospital readmission in patients with acute pulmonary embolism.
>
> [5] Updating and validating the Charlson comorbidity index and score for risk adjustment in hospital discharge abstracts using data from 6 countries.
>
> Please view W4 below for real-world clinical utility.
>
> # W3:
>  We respectfully disagree that the methodological innovation is limited. This is the first work to formulate comorbidity index construction as a data-driven machine-learning problem, rather than a manually engineered scoring rule, and to provide an accompanying theoretical analysis. Beyond the algorithm, our contribution is to show that a single learned scalar can be both theoretically justified and clinically meaningful as a shared severity axis across multiple outcomes. The novelty lies in combining (i) a data-driven comorbidity index, (ii) dependence-based multi-outcome training, and (iii) a theoritical analysis.
>
> # W4: Risk Adjustment
> We agree that treatment-policy utility would require causal evaluation beyond this work. For its intended use in risk adjustment, severity stratification, and prognosis, we evaluated MLCI in a risk-adjustment setting [1] by adding alternative comorbidity indices to a baseline model and comparing performance. Starting from baseline covariates (age, sex, admission location, insurance), we added diagnosis count, drug count, CCI, ECI, and MLCI, including drug-augmented variants; for each outcome, we fit logistic regression on validation and report test C-statistic and Brier in both the full cohort and the age65plus subgroup for MIMIC III/IV.
>
> Results: https://anonymous.4open.science/r/RiskAdjustmentEvaluation-58C8/RiskAjustmentResults.png
>
> MLCI yields larger mortality gains than CCI/ECI and remains competitive on long stay and ICU transfer.
>
> [1] Comparing comorbidity measures for predicting mortality and hospitalization in three population-based cohorts
>
> ## Q1:
> Our theory studies the approximately rank-one setting, and we test that condition empirically. The diagnostics show that the centered task profiles are not exactly rank-one, but do exhibit meaningful low-rank structure with a leading shared component. We further analyzed the centered task-profile matrix $\widetilde W$ through its singular directions.
>
> https://anonymous.4open.science/r/RankAnalysis--0765/RankAnalysis.png
>
> **Results.** This supports interpreting MLCI as a single scalar score that captures the leading shared severity signal, while additional singular directions reflect residual task-specific structure.

---

> > ### Author Rebuttal · Reviewer_gw2i · 2026-04-04
> >
> > I appreciate the authors' effort to address my comments. But I don't find my concerns fully addressed.
> >
> > W1: I agree that summary comorbidity measures such as the Charlson Comorbidity Index and Elixhauser score are useful in practice, but this is not the same as making an assumption that T binary clinical outcomes are coupled through a single unobserved, real-valued latent severity score Z representing overall sickness/disease burden, which is much stronger than using a summary measure for certain task(s).
> >
> > W3: I still think that this paper offers incremental advances in methodological innovation by the ICML standards, particularly given that the major area of this paper is General Machine Learning, not Applications.
> >
> > W4: This remains a significant limitation.
> >
> > Q1:  The results show that the first singular direction accounts for only 38% of the overall objective in MIMIC-III. I am unclear why this offers strong evidence for approximately rank one.

---

> > > ### Author Response · Authors · 2026-04-04
> > >
> > > # W1
> > > We thank the reviewer for acknowledging the practical use cases of comorbidity scores that we discussed. Regarding the concern that our framework assumes multiple binary clinical outcomes are coupled through a single latent severity score, we would like to clarify that this assumption is clinically motivated by multilevel item response theory (MLIRT) models in medical domains. MLIRT models often adopt a unidimensional assumption, under which multiple outcomes are treated as clinical manifestations of a univariate latent variable, and they have been increasingly used in longitudinal clinical studies with multiple outcomes of mixed types (e.g., continuous, **binary**, and ordinal) [1]. In this sense, MLIRT provides clinical grounding for our modeling of a shared latent severity score across multiple binary clinical outcomes.
> > >
> > > [1] Robust Bayesian hierarchical model using normal/independent distributions
> > >
> > > # W4: Pre-Treatment Baseline Adjustment Analysis
> > >
> > > To address the limitation about confounding by downstream treatment and care processes, we performed an additional pre-treatment baseline adjustment experiment on MIMIC-IV using same patient-disjoint splits and downstream logistic-regression evaluation. This experiment answers **whether, given only information available at admission, MLCI improves risk prediction more than CCI or ECI**. We used only admission-time covariates—such as age and demographic/administrative information available at admission—and excluded downstream treatment and process variables. We then compared baseline-only models against baseline-plus-score models using MLCI, CCI, or ECI, with all models fit on validation.
> > >
> > > | Outcome | Baseline-only AUROC | Baseline + MLCI AUROC (Δ) | Baseline + CCI AUROC (Δ) | Baseline + ECI AUROC (Δ) |
> > > | ------- | ------------------: | ------------------------: | -----------------------: | -----------------------: |
> > > | M       |              0.8275 |  **0.9753** (**+0.1478**) |         0.8539 (+0.0264) |         0.8899 (+0.0623) |
> > > | 30M     |              0.8018 |  **0.9111** (**+0.1094**) |         0.8421 (+0.0404) |         0.8619 (+0.0601) |
> > > | LOS     |              0.7607 |  **0.8483** (**+0.0876**) |         0.7859 (+0.0251) |         0.8113 (+0.0506) |
> > > | ICU     |              0.8051 |  **0.9140** (**+0.1089**) |         0.8099 (+0.0048) |         0.8277 (+0.0226) |
> > >
> > > | Outcome | Baseline-only AUPRC | Baseline + MLCI AUPRC (Δ) | Baseline + CCI AUPRC (Δ) | Baseline + ECI AUPRC (Δ) |
> > > | ------- | ------------------: | ------------------------: | -----------------------: | -----------------------: |
> > > | M       |              0.1625 |  **0.6708** (**+0.5083**) |         0.1690 (+0.0065) |         0.2253 (+0.0628) |
> > > | 30M     |              0.1680 |  **0.5656** (**+0.3976**) |         0.1954 (+0.0273) |         0.2372 (+0.0692) |
> > > | LOS     |              0.4105 |  **0.5475** (**+0.1370**) |         0.4461 (+0.0356) |         0.5129 (+0.1024) |
> > > | ICU     |              0.4702 |  **0.7202** (**+0.2500**) |         0.4768 (+0.0066) |         0.5175 (+0.0474) |
> > >
> > > MLCI adds substantial predictive value beyond admission-time covariates, with larger gains than CCI or ECI, supporting that it captures admission-time prognostic severity/comorbidity. If the concern is that the outcomes themselves are confounded, that is a generic limitation of observational clinical ML, since most realized clinical outcomes are influenced by downstream treatment and workflow. We hope this addresses the concern underlying W4.
> > >
> > > # Q1
> > > We agree that $J_1/J = 0.3808$ in MIMIC-III is not, by itself, evidence of a near-exact rank-one regime. Our claim is narrower: $J_1/J$ is not itself a measure of rank-one structure; rather, it reflects how much of the realized learned-score objective is captured by the rank-one projected objective, and therefore depends both on the spectrum of $\widetilde W$ and on how the learned score aligns with its singular directions. A direct diagnostic is the singular spectrum of $\widetilde W$, in particular
> > >
> > > $$
> > > e_1 = \sigma_1^2 / \|\widetilde W\|_F^2,
> > > $$
> > >
> > > which is $0.4406$ for MIMIC-III and $0.4899$ for MIMIC-IV. We interpret this as indicating a meaningful leading shared direction in MIMIC-III, together with additional higher-order structure.
> > >
> > > MIMIC-III is a challenging setting for a rank-one interpretation because ICU use is almost universal in this cohort: any ICU stay has prevalence $\approx 0.98$, with median time-to-ICU about $0.09$ hours, $\approx 59\%$ within 1 hour, and $\approx 82\%$ within 24 hours. As a result, the ICU label used in our experiments is effectively a **late ICU transfer** label ($>24$h; prevalence $\approx 0.176$), rather than a broad ICU-use endpoint. This label is therefore less cleanly aligned with the same shared severity axis as mortality or long stay, which helps explain the weaker rank-one behavior in MIMIC-III. By contrast, the larger $J_1/J = 0.6194$ in MIMIC-IV is more consistent with a stronger shared severity direction.

---

### Official Review · Reviewer_88QL · 2026-03-12

**Soundness:** 3
**Presentation:** 2
**Significance:** 3
**Originality:** 3
**Overall Recommendation:** 3
**Confidence:** 3

**Summary:**

The authors proposed a Machine-Learned Comorbidity Index (MLCI) that learns a single scalar score from comorbidity codes by maximizing the nHSIC between the score and multiple clinical outcomes. By using a kernel-based objective, the method aims to capture nonlinear relationships between comorbidities and outcomes without relying on manually engineered interaction terms. The learned score can be reused across different clinical outcomes through post-hoc modeling and present theoretical analysis on when such a unified patient ordering is possible. Empirically, the evaluation focuses on dependence-based metrics such as distance correlation and mutual information, comparing mainly against other single-index scoring approaches on ICU datasets

**Compliance With Llm Reviewing Policy:**

Affirmed.

**Final Justification:**

Dear AC,

While most of the issues I raised have been addressed, my central concern regarding the clinical usefulness of the proposed score remains unresolved. In particular, MLCI (as well as traditional comorbidity indices) does not account for the temporal progression of the underlying disease or patient condition. At the same time, its black-box nature limits interpretability—specifically, it remains unclear which comorbidities are most influential (the provided leave-one-out post-hoc analysis only shows a linear impact) and how their interactions contribute to the outcome. So, I have maintained my original score weak reject.

**Key Questions For Authors:**

-	MIMIC is inherently longitudinal, where the specific temporal trajectory of comorbidities often dictates the patient's prognosis. Could the authors clarify how the temporal evolution of these comorbidities is factored into the final score? Specifically, does the model distinguish between a comorbidity that was present at admission versus one that developed later in the trajectory, and how does this affect the calculation of the final target outcome, which is only available at the end of the comorbidity sequence?
-	Regarding Weakness #1: How does the proposed method’s clinical utility differ from that of a baseline deep survival or classification model optimized for predicting the targer clinical outcome? Given that both architectures result in a one-dimensional (i.e. single-indexed) score, the manuscript would benefit from a head-to-head comparison showing where the dependence-based ordering succeeds while standard BCE-optimized logits fail.
-	The authors set the RBF bandwidth $\sigma$ using a median heuristic on training scores. Since the nHSIC objective is highly sensitive to this parameter, how stable are the rankings if $\sigma$ is shifted?

**Strengths And Weaknesses:**

**Strength:**

-	The authors tackle an important problem of developing a unified, data-driven framework for clinical risk assesment that overcomes the mortality-centric and linear limitations of traditional comorbidity indices.
-	By optimizing a kernel-based objective (nHSIC), the proposed model is able to capture nonlinear relationships between risk factors and outcomes without requiring manually engineered interaction terms.
-	The learned score can also be adapted to different clinical outcomes by fitting outcome-specific models post hoc using the same score.

**Weakness:**

-	A core requirement of a comorbidity index is explicit transparency -- specifically, the ability to delineate exactly how individual diagnoses integrate to the final score. By employing a black-box neural architecture, the clinical utility of the MLCI is inherently limited. The proposed method fails to provide a white-box perspective; while the DeepSets-style aggregation can capture complex interactions and can be applied to any combinations of comorbidities, it does not quantify the specific magnitude or direction of influence for individual comorbidities. Without an explicit feature-attribution mechanism, clinicians cannot verify the underlying drivers of a patient's risk, making the score difficult to adopt in practice.
-	The evaluation mainly relies on distance correlation and mutual information and is compared primarily against benchmark methods that produce single-index scores. However, a more informative evaluation, such as using a proper scoring rule or assessing how predictive and discriminative the learned score is, would be more helpful to understand the behavior of the proposed score. For example, one could train a simple logistic regression model using the learned score as the only input and report standard metrics such as AUROC, AUPRC, or Brier score. In addition, stronger baselines would include deep learning-based predictive models that take comorbidities as input and directly predict the target clinical outcomes.
-	The flow of the theoretical results feels somewhat disjointed. The transition from maximizing general kernel dependence (nHSIC) to the certified fraction of a two-level monotone threshold (Lemma 6.2) requires more clear descriptions. Currently, the derivation of the threshold feels isolated from the earlier discussion on shared latent signals.
- Applying other types of EHR datasets, such as EHRShot or MIMIC-ER, could help evaluate how well the proposed score generalizes beyond ICU settings.

---

> ### Author Rebuttal · Authors · 2026-03-30
>
> We thank the reviewer for their comments.
>
> # W1: Interpretability Analysis
>
> While MLCI is a black box model, we can leverage post-hoc feature attribution to understand the drivers of the risk.  To this end, we performed a **leave-one-diagnosis-out attribution analysis** [1] on held-out admissions in MIMIC-III/IV. For an admission with diagnoses $x$ and score $s(x)$, we remove one code $c$ at a time and compute $\Delta(c)=s(x)-s(x\setminus c)$. Positive $\Delta(c)$ increases the MLCI score; negative $\Delta(c)$ decreases it. This yields direct diagnosis-level explanations of the learned scalar at both the patient and cohort levels. We applied this analysis to the **500 highest-scoring held-out admissions** in each dataset. Across both datasets, MLCI is **not** simply counting diagnoses: a small set of prefixes consistently drives the score upward or downward.
>
> | **Dataset** | **Positive prefixes (mean $\Delta$)**    | **Negative prefixes (mean $\Delta$)**       |
> | ----------- | ---------------------------------------- | ------------------------------------------- |
> | MIMIC-III   | V667 (1.437), 3481 (1.001), 3484 (0.783) | 4824 (-0.394), 7288 (-0.361), 2859 (-0.348) |
> | MIMIC-IV    | Z515 (0.974), I469 (0.778), Z66 (0.422)  | F05 (-0.362), M810 (-0.292), G892 (-0.242)  |
>
> The same pattern appears at the patient level: very high scores are typically driven by only a few dominant prefixes rather than all diagnoses contributing equally.
>
> | **Dataset** | **Example admission** | **Top contributing prefixes ($\Delta$)** |
> | ----------- | --------------------- | ---------------------------------------- |
> | MIMIC-III   | 142501 / 3.355        | 3481 (1.235), 4275 (0.766), 5849 (0.242) |
> | MIMIC-IV    | 26980076 / 4.103      | Z515 (3.868), Z66 (0.916), J690 (-0.154) |
>
> We leave white-box style extension of MLCI to future work.
>
> References
>
> [1] Occlusion-Based Explanations in Deep Recurrent Models for Biomedical Signals
>
> # W2: Downstream Prediction
>
> We thank the reviewer for this suggestion. To complement the dependence-based metrics in the main paper, we added a supplementary downstream prediction evaluation on MIMIC III comparing both scalar-index methods (CCI, ECI, CCI+ECI, MLCI) and stronger classical ML/deep baselines trained directly on diagnosis codes. For scalar-index methods, we fit a one-dimensional logistic regression on the validation split using only the scalar score $s(x)$ and report AUROC, AUPRC, and Brier on the test split.
>
> please view results here: https://anonymous.4open.science/r/PredictionEvaluation--21DE/Evaluation.png
>
> **Downstream prediction results.** MLCI is strongest on the mortality-related outcomes and remains competitive elsewhere.
>
> ## W4
>
> In response to this weakness, we have conducted additional experiments on EHRShot dataset. Please see our response W1 to reviewer vXge.
>
> ### Q1:
>
> MLCI is designed as an admission-level scalar summary of comorbidity burden and clinical severity, in the same general spirit as indices such as CCI and ECI, rather than as a sequence model for disease progression. In the current formulation, the model aggregates the diagnoses observed for an admission into a single score, without explicitly modeling temporal order or separately parameterizing diagnoses present at admission versus those recorded later in the stay. Because the score is computed from the diagnoses available at a given time, it can be recomputed as the patient's condition evolves, so that newly observed complications are naturally reflected in an updated severity estimate.
>
> ### Q2:
>
> The deep learning baselines already included in the paper are BCE-trained direct predictive models that take the same diagnosis-code set as input and produce a single scalar logit, so they provide the relevant point of comparison. Regarding the downstream prediction analysis, please view newly performed downstream prediction results above. The key distinction is that MLCI is not intended as a task-specific predictor for a single endpoint, but as a shared admission-level comorbidity index learned to preserve a common ordering across multiple outcomes. In contrast, BCE-trained classifiers optimize one target directly and may yield logits that are strong for that endpoint but less suitable as a reusable cross-outcome scalar index.
>
> ### Q3:
>
> Since nHSIC uses an RBF kernel on the learned scalar score, we retrained the model with $\sigma_{\mathrm{train}}^{(\gamma)}=\gamma\widehat{\sigma}_{\mathrm{median}}$ for $\gamma\in{0.50,0.75,1.00,1.25,2.00}$ around the median-heuristic baseline, then compared each setting to $\gamma=1.00$ using ranking stability metrics and downstream one-score prediction.
>
> please view results here: https://anonymous.4open.science/r/RBFBandwidthSensitivity-D3C5/RBFBandwidthResults.png
>
> **Results.** The learned ordering is robust to moderate bandwidth shifts: rankings remain highly stable on MIMIC-IV, and broadly stable on MIMIC-III.

---

> > ### Author Rebuttal · Reviewer_88QL · 2026-04-03
> >
> > I thank the authors for the response. I have still have remaining concerns:
> >
> > - While the authors position MLCI as a comorbidity index, it is unclear why the evaluation is conducted on a dataset such as MIMIC, where longitudinal information during ICU stays plays a critical role. The current results suggest that a set of diagnoses at admission alone is highly predictive of ICU outcomes, seemingly independent of the measurements and clinical interventions that occur over time. This pattern is observed not only for the proposed MLCI but also for the DeepSet-based variants, raising concerns about the validity and interpretation of the results.
> >
> > - From a clinical perspective, a key question is how comorbidities evolve over time and contribute to disease progression and eventual outcomes. For example, the temporal ordering and longitudinal development of comorbidities are likely to carry important information. However, the proposed comorbidity score does not appear to capture such dynamics, despite relying on a black-box deep neural network.
> >
> > - Finally, the choice of baseline is limited. The current baseline essentially adopts a DeepSet-style architecture, which treats inputs as unordered sets. A more appropriate comparison would include models that can capture temporal structure, particularly those that explicitly model ordering and progression over time.

---

> > > ### Author Response · Authors · 2026-04-04
> > >
> > > We thank the reviewer for this response. We use MIMIC because it is a large, widely used critical care benchmark, and our diagnosis-only setup follows prior MIMIC-based ICU risk modeling that uses Charlson- and Elixhauser-style comorbidity information as predictors [1,2]. The reviewer is correct that richer ICU information would likely improve performance, but we intentionally excluded it to remain comparable to CCI/ECI. It is very likely that downstream results (e.g, AUROC of 0.829 and AUPRC of 0.485 for Mortality Prediction) can be further improved by leveraging more information.
> > >
> > > We also agree that temporal evolution is important. We did not include temporal structure in our formulation as we wanted the score to remain analogous to CCI/ECI. Our framework can be extended to temporal settings, and following the reviewer’s suggestion, we conducted two additional experiments. For EHRShot, we added an intra-visit temporal ordering modeling experiment. Each example was restructured as an ordered diagnosis sequence within a single visit by sorting diagnoses by recorded time and converting them to truncated ICD prefix tokens. For this experiment, we also changed our MLCI encoder from original DeepSets-style set encoder to a Transformer encoder, so that MLCI could use sequential structure within diagnosis. MLCI performed best on all four EHRShot tasks.
> > >
> > > | Model       |     M AUC |   M AUPRC |   M Brier |   30M AUC | 30M AUPRC | 30M Brier |   LOS AUC | LOS AUPRC | LOS Brier | READMIT AUC | READMIT AUPRC | READMIT Brier |
> > > | ----------- | --------: | --------: | --------: | --------: | --------: | --------: | --------: | --------: | --------: | ----------: | ------------: | ------------: |
> > > | Transformer |     0.804 |     0.125 |     0.038 |     0.681 |     0.079 |     0.046 |     0.662 |     0.425 |     0.126 |       0.671 |         0.478 |         0.273 |
> > > | LSTMs       |     0.707 |     0.062 |     0.039 |     0.682 |     0.068 |     0.046 |     0.557 |     0.237 |     0.135 |       0.665 |         0.492 |         0.274 |
> > > | GRU         |     0.684 |     0.071 |     0.039 |     0.650 |     0.074 |     0.047 |     0.537 |     0.226 |     0.137 |       0.674 |         0.509 |         0.273 |
> > > | HITANET     |     0.765 |     0.091 |     0.035 |     0.651 |     0.063 |     0.044 |     0.748 |     0.519 |     0.119 |       0.673 |         0.477 |         0.280 |
> > > | MLCI        | **0.915** | **0.153** | **0.013** | **0.717** | **0.094** | **0.025** | **0.897** | **0.726** | **0.077** |   **0.731** |     **0.611** |     **0.205** |
> > >
> > > *Average performance across three runs for each model on the EHRSHOT.*
> > >
> > > For MIMIC-III, we conducted a visit-level longitudinal modeling experiment, motivated by prior longitudinal EHR models such as HiTANet [3]. Each example consists of a target admission with all previous admissions for the same patient, ordered by time. Diagnosis codes within each visit are compressed into a visit-level representation, and the ordered sequence of visits is then modeled to produce a single scalar score for target admission. For MLCI, we also replaced the original DeepSets encoder with a Transformer-based temporal encoder. MLCI performed best on Length of stay and ICU transfer and achieved the lowest Brier scores overall.
> > >
> > > | Model       |     M AUC |   M AUPRC |   M Brier |   30M AUC | 30M AUPRC | 30M Brier |   LOS AUC | LOS AUPRC | LOS Brier | ICU AUROC | ICU AUPRC | ICU Brier |
> > > | ----------- | --------: | --------: | --------: | --------: | --------: | --------: | --------: | --------: | --------: | --------: | --------: | --------: |
> > > | Transformer |     0.794 |     0.337 |     0.107 |     0.760 |     0.322 |     0.120 |     0.773 |     0.741 |     0.263 |     0.668 |     0.299 |     0.149 |
> > > | LSTM        |     0.792 |     0.329 |     0.105 |     0.759 |     0.324 |     0.118 |     0.776 |     0.740 |     0.269 |     0.664 |     0.302 |     0.147 |
> > > | GRU         |     0.794 |     0.321 |     0.112 | **0.764** |     0.319 |     0.123 |     0.776 |     0.742 |     0.255 |     0.664 |     0.292 |     0.151 |
> > > | HITANET     | **0.794** | **0.342** |     0.105 |     0.761 | **0.324** |     0.118 |     0.773 |     0.745 |     0.267 |     0.661 |     0.302 |     0.148 |
> > > | MLCI        |     0.757 |     0.310 | **0.093** |     0.732 |     0.301 | **0.113** | **0.777** | **0.751** | **0.193** | **0.673** | **0.325** | **0.141** |
> > >
> > > *Average performance across three runs for each model on the MIMIC III.*
> > >
> > > These experiments show that MLCI is not tied to the original DeepSets encoder and can compete/outperform against strong deep learning temporal baselines
> > >
> > > [1] Using Demographic Factors and Comorbidities to Develop a Predictive Model for ICU Mortality in Patients with Acute Exacerbation COPD
> > >
> > > [2] Elixhauser comorbidity measures-based risk factors associated with 30-day mortality in elderly population after femur fracture surgery: a propensity score-matched retrospective case-control study
> > >
> > > [3] HiTANet

---

### Decision · Program_Chairs · 2026-04-30

**Decision:**

Accept (regular)

**Comment:**

This work proposes an approach to learning a Machine-Learned Comorbidity Index (MLCI) to serve as a univariate descriptor of patient comorbidity across several clinical outcomes, motivated by a desire to capture information not well described traditional comorbidity scores (Charloson and Elixhauser). The reviews for the work were mixed. In my view, the authors have provided a convincing rebuttal (including several new experiments) that addresses the substantive issues raised. To highlight a few aspects of the reviews and the discussion period:
* Reviewer 88QL raised several questions (detailed below). It should be noted that while the reviewer provided a final justification and weak rejection recommendation, the time that their review was revised earlier than the author’s reply rebuttal comment (which introduced several key experimental results), indicating that the final justification does not incorporate information from the final author response:
  * The reviewer argued that it is necessary that the score be interpretable if it is to be clinically useful. In response the authors provided a new analysis using post-hoc interpretability methods that provides some insight into the extent to which the score captures nonlinear relationships. This analysis is appreciated, but arguably the reviewer’s concern is misplaced, as there is active debate as to whether interpretability/explainability approaches are actually necessary or appropriate for clinical AI systems (see Ghassemi et al 2021, Lancet Digital Health, “The false hope of current approaches to explainable artificial intelligence in health care”).
  * The reviewer asked that the predictive value of the score be compared to direct supervised learning baselines and that the work be replicated on other datasets. The authors added several new experiments that replicate the approach on the EHRShot dataset and compare to supervised learning baselines, including sophisticated deep learning ones. The results demonstrate that MLCI is competitive with those baselines.
  * The reviewer raised a concern that the current score does not capture complex temporal dynamics. The authors conducted additional experiments where they modified MLCI to account for temporal dynamics with a transformer based encoder rather than a DeepSets encoder. The results show that the approach does generalize to a temporal representation and that the results are not sensitive to the choice of encoder. This question is listed as a key unresolved point in the reviewer’s response and it is one that was addressed in the final author response submitted after the reviewer posted their final justification.
* Reviewer gw2i raised several questions, primarily related to the premise of the work as a whole:
  * A key argument relates to whether it is reasonable to assume that several binary outcomes are coupled through a single unobserved, real-valued latent severity score. As evidence for the critique, the reviewer points to experiments demonstrating that the first-singular dimension accounts for only roughly 50% the variability. In my view, the author’s response and the perspective presented in the paper is reasonable. It is clear that the assumption of the rank-one structure is a modeling assumption that will not be appropriate in all cases, but the approach does permit post-hoc analysis of the extent to which the assumption holds through analysis of the variability explained by the first-singular dimension and analysis into the structure of task-specific residuals. The reviewer’s suggestion for a reframing is also reasonable: “Based on their real data experiments, a more plausible assumption is that T binary clinical outcomes are coupled through an unobserved low-dimensional vector Z representing overall sickness/disease burden and the direction with the highest variability has some valuable clinical utility.” Further minor revision for clarity along these lines would improve the paper.
  * The reviewer was skeptical that MLCI is clinically useful. The authors point to the various ways that univariate scores are currently used as evidence for the potential utility of MLCI. I find these arguments to be reasonable.
One line of critique relates to the need to account for confounding by treatment in order to use the score for clinical decision making, given that it is learned from observational data. While this is in principle true, this is a concern that affects essentially every clinical predictive model learned from observational data, and is not unique to this work. In my view, this is not a fundamental concern for this work, as scores of this nature can still be clinically useful even if their validity is appropriately interpreted as internal to the clinical context in which they were developed.
  * Regarding novelty, the reviewer points to limited methodological novelty. In my view, the level of novelty is reasonable for an applications paper motivated to address a domain-specific challenge.
* Reviewer pN3G asked several detailed questions regarding external validity and sensitivity analysis / ablation of various components. The EHRShot experiments mentioned above address external validity and the authors conduct various new experiments to address the other points. The reviewer’s revised recommendation is to accept the paper.
* Reviewer vXge provided a positive assessment (weak accept), but a shorter review. The explicit critiques were (1) a need for external validation and (2) need for interpretability. Both of these are addressed by the response to 88QL.
* A point flagged by multiple reviewers, but not elaborated upon in the discussion period, is that the paper does not include a section that discusses limitations. This is a legitimate weakness of the work. Such a section would be appreciated, especially if paired with empirical and/or theoretical analysis of failure modes of the method in settings where the underlying assumptions do not hold.

For the reasons mentioned above, I recommend that this paper be accepted for publication, but encourage the authors to revise to address the remaining minor critiques.